# Superior colliculus drives stimulus-evoked directionally biased saccades and attempted head movements in head-fixed mice

Sebastian H Zahler[1,2†], David E Taylor[1,2†], Joey Y Wong[1], Julia M Adams[1], Evan H Feinberg[1,2,3]*

[1]Department of Anatomy, University of California, San Francisco, San Francisco, United States; [2]Neuroscience Graduate Program, University of California, San Francisco, San Francisco, United States; [3]Kavli Institute for Fundamental Neuroscience, University of California, San Francisco, San Francisco, United States

*For correspondence: evan.feinberg@ucsf.edu

†These authors contributed equally to this work

Competing interest: The authors declare that no competing interests exist.

**Abstract:** Animals investigate their environments by directing their gaze towards salient stimuli. In the prevailing view, mouse gaze shifts entail head rotations followed by brainstem-mediated eye movements, including saccades to reset the eyes. These 'recentering' saccades are attributed to head movement-related vestibular cues. However, microstimulating mouse superior colliculus (SC) elicits directed head and eye movements resembling SC-dependent sensory-guided gaze shifts in other species, suggesting that mouse gaze shifts may be more flexible than has been recognized. We investigated this possibility by tracking eye and attempted head movements in a head-fixed preparation that eliminates head movement-related sensory cues. We found tactile stimuli evoke directionally biased saccades coincident with attempted head rotations. Differences in saccade endpoints across stimuli are associated with distinct stimulus-dependent relationships between initial eye position and saccade direction and amplitude. Optogenetic perturbations revealed SC drives these gaze shifts. Thus, head-fixed mice make sensory-guided, SC-dependent gaze shifts involving coincident, directionally biased saccades and attempted head movements. Our findings uncover flexibility in mouse gaze shifts and provide a foundation for studying head-eye coupling.

## Editor's evaluation

Animals investigate their environments by directing their gaze towards salient stimuli. However, whether non-foveal mammals like mice can make directed saccades independent of head movements in response to sensory stimuli remains unclear. Feinberg et al. systematically investigate how tactile, auditory and visual stimuli drive saccade and head movement patterns. Mice make sensory-guided gaze shifts that depend on superior colliculus and involve coincident, directionally biased saccades and attempted head movements, with flexibility in saccade kinematics relative to attempted head movements.

## Introduction

Natural environments are complex and dynamic, and animals frequently redirect their gaze to scrutinize salient sensory stimuli. Gaze shifts employ head and eye movement coupling strategies that depend on context and can vary between species (*Goldring et al., 1996*; *Land, 2019*; *Land and Nilsson, 2012*; *Populin, 2006*; *Populin and Rajala, 2011*; *Populin et al., 2004*; *Ruhland et al., 2013*;

*Tollin et al., 2005*). Mice are an increasingly important model organism in vision research, yet the strategies they use to shift their gaze remain incompletely understood. Revealing these strategies is essential to understanding mouse visual ethology and the underlying neural mechanisms.

The prevailing view holds that species whose retinae lack high-acuity specializations (afoveates) such as mice generate gaze shifts driven by head movements and followed by 'recentering' saccades (*Land and Nilsson, 2012*). Indeed, recent studies tracking head and eye movements in freely moving mice found that spontaneous and visually evoked mouse gaze shifts matched this description (*Meyer et al., 2018*; *Meyer et al., 2020*; *Michaiel et al., 2020*; *Payne and Raymond, 2017*). Specifically, during a gaze shift, slow eye movements stabilize the retinal image by countering the rotation of the head and are punctuated by fast saccadic eye movements to recenter the eyes in the orbits as they approach the end of their range of motion. These recentering saccades—also known as 'compensatory' saccades or the quick phase of nystagmus—are centripetal, occur in the direction of the head movement, and are thought to be driven by vestibular or optokinetic signals acting on circuits in brainstem (*Curthoys, 2002*; *Hepp et al., 1993*; *Kitama et al., 1995*; *Meyer et al., 2020*; *Michaiel et al., 2020*; *Payne and Raymond, 2017*). These recent observations have buttressed the view that gaze shifts in mice and other afoveates are led by head movements, with eye movements made only to compensate for the effects of head movements. In contrast, primates and other foveate species are capable of an additional form of gaze shift led by directed saccades, with or without directed head movements, to redirect their gaze toward salient stimuli (*Bizzi et al., 1972*; *Freedman, 2008*; *Lee, 1999*; *Zangemeister and Stark, 1982*). Directed saccades differ from recentering saccades in that they have endpoints specified by the location of the stimulus (and therefore are often centrifugally directed), typically occur simultaneously with head movements during gaze shifts, and are driven by midbrain circuits, particularly the superior colliculus (SC). To date, there is no behavioral evidence that mice or any afoveate species generate directed saccades or gaze shifts not initiated by head movements.

However, three observations are inconsistent with the model that saccades in mice are exclusively recentering and made to compensate for head movements. First, mouse saccades are not only a product of vestibular or optokinetic cues, because head-fixed mice, in which these signals do not occur, generate saccades, albeit less frequently. Second, neuroanatomical and functional studies suggest that the circuits that underlie directed saccades are conserved in mice (*May, 2006*; *Sparks, 1986*; *Sparks, 2002*). Specifically, microstimulation of the mouse superior colliculus (SC) showed that it contains a topographic map of saccade and head movement direction and amplitude (*Masullo et al., 2019*; *Wang et al., 2015*) roughly aligned with maps of visual, auditory, and somatosensory space (*Dräger and Hubel, 1975*). These SC sensory and motor maps resemble those believed to underlie primates' and cats' ability to make gaze shifts led by directed saccades toward stimuli of these modalities (*Sparks, 1986*; *Sparks, 2002*). Third, saccade-like eye movements in the absence of head movements have occasionally been observed in freely moving mice, albeit infrequently and usually in close proximity to head movements (*Meyer et al., 2018*; *Meyer et al., 2020*; *Michaiel et al., 2020*).

We therefore hypothesized that mice innately generate gaze shifts that incorporate directionally biased saccades. We predicted that this ability was obscured in previous studies for several reasons. First, in freely moving mice it is difficult to uncouple the contributions of reafferent vestibular and optokinetic inputs from those of exafferent (extrinsic) sensory inputs to saccade generation. Second, previous analyses in mice were mostly confined to spontaneous or visually guided gaze shifts, and there is evidence in humans, non-human primates, and cats that gaze shifts in response to different sensory modalities can involve distinct head-eye coupling strategies (*Goldring et al., 1996*; *Populin, 2006*; *Populin and Rajala, 2011*; *Populin et al., 2004*; *Ruhland et al., 2013*; *Tollin et al., 2005*). Third, in freely moving mice it is difficult to present stimuli in specific craniotopic locations. We therefore reasoned that by using a head-fixed preparation both to eliminate vestibular and optokinetic cues and to present stimuli of different modalities at precise craniotopic locations, we could systematically determine whether mice are capable of gaze shifts not initiated by head movements and involving saccades whose endpoints depend on stimulus location and show different coupling to head movements. We found that tactile stimuli evoke saccades whose endpoints depend on stimulus location, that these saccades are coincident with attempted head rotations, and that these touch-evoked gaze shifts are SC-dependent. Together, these results resolve an apparent discrepancy between mouse

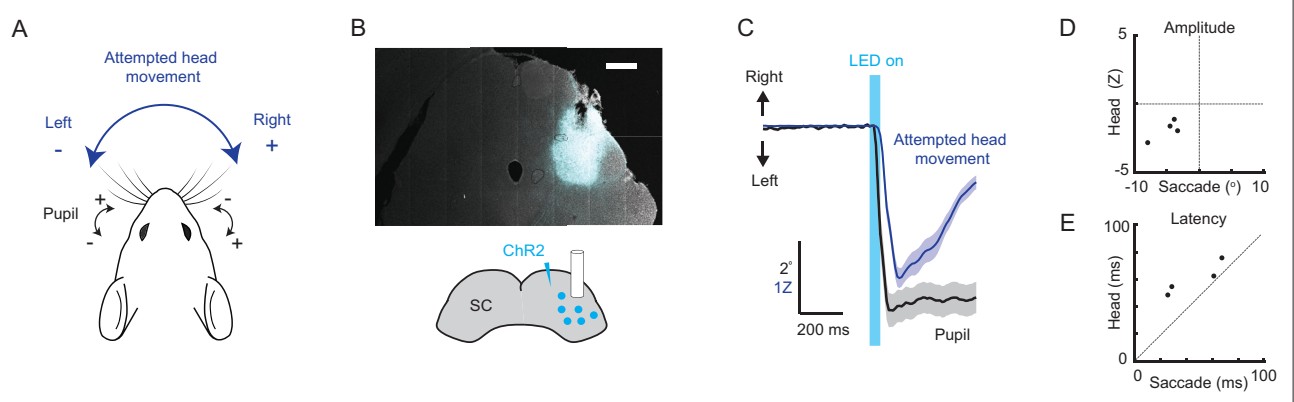

**Figure 1.** Optogenetic stimulation of the superior colliculus evokes coincident and directionally biased attempted head and eye movements. (**A**) Behavioral schematic. Naive mice are head-fixed, both eyes are tracked using cameras, and attempted head rotations are measured using a strain gauge (load cell). In subsequent quantification, eye positions to the right of center (nasal for left eye, temporal for right eye) are positive, and eye positions to the left of center (temporal for left eye, nasal for right eye) are negative, with zero defined as the mean eye position. Likewise, attempted rightward head movements are positive, and leftward head movements are negative. (**B**) Schematic of right SC optogenetic stimulation using ChR2 and example histology for representative mouse. Scale bar, 0.5 mm. (**C**) Mean attempted head (blue) and eye (black) movement traces (n = 44 trials, 4 mice) in the 1 s period surrounding optogenetic stimulation. Optogenetic illumination (1 mW) was delivered for 40 ms. (**D**) Relationship between saccade amplitude and attempted head movement amplitude for individual mice. (**E**) Relationship between saccade latency and head movement latency for individual mice.

neuroanatomy and behavior, demonstrating that head-fixed mice are capable of generating gaze shifts involving directionally biased saccades coincident with attempted head movements.

## Results

### Stimulus-evoked gaze shifts in head-restrained mice

Previous studies established that SC microstimulation or optogenetic stimulation evoke both head and eye movements and that these movements roughly match the topographic map of sensory spatial receptive fields within SC, with each SC hemisphere driving contraversive movements whose amplitudes are larger for more posterior stimulation sites (*Masullo et al., 2019*; *Wang et al., 2015*). However, to our knowledge, a simultaneous measurement of eye and attempted head movements elicited by SC microstimulation had not been conducted. To perform this comparison, we pursued an optogenetic approach. We stereotaxically injected adeno-associated virus (AAV) encoding the light-gated ion channel ChR2 under the control of a pan-neuronal promoter and implanted a fiber optic in right SC (*Figure 1B*). Several weeks later, we head-fixed mice and used infrared cameras to track both pupils and a strain gauge (also known as a load cell) to measure attempted head rotations (*Figure 1A*). Consonant with previous studies, SC optogenetic stimulation elicited contraversive eye and attempted head movements (*Figure 1C and D*). Strikingly, optogenetically evoked saccades and attempted head movements were roughly coincident, similar to what has been described for SC-dependent sensory-guided gaze shifts in other species and unlike the temporal relationship previously documented for mouse spontaneous and visually evoked gaze shifts (*Figure 1C and E*).

In light of this observation, we hypothesized that mice possess an innate ability to make sensory-evoked gaze shifts that incorporate directionally biased saccades coupled to head movements. To test this hypothesis, we head-fixed naive, wild-type adult animals and tracked eye and attempted head movements. Previous studies in head-fixed mice observed occasional undirected saccades in response to changes in the visual environment (*Samonds et al., 2018*) and visually guided saccades only after weeks of training and at long (~1 s) latencies (*Itokazu et al., 2018*). We therefore tested a panel of stimuli of different modalities to determine whether they could evoke saccades. We began by presenting the following stimuli from a constant azimuthal location: (1) a multisensory airpuff that provides tactile input to the ears and generates a loud, broadband sound; (2) an auditory stimulus consisting of the same airpuff moved away from the animal so as not to provide tactile input; (3) a tactile stimulus consisting of a bar that nearly silently taps the ear; and (4) a visual stimulus consisting

of a bright LED. Stimuli were presented on either side of the animal every 7–12 s in a pseudorandom sequence (*Figure 2A*). The probability of horizontal eye movements increased sharply and significantly above the low baseline level (1.3% ± 0.2%, mean ± s.d.) in the 100ms period following delivery of multisensory airpuffs (ear airpuff: 29.0% ± 7.5%, p < 0.001 paired Student's t-test; whisker airpuff: 12.5% ± 2.3%, p < 0.001), auditory airpuffs (3.5% ± 1.2%, p < 0.05), and tactile stimuli (4.5% ± 0.5%, p < 0.001) and remained slightly elevated for at least 500 ms (*Figure 2B–F and G–K*; *Figure 2—figure supplements 1–2*). In contrast, the probability of saccade generation was not changed by visual stimuli (*Figure 2F and K*; 1.3% ± 0.2%, p = 0.61). We consider these stimulus-evoked eye movements to be saccades because they reached velocities of several hundred degrees per second (*Figure 2— figure supplement 1*), displayed a main sequence, that is, peak velocity scaled linearly with amplitude (*Figure 2—figure supplement 1*), and were bilaterally conjugate (*Figure 2—figure supplement 1*; *Bahill et al., 1975*). As in previous studies, saccade size for temporal-to-nasal movements was slightly larger than for nasal-to-temporal movements (*Meyer et al., 2018*), but this asymmetry was eliminated by averaging the positions of both pupils (before averaging: temporal saccade amplitude = 10.9 ± 0.8° (mean ± s.d.), nasal saccade amplitude = 8.4 ± 1.1°, p = 0.0009; after averaging, leftward amplitude = 10.2 ± 1.0°, rightward amplitude = 9.3 ± 0.4°, p = 0.0986).

We next examined attempted head movements. The baseline frequency of attempted head movements was much higher than that of eye movements (27.8% ± 4.8%, mean ± s.d.). Mirroring results for saccades, auditory, tactile, and audiotactile stimuli evoked attempted head movements but visual stimuli did not (*Figure 2L–P*; auditory: 53.2% ± 21.9%, p = 0.045; tactile: 67.1% ± 10%, p < 0.01; ear airpuff: 86.6% ± 8.7%, p < $10^{-5}$; whisker airpuff: 79.2% ± 9.5%, p < 0.001; visual: 26.9% ± 2.0%, p = 0.0505, paired Student's t-test). These data demonstrate that both auditory and tactile stimuli are sufficient to evoke gaze shifts in head-fixed mice, and that mice make sensory-evoked saccades in the absence of vestibular and optokinetic inputs.

## Tactile stimuli evoke directionally biased eye and attempted head movements

To determine whether sensory-evoked saccades are directionally biased, we asked whether saccade endpoints are dependent on stimulus location. We began by examining the endpoints of saccades evoked by left and right ear airpuffs (*Figure 3A*, *Figure 3—figure supplement 1*). We found that left ear airpuffs evoked saccades with endpoints far left of center (with center defined as the mean eye position), whereas right ear airpuffs evoked saccades with endpoints far right of center (left: –5.4 ± 4.5°, right: 5.4 ± 3.4°, mean ± s.d., p < $10^{-10}$ Welch's t-test, n = 2155 trials). To understand how this endpoint segregation arises, we examined the trajectories of individual saccades (*Figure 3E*). We found that left ear airpuffs elicited nearly exclusively leftward saccades (94.2% ± 3.1%, mean ± s.d., n = 5 mice), whereas right ear airpuffs elicited nearly exclusively rightward saccades (96.0% ± 2.1%)— often from the same eye positions. By definition, from any eye position, one of these directions must lead away from center and is thus centrifugal rather than centripetal. In addition, puff-evoked saccades that began toward the center often overshot to reach endpoints at eccentricities of 5–10 degrees. To further test whether saccade endpoints are specified by stimulus location, we repositioned the airpuff nozzles to stimulate the whiskers and repeated the experiments. We reasoned that saccade endpoints should become less eccentric as stimulus eccentricity decreases. Indeed, airpuffs applied to the whiskers evoked saccades with endpoints central to those evoked by ear airpuff stimulation, such that the ordering of saccade endpoints mirrored that of stimulus locations (Mean endpoints: left ear, –5.4 ± 4.5°; left whiskers, –0.4 ± 3.8°; right whiskers, 0.8 ± 3.9°; right ear, 5.4 ± 3.4°; mean ± s.d., p < 0.05 for all pairwise comparisons, paired two-tailed Student's t-test) (*Figure 3A and B*; *Figure 3— figure supplements 1–2*). In this cohort, the separation between whisker-evoked saccade endpoints, although significant, was small, but in other cohorts we observed larger separation (as well as higher auditory-evoked saccade probabilities) (*Figure 3—figure supplement 3*). Taken together, these data suggest that airpuff-evoked saccades are biased toward particular eye positions that are specified by stimulus location.

We next examined the endpoints of saccades evoked by tactile and auditory stimuli. Similar to the multisensory airpuff stimuli, tactile stimuli delivered to the left and right ears evoked saccades whose endpoints were significantly different (–3.9 ± 5.3° [left] vs. 1.5 ± 5.0° [right]; mean ± s.d., p < $10^{-10}$, Welch's t-test, n = 452 trials) and whose directions were largely opposite from nearly all eye positions

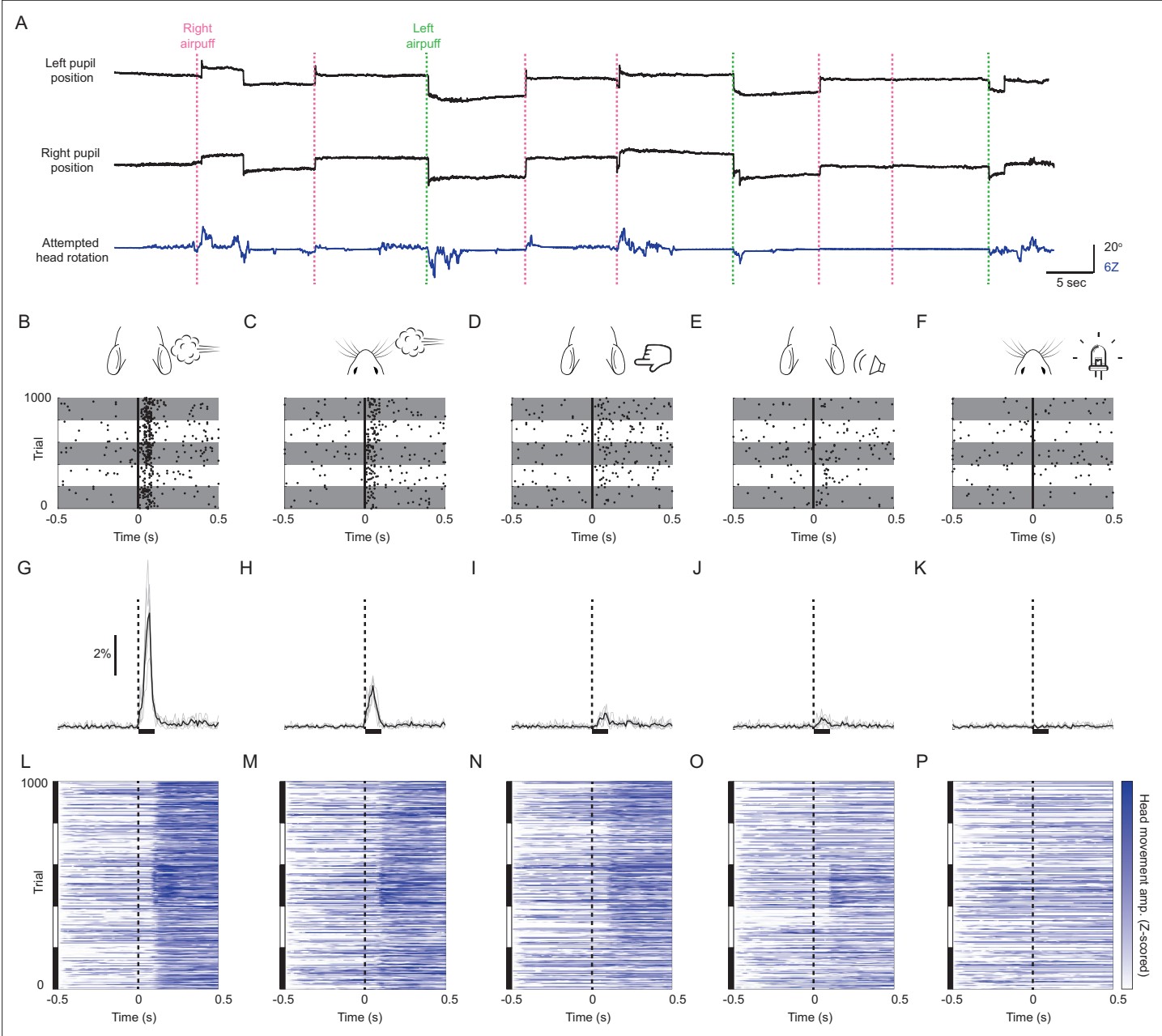

**Figure 2.** Mice innately make sound- and touch-evoked gaze shifts. (**A**) Sample eye and attempted head movement traces. Dashed vertical lines indicate right (green) and left (magenta) ear airpuff delivery. (**B–F**) Saccade rasters for five representative mice in response to (**B**) ear airpuffs, (**C**) whisker airpuffs, (**D**) ear tactile stimuli, (**E**) auditory airpuffs, and (**F**) visual stimuli. Each row corresponds to a trial. Each dot indicates onset of a saccade. Vertical black lines denote time of left or right stimulus delivery. Each gray or white horizontal stripe contains data for a different mouse. n = 1000 randomly selected trials (200 /mouse). (**G–K**) Peri-stimulus time histograms showing instantaneous saccade probabilities in response to (**G**) ear airpuffs, (**H**) whisker airpuffs, (**I**) ear tactile stimuli, (**J**) auditory airpuffs, and (**K**) visual stimuli for mice from (**B–F**). Each light trace denotes a single animal; black traces denote population mean. Dashed lines denote time of stimulus delivery. Horizontal bar indicates the 100ms response window used in subsequent analyses. (**L–P**) Heatmaps of attempted head movements in response to (**L**) ear airpuffs, (**M**) whisker airpuffs, (**N**) ear tactile stimuli, (**O**) auditory airpuffs, and (**P**) visual stimuli for mice from (**B–K**). Each row corresponds to an individual trial from B-F. Black and white bars at left indicate blocks of trials corresponding to each of five different mice. Dashed line denotes stimulus delivery time.

The online version of this article includes the following figure supplement(s) for figure 2:

**Figure supplement 1.** Airpuffs evoke horizontal saccades.

**Figure supplement 2.** Evoked saccades occur within a narrow window after stimulus delivery.

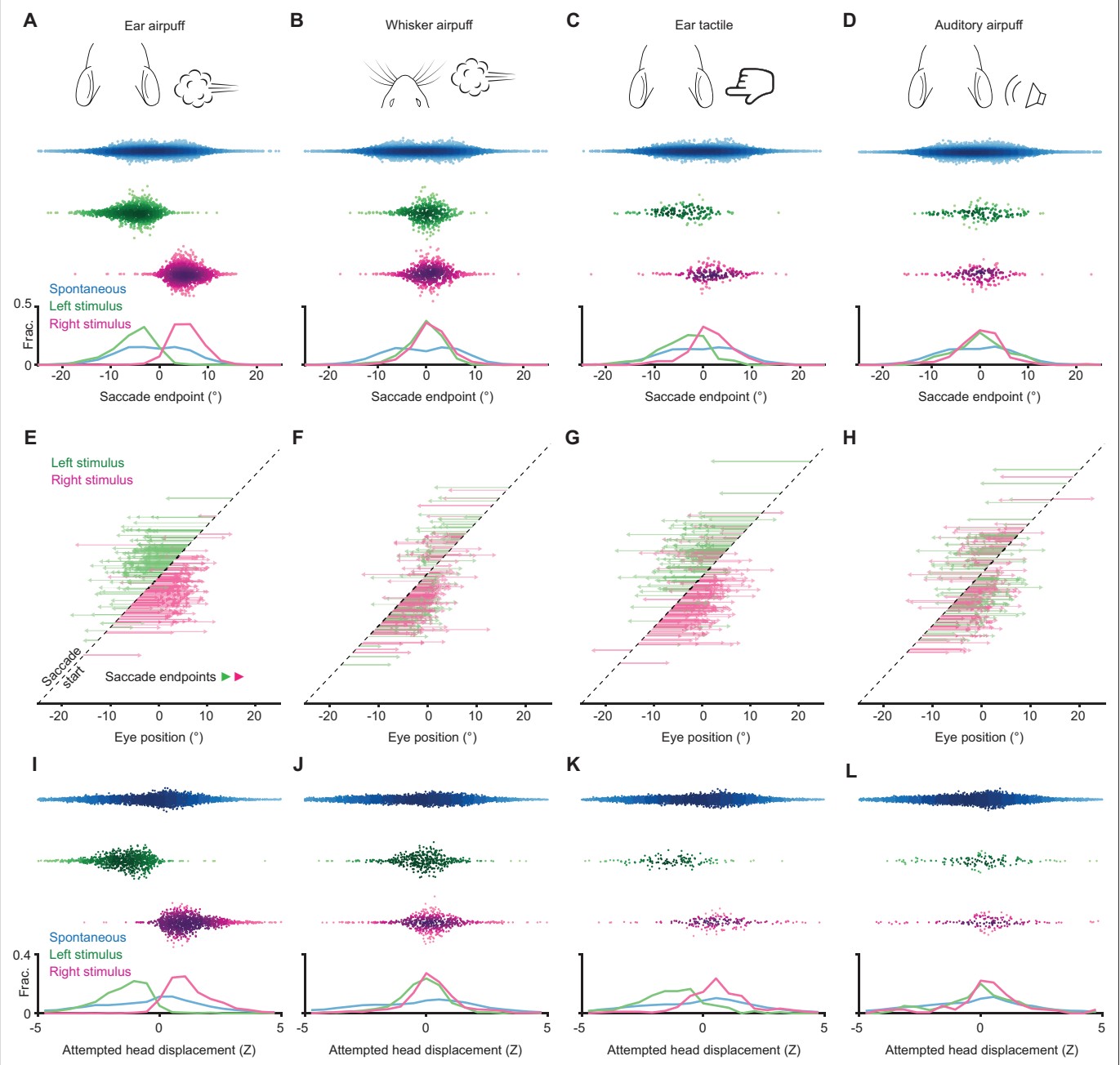

**Figure 3.** Sensory-evoked eye and attempted head movements. (**A–D**) Endpoints for ear airpuff-, whisker airpuff-, ear tactile-, and auditory airpuff-evoked saccades. Top, schematics of stimuli. Middle, scatter plots showing endpoints of all saccades for all animals (n = see below, 5 animals) made spontaneously (blue) and in response to left (green) and right (magenta) stimuli. Darker shading indicates areas of higher density. Bottom, histograms of endpoint distributions for spontaneous and evoked saccades. (**E–H**) Trajectories of individual stimulus-evoked saccades. Each arrow denotes the trajectory of a single saccade. Saccades are sorted according to initial eye positions, which fall on the dashed diagonal line. Saccade endpoints are indicated by arrowheads. Because the probability of evoked gaze shifts differed across stimuli, data for ear and whisker airpuffs are randomly subsampled (15% and 30% of total trials, respectively) to show roughly equal numbers of trials for each condition. (**I–L**) Ear airpuff-, whisker airpuff-, ear tactile-, and auditory airpuff-evoked attempted head displacements associated with saccades in A-D. Top, scatter plots showing displacements of attempted head movements associated with saccades made spontaneously (blue) and in response to left (green) and right (magenta) stimuli (n = see below, 5 animals). Darker shading indicates areas of higher density. Bottom, histograms of attempted displacement distributions for spontaneous and evoked attempted head movements. Saccade numbers in A-L: ear airpuff sessions, spontaneous = 7146, left ear airpuff-evoked = 942 (141 in E), right ear airpuff-evoked = 1213 (182 in E); whisker airpuff sessions: spontaneous = 7790, left whisker airpuff-evoked = 440 (132 in F), right whisker airpuff-evoked = 606 (181 in F); ear tactile sessions, spontaneous = 6706, left ear tactile-evoked = 133, right ear tactile-evoked = 186; auditory sessions, spontaneous = 10240, left auditory-evoked = 140, right auditory-evoked = 158.

*Figure 3 continued on next page*

Figure 3 continued

The online version of this article includes the following figure supplement(s) for figure 3:

**Figure supplement 1.** Saccade endpoints as a function of initial eye position.

**Figure supplement 2.** Endpoints of airpuff-evoked saccades are ordered according to site of stimulation.

**Figure supplement 3.** Endpoints and trajectories of sensory-evoked saccades for an additional cohort of mice.

(*Figure 3C and G*, *Figure 3—figure supplement 1*) (left stimuli evoked 77.5% ± 23.7% leftward saccades, right stimuli evoked 78.3% ± 11.6% rightward saccades, n = 5 mice). This result suggests that tactile stimuli are sufficient to induce gaze shifts that involve directionally biased saccades. We next examined the endpoints and trajectories of saccades evoked by left and right auditory stimuli (*Figure 3D and H*, *Figure 3—figure supplement 1*). Strikingly, the endpoint locations did not differ significantly for saccades elicited by left and right auditory stimuli and were located centrally (0.1 ± 5.1° (left) vs. −0.1 ± 5.1° (right); mean ± s.d., p = 0.72, Welch's t-test, n = 298 trials). Because we had fewer trials with sound-evoked gaze shifts overall, to confirm that this lack of significant endpoint separation was not a result of lower statistical power, we repeated our analyses on equal numbers of sound- and touch-evoked saccades sampled at random, once again observing that left and right ear airpuff-, whisker airpuff-, and ear tactile-evoked saccade endpoints were significantly different (ear airpuff, $p < 10^{-10}$; whisker airpuff, p = 0.0039; ear tactile, $p < 10^{-10}$; auditory airpuff, p = 0.50; Welch's t-test). The central endpoints of sound-evoked saccades arose because, in contrast to touch-evoked saccades, saccades evoked by both right and left stimuli traveled in the same, centripetal direction from all initial eye positions: rightward from eye positions to the left of center, and leftward from eye positions to the right of center (fraction centripetal: left airpuff for initial eye positions left of center, 0.89 ± 0.08; left airpuff for initial eye positions right of center, 0.90 ± 0.09; right airpuff for initial eye positions left of center, 0.87 ± 0.07; right airpuff with initial eye positions right of center, 0.76 ± 0.12). We compared the mean sound-evoked saccade endpoint to the mean overall eye position and found no significant difference, suggesting that sound-evoked saccades function to center the eye (p = 0.93, one-sample Student's t-test) (*Land and Nilsson, 2012*; *Meyer et al., 2018*; *Meyer et al., 2020*; *Michaiel et al., 2020*; *Paré and Munoz, 2001*; *Tatler, 2007*). These responses are unlikely to be due to an auditory startle response, which is elicited by stimuli louder than 80 dB, because the airpuffs were <65 dB (*Gómez-Nieto et al., 2020*). Thus, both the auditory and tactile components of the airpuff stimuli are sufficient to evoke saccades but only tactile stimulation evokes directionally biased saccades whose endpoints are specified by the site of stimulation.

We next analyzed the relationship between stimuli and attempted head movements. The amplitude distributions for attempted head movements to left and right ear airpuff stimuli were well separated (left: −1.56 ± 1.25 Z, right: 1.29 ± 1.16 Z, mean ± s.d., $p < 10^{-10}$ Welch's t-test, n = 2155 trials), mirroring the separation of the endpoints of saccades elicited by left and right stimuli (*Figure 3I*). Similarly, whisker and tactile stimuli elicited attempted head movements whose amplitude distributions were separated (left whiskers: −0.22 ± 1.58 Z, right whiskers: 0.20 ± 1.42 Z, $p < 10^{-4}$, n = 1046; left tactile: −1.45 ± 1.84 Z, right tactile: 0.55 ± 1.75 Z, $p < 10^{-10}$, n = 319; mean ± s.d., Welch's t-test) but less so than those of ear airpuffs (*Figure 3J and K*). Left and right auditory airpuffs elicited attempted head movements whose distributions overlapped (left: −0.25 ± 2.04 Z, right: 0.07 ± 1.94 Z, mean ± s.d., p = 0.16, Welch's t-test, n = 298 trials), similar to what was observed for saccades (*Figure 3L*). Thus, it appeared that the patterns of evoked eye and attempted head movements were similar for a given stimulus but different across stimuli, with tactile but not auditory stimuli able to evoke directionally biased movements.

## Stimulus-evoked gaze shifts lack pre-saccadic head movements

Having observed that mice are able to make stimulus-evoked directionally biased gaze shifts, similar to what we had observed in SC optogenetic stimulation, we next compared the relationships between attempted head rotations and saccades during spontaneous and stimulus-evoked gaze shifts. We first analyzed the relative timing of eye and attempted head movements. Whereas our SC optogenetic stimulation evoked roughly coincident saccades and attempted head movements, previous studies found that on average, spontaneous and visually evoked saccades in freely moving mice are preceded by head rotations, and that spontaneous saccades in head-fixed mice are similarly

preceded by slow attempted head rotations (*Figure 1*; *Meyer et al., 2018*; *Meyer et al., 2020*; *Michaiel et al., 2020*; *Payne and Raymond, 2017*). Consistent with these observations, we found that most spontaneous saccades were preceded by attempted head rotations in the same direction as the ensuing saccade (*Figure 4B, D and E*). Interestingly, attempted head rotations during spontaneous saccades appeared biphasic, with a slow phase starting 100–200 ms before saccade onset followed by a fast phase beginning roughly simultaneously with saccade onset (*Figure 4B, D and E*). This biphasic response closely resembles the eye-head coupling pattern reported in freely moving mice (*Meyer et al., 2020*).

We hypothesized that touch-evoked gaze shifts are mediated by SC and therefore that the relative timing of eye and attempted head movements would more closely resemble those observed in other species and in our SC optogenetic stimulation. Consistent with our prediction, ear airpuff-evoked saccades were typically not preceded by slow attempted head rotations but were accompanied by roughly coincident fast attempted head rotations (*Figure 4B, D and F*). On average, these fast attempted head rotations had similar peak velocities and latencies relative to saccade onset to those observed from SC optogenetic stimulation as well as those made during the fast phase of head movements during spontaneous gaze shifts (*Figure 4B and D*). This pattern was mirrored in the average eye and attempted head movement traces for whisker airpuff-, auditory airpuff-, and ear tactile-evoked saccades (*Figure 4—figure supplement 1A-D*).

To better understand how these patterns arose, we examined eye and attempted head movement timing at the single-trial level. For spontaneous gaze shifts, attempted head movement onset fell along a continuum starting well before saccade onset, with attempted head displacement roughly 60% predictive of the direction of the ensuing saccade starting as early as 200 ms before saccade onset and approximately 80% predictive by 100 ms before saccade onset (*Figure 4E and G*). In contrast, for stimulus-evoked gaze shifts, the vast majority of attempted head movements began roughly coincidently with saccade onset, and attempted head displacement was not predictive of head position prior to saccade onset (*Figure 4F and G*). Attempted head velocity showed similar trends (*Figure 4I–K*). Thus, these data suggest that spontaneous gaze shifts are typically preceded by attempted head movements in the direction of the ensuing saccade, whereas stimulus-evoked gaze shifts are not.

We next examined the amplitudes of head and eye movements during spontaneous and stimulus-evoked gaze shifts. During spontaneous gaze shifts, both the slow and fast phases of attempted head rotations were in the same direction as the ensuing saccades and scaled with saccade amplitude (*Figure 4A, B, E, H, K and L*). These data are consistent with eye-head coupling patterns previously observed in both head-fixed and freely moving conditions (*Meyer et al., 2020*). Similarly, stimulus-evoked gaze shifts involved attempted head rotations that were made in the same direction as saccades and scaled with saccade amplitude (*Figure 4A, B, F, H, J, K and L*). Interestingly, stimulus-evoked saccades of a given amplitude were coupled to an average of 24% smaller attempted head movements than were spontaneous saccades (linear regression slopes 0.162 vs 0.214, $p < 10^{-5}$, permutation test) (*Figure 4H*). This difference was due to the slow pre-saccadic attempted head movements observed during spontaneous gaze shifts, as the fast phase of both spontaneous and evoked gaze shifts was nearly identical (*Figure 4L*). To confirm that the differences in coupling we observed were not an artifact of differences in saccade size and starting position between saccade types, we performed an additional analysis using subsets of gaze shifts matched for saccade amplitude and initial eye position, observing the same effects (*Figure 4—figure supplement 2*; linear regression slopes 0.162 vs 0.221, $p < 10^{-5}$, permutation test). To determine whether these differences could be attributed to experience (for example, if animals learn that attempted head movements in response to sensory stimuli are futile) we compared evoked and spontaneous gaze shifts across five sessions and within sessions (*Figure 4—figure supplement 3*). Interestingly, the gain of head-eye coupling did appear significantly lower for evoked but not spontaneous gaze shifts by the fifth session, but at every time point analyzed, evoked gaze shifts involved smaller head movements than did spontaneous gaze shifts (*Figure 4—figure supplement 3*). These data indicate that mice may subtly and slowly change their strategies with experience but differences between spontaneous and evoked gaze shifts do not reflect learning.

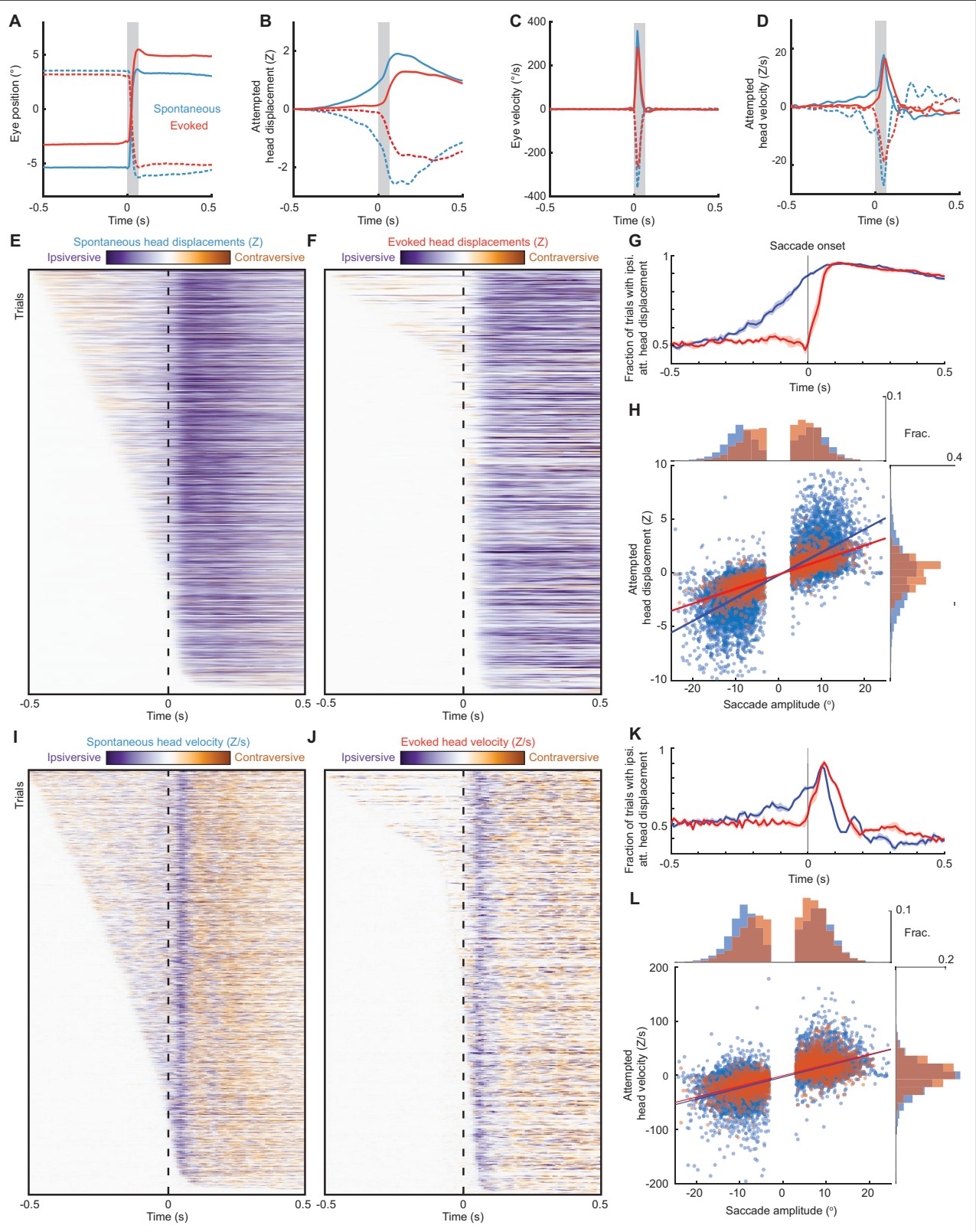

**Figure 4.** Head-eye coupling during spontaneous and touch-evoked gaze shifts. (**A**) Mean trajectories of all rightward (solid traces) and leftward (dashed traces) saccades during spontaneous (blue, n = 7146) and ear airpuff-evoked (red, n = 1437) gaze shifts. Means + s.e.m. (smaller than line width). Gray bar indicates average saccade duration. (**B**) Mean attempted head displacement accompanying rightward (solid traces) and leftward (dashed traces) saccades during spontaneous (blue) and ear airpuff-evoked (red) gaze shifts. (**C**) Mean velocities of all rightward (solid traces) and

*Figure 4 continued on next page*

*Figure 4 continued*

leftward (dashed traces) saccades during spontaneous (blue) and ear airpuff-evoked (red) gaze shifts. (**D**) Mean attempted head movement velocities accompanying rightward (solid traces) and leftward (dashed traces) saccades during spontaneous (blue) and ear airpuff-evoked (red) gaze shifts. (**E, F**) Timing of attempted head movements relative to saccades during all spontaneous (**E**) and ear airpuff-evoked (**F**) gaze shifts. Each row corresponds to a single gaze shift. Darker shades indicate larger attempted head displacement. Purple hues denote attempted displacement in the same direction as the saccade (ipsiversive), and orange hues denote displacement in the opposite direction of the saccade (contraversive). Dashed vertical line indicates time of saccade onset. Trials are sorted by latency of attempted head movements. (**G**) Fraction of trials with ipsiversive attempted head displacements at different timepoints relative to saccade onset for spontaneous (blue) and evoked (red) saccades. (**H**). Head-eye amplitude coupling of spontaneous (blue) and ear airpuff-evoked saccades (red). Each dot corresponds to a single gaze shift. Attempted head amplitude was measured 150ms after saccade onset. Spontaneous: $R^2 = 0.58$, slope = 0.214, $p < 10^{-10}$. Evoked: $R^2 = 0.64$, slope = 0.162, $p < 10^{-10}$. Spontaneous and evoked regression slopes were significantly different ($p < 10^{-5}$, permutation test). Histograms above and beside scatter plot indicate distributions of saccade and attempted head movement amplitudes. Difference in means significant ($p < 10^{-5}$ for saccades, $p < 10^{-5}$ for head, permutation test). (**I–J**) As in (**E–F**), but for attempted head movement velocity. (**K**) As in (**G**), but for attempted head movement velocity. (**L**) As in (**H**), but for attempted head movement velocity. Peak attempted head velocity was measured 60ms after saccade onset. Spontaneous: $R^2 = 0.40$, slope = 2.04, $p < 10^{-10}$. Evoked: $R^2 = 0.52$, slope = 1.98, $p < 10^{-10}$. Spontaneous and evoked regression slopes were not significantly different ($p = 0.08$, permutation test). Histograms above and beside scatter plot indicate distributions of saccade amplitudes and peak attempted head velocities. Difference in means was significant ($p < 10^{-5}$ for saccades, $p < 10^{-5}$ for head, permutation test).

The online version of this article includes the following figure supplement(s) for figure 4:

**Figure supplement 1.** Head-eye coupling for different stimuli.

**Figure supplement 2.** Different head-eye coupling during spontaneous and touch-evoked gaze shifts is not due to differences in saccade start and end points.

**Figure supplement 3.** Head-eye coupling across and within sessions.

**Figure supplement 4.** Ear and whisker airpuff trials matched for attempted head movement amplitude have different saccade endpoints.

**Figure supplement 5.** Gain of head-eye coupling variability across mice.

## Directionally biased saccades reflect different stimulus-dependent relationships between initial eye position and saccade direction and amplitude

We next sought to understand how different stimuli evoke saccades with distinct endpoints. Given the prevailing view that head movements drive saccades during mouse gaze shifts, one possibility was that directionally biased saccade endpoints are the result of larger or more directionally biased attempted head movements. Indeed, the distributions of attempted head displacements seemed similar to those of saccade endpoints, suggesting that this was possible, although there was not a clear relationship between average attempted head movement amplitude and saccade endpoint eccentricity (*Figure 3A-D, I-L, Figure 4—figure supplement 1A-D*). Therefore, to directly test whether different distributions of attempted head movement direction and amplitude across stimuli could explain the distinct saccade endpoints (e.g. with stimuli that evoke larger head movements causing saccades that overshoot to more directionally biased endpoints) we compared trials across stimuli with matched attempted head movement direction and amplitude (*Figure 3A, B, I, J, Figure 4—figure supplement 4*). If differences in attempted head movements underlie directionally biased saccade endpoints, then controlling for attempted head movement amplitude in this manner should cause whisker and ear airpuff-evoked saccade endpoints to be similar. Strikingly, however, endpoints for attempted head movement-matched whisker and ear airpuff trials remained well-separated, indicating that the more directionally biased endpoints of ear airpuff-evoked saccades are not simply due to larger or more directionally biased attempted head movements (*Figure 4—figure supplement 4*).

Because saccade endpoints are a function of both saccade amplitude and initial eye position, we next examined whether stimulus-dependent differences in the relationship between initial eye position and saccade direction and amplitude could contribute to resulting endpoint differences. Indeed, for all stimuli there was an inverse relationship between initial eye position and saccade amplitude but the slopes and intercepts of the lines of best fit describing these relationships differed depending on stimulus modality and location (*Figure 5A–D, Figure 5—figure supplement 1*). For example, stimuli with central saccade endpoints had lines of best fit passing through the origin, whereas stimuli with more leftward or rightward endpoints had lines of best fit that were shifted downward or upward, respectively. In other words, saccades of a given amplitude were typically elicited from distinct initial eye positions by different stimuli: for example, 5° leftward saccades were evoked by left ear airpuffs

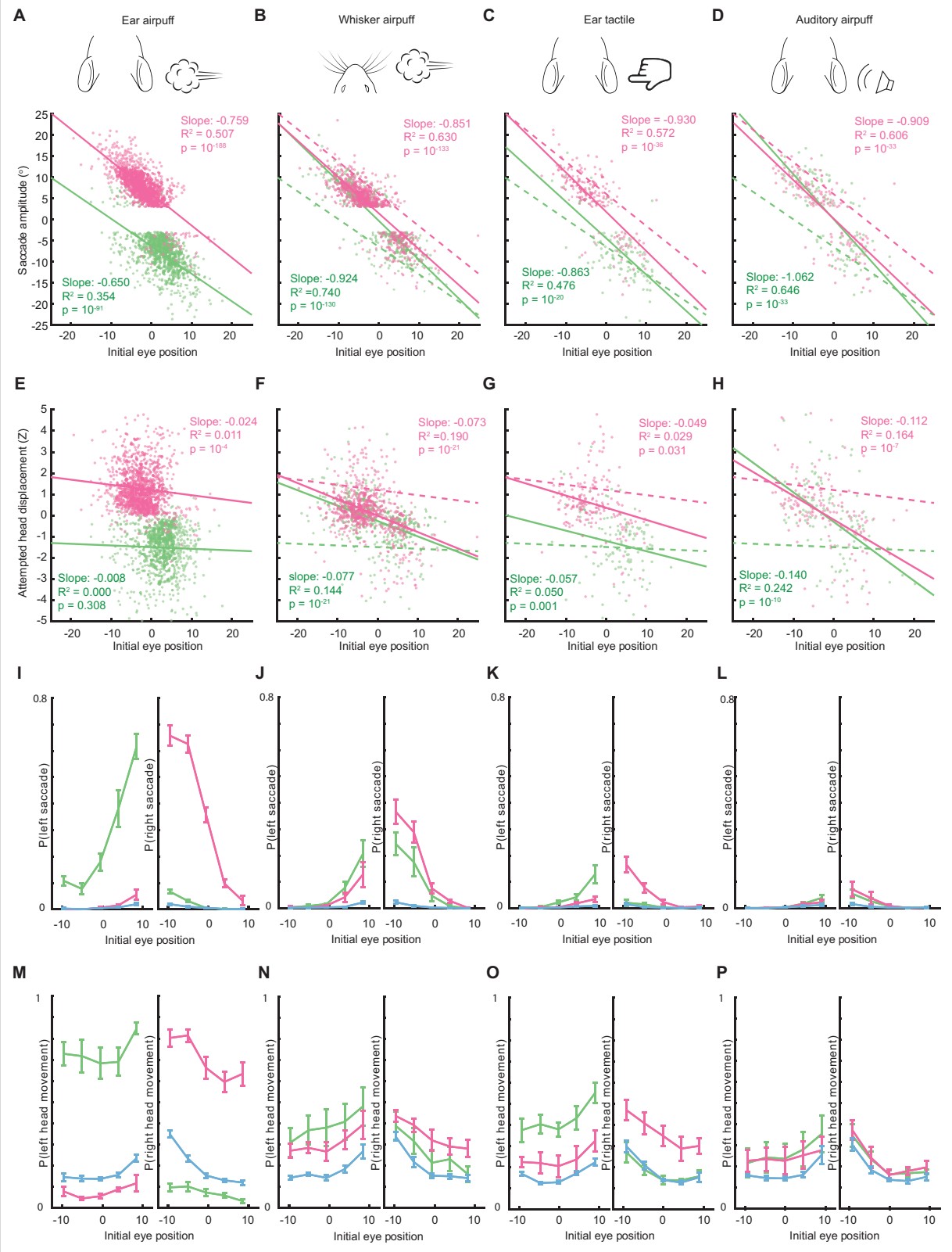

**Figure 5.** Saccade and head movement direction, amplitude, and probability depend on initial eye position. (**A–D**) Relationship between saccade amplitude and eye position for ear airpuffs (**A**), whisker airpuffs (**B**), ear tactile (**C**), and auditory airpuffs (**D**). Dotted lines in B-D are lines of best fit from A for comparison. (**E–H**) Relationship between attempted head displacement and eye position for ear airpuffs (**E**), whisker airpuffs (**F**), ear tactile (**G**), and auditory airpuffs (**H**). Dotted lines in F-H are lines of best fit from E. The trials in A-H are the same as those in *Figure 2*. (**I–L**) Relationship between

*Figure 5 continued on next page*

*Figure 5 continued*

initial eye position and saccade probability for left and right saccades. (**M–P**) Relationship between initial eye position and attempted head movement probability for left and right attempted head movements. Green and magenta lines in I-P indicate population means for movements evoked by left and right stimuli, respectively. Blue lines indicate spontaneous saccades or head movements. Error bars indicate s.e.m. Total trial numbers for I-P: ear airpuff sessions, spontaneous = 13,384, left ear airpuff = 3506, right ear airpuff = 3497; whisker airpuff sessions, spontaneous = 14,511, left whisker airpuff = 3926, right whisker airpuff = 4026; tactile ear sessions, spontaneous = 13,529, left tactile ear = 3646, right tactile ear; = 3695; auditory airpuff, spontaneous = 13,404, left auditory airpuff = 6362, right auditory airpuff = 6385.

The online version of this article includes the following figure supplement(s) for figure 5:

**Figure supplement 1.** Saccade amplitude, saccade endpoint, and attempted head displacement binned by initial eye position.

**Figure supplement 2.** Relationship between initial eye position and saccade probability.

**Figure supplement 3.** Effects of sensory history and arousal on saccade generation.

**Figure supplement 4.** Relationship between initial head position and saccade and head movement amplitude.

from central initial eye positions but by left auditory airpuffs from initial eye positions roughly 5° right of center (*Figure 5A and D*, *Figure 5—figure supplement 1*). In this way, otherwise identical 5° saccades have central endpoints in response to an auditory airpuff and leftward endpoints in response to a left ear airpuff. Thus, endpoint differences between stimuli arise at least in part from distinct stimulus-dependent relationships between initial eye position and saccade direction and amplitude.

Having observed that initial eye position and stimulus location jointly shape the direction and amplitude of stimulus-evoked saccades, we next examined whether they had any relationship with saccade probability. Indeed, initial eye position strongly influenced saccade probability, and the relationship between initial eye position and saccade probability differed across stimuli, with the lowest probability coinciding with the mean endpoint of saccades evoked by that stimulus (*Figure 5I–L*, *Figure 5—figure supplement 2*). For all stimuli, leftward eye movements were more likely from initial eye positions to the right of the mean saccade endpoint for that stimulus, whereas rightward eye movements were more likely from initial eye positions to the left of the mean saccade endpoint for that stimulus. In contrast, saccade probability did not differ between trials with high (dilated pupils) or low (constricted pupils) arousal (*Figure 5—figure supplement 3*; *Reimer et al., 2014*) and declined only slightly over successive sessions or within sessions.

Finally, because the preceding analyses found that saccade direction and amplitude depend on initial eye position and stimulus location (*Figure 5A-D,I-L*), and because head and eye movement direction and amplitude are highly correlated, we asked whether initial eye position and stimulus location shape the direction and amplitude of evoked head movements. We first analyzed whisker airpuff-evoked gaze shifts because these involved a mixture of saccade directions. Strikingly, attempted head movement direction and amplitude were dependent on initial eye position (*Figure 5F*, *Figure 5—figure supplement 1*). We then analyzed auditory airpuff-evoked gaze shifts, which also involved a mixture of saccade directions. As for whisker airpuffs, both attempted head movement direction and amplitude were dependent on initial eye position (*Figure 5D*, *Figure 5—figure supplement 1*). We next examined ear tactile stimuli, which elicited mostly contraversive movements but also some ipsiversive movements. Indeed, as for both whisker and auditory airpuffs, attempted head movement direction and amplitude depended on eye position, but the slope of this relationship was shallower (*Figure 5G*, *Figure 5—figure supplement 1*). We then examined ear airpuffs. These stimuli elicited nearly exclusively contraversive movements, such that no effect of eye position on direction could be observed, and effects on amplitude were subtle, with shallower slopes, and significant for stimuli only on one side (*Figure 5E*, *Figure 5—figure supplement 1*). However, eye position strongly influenced the probability of attempted head movements evoked by all stimuli, including ear airpuffs (*Figure 5M–P*, *Figure 5—figure supplement 1*). As a control, we examined the relationship between the initial position of the head, as measured by the strain gauge, and saccade and attempted head movement direction and amplitude. For all stimuli, initial eye position was a much stronger predictor of saccade direction and amplitude than was initial position of the head (*Figure 5—figure supplement 4*). Likewise, for each of the stimuli for which attempted head movement direction and amplitude were well predicted by initial eye position—whisker airpuffs, ear tactile stimuli, and auditory airpuffs—initial head position was a weak predictor of attempted head movements (*Figure 5—figure*

*supplement 4*). Taken together, these data indicate that starting eye position and stimulus location jointly shape attempted head and eye movement probability, direction, and amplitude.

## The superior colliculus mediates airpuff-evoked gaze shifts

We next sought to identify the neural circuitry underlying airpuff-evoked gaze shifts. As discussed previously, in other species, stimulus-evoked gaze shifts involving directed head and eye movements are driven by SC (*Freedman, 2008*; *Freedman et al., 1996*; *Guitton, 1992*; *Guitton et al., 1980*; *Paré et al., 1994*). In contrast, it is widely believed that the recentering saccades observed in mice are driven by brainstem circuitry in response to head rotation (*Curthoys, 2002*; *Hepp et al., 1993*; *Kitama et al., 1995*; *Meyer et al., 2020*; *Michaiel et al., 2020*; *Payne and Raymond, 2017*). To determine whether SC is required to generate touch-evoked gaze shifts in mice, we pursued an opto-genetic strategy to perturb SC activity in the period surrounding airpuff onset. For inhibition exper-iments, we stereotaxically injected adeno-associated virus (AAV) encoding the light-gated chloride pump eNpHR3.0 under the control of a pan-neuronal promoter and implanted a fiber optic in right SC (*Gradinaru et al., 2010*). Consistent with data in foveate species (*Hikosaka and Wurtz, 1985*; *Robinson, 1972*; *Schiller and Stryker, 1972*), optically reducing right SC activity shifted airpuff-evoked saccade endpoints to the right (i.e. ipsilaterally) for both left (–3.7 ± 4.3° [control] vs. –2.3 ± 4.7° [LED on], p < 0.001, Welch's t-test) and right ear airpuffs (4.5 ± 4.1° [control] vs. 5.4 ± 4.7° [LED on], p = 0.011, Welch's t-test) (*Figure 6A–C*). To control for potential mismatches in starting eye posi-tion between LED-off and LED-on trials, we performed additional analyses using matched trials and found that the endpoint and amplitude differences persisted (*Figure 6—figure supplement 1*). For stimulation experiments, we again stereotaxically injected AAV encoding the light-gated ion channel ChR2 under the control of a pan-neuronal promoter and implanted a fiber optic in right SC (*Gradinaru et al., 2010*). Because strong SC stimulation evokes saccades, we used weak stimulation (50–120 µW) in order to bias SC activity. This manipulation caused the reciprocal effect of right SC inhibition, biasing endpoints leftwards (i.e. contraversively) for both left (–5.8 ± 4.6° [control] vs. –8.0 ± 6.1° [LED on], p = 0.0016, Welch's t-test) and right (5.6 ± 3.5° [control] vs. 1.9 ± 6.1° [LED on], p < 10$^{-5}$, Welch's t-test) airpuffs (*Figure 6E–G*). Once again, controlling for differences in starting eye position between conditions yielded similar results (*Figure 6—figure supplement 1*).

To understand how SC manipulations affect attempted head movements and head-eye coupling, we examined the distribution of head movements as a function of saccade amplitude. As expected given the role of SC in generating both head and eye movements, attempted head movements were shifted to the right (i.e. ipsiversively) by SC inhibition (*Figure 6D*) and to the left (i.e. contraversively) by SC excitation (*Figure 6H*). To examine the effects of SC manipulations on head-eye coupling, we identified trials with identical saccade trajectories in LED-on and LED-off conditions and examined the corresponding head movement amplitudes. Interestingly, SC manipulations had no effect on the relationship between saccade and head movement amplitudes, suggesting that SC manipulations do not change head-eye coupling during gaze shifts (*Figure 6—figure supplement 1*). Taken together, these bidirectional manipulations indicate that SC serves a conserved necessary and sufficient role in generating ear airpuff-evoked gaze shifts.

## Discussion

Here, we investigated whether mouse gaze shifts are more flexible than had previously been appre-ciated. In the prevailing view, mouse gaze shifts are led by head rotations that trigger compensatory eye movements, including saccades that function to reset the eyes (*Land, 2019*; *Land and Nilsson, 2012*; *Liversedge et al., 2011*; *Meyer et al., 2020*; *Michaiel et al., 2020*; *Payne and Raymond, 2017*). These 'recentering' saccades are attributed to head movement-related vestibular and optoki-netic cues (*Curthoys, 2002*; *Meyer et al., 2020*; *Michaiel et al., 2020*; *Payne and Raymond, 2017*). Working in a head-fixed context to eliminate vestibular and optokinetic cues and to present stimuli of different modalities at precise craniotopic locations, we found that mouse gaze shifts are more flexible than previously thought. As discussed below, we identified unexpected flexibility in the endpoints of saccades, saccade timing and amplitude relative to attempted head movements, the eye positions from which gaze shifts are made, and the brain regions that drive them.

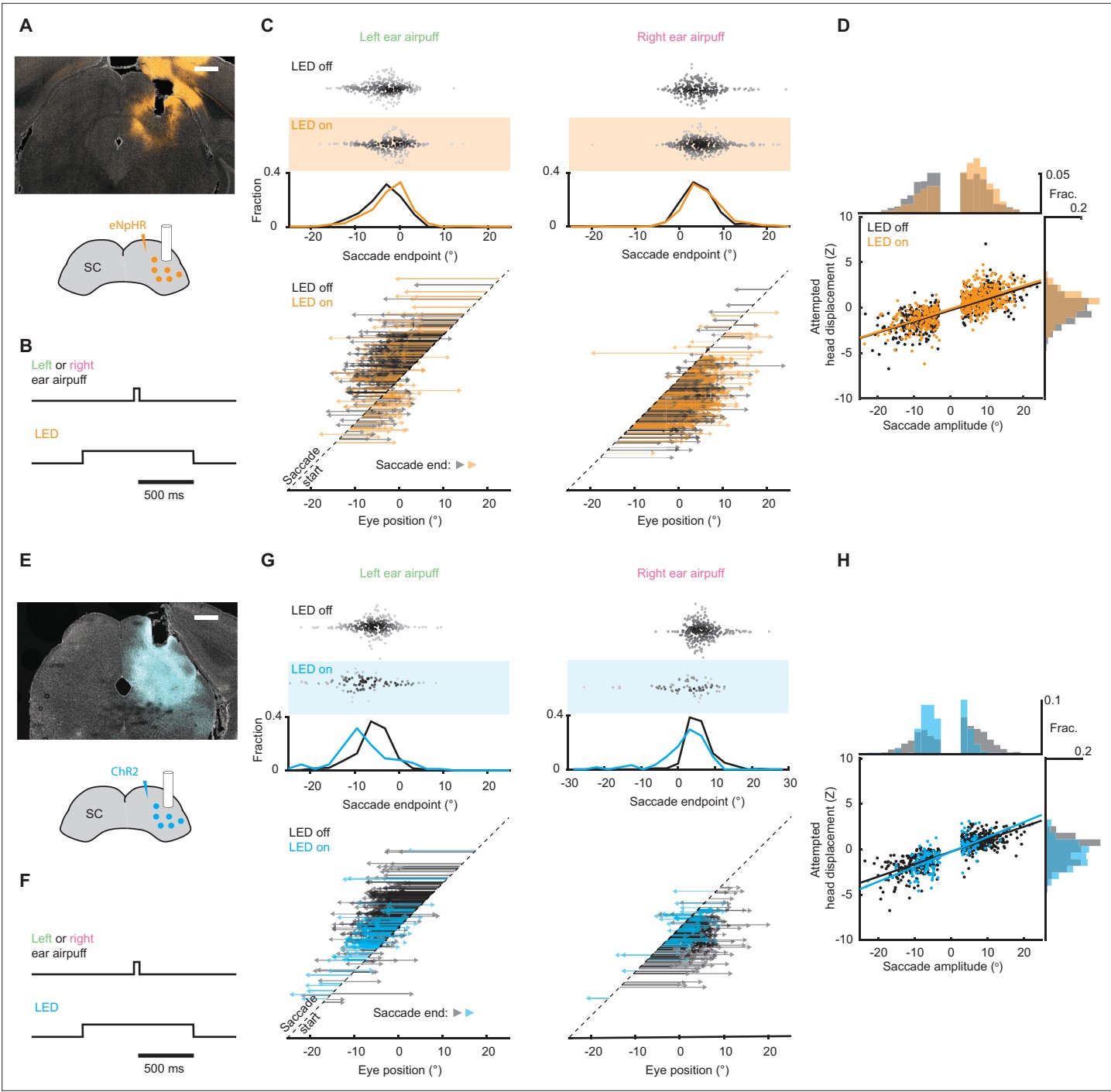

**Figure 6.** Superior colliculus controls touch-evoked gaze shifts. (**A**) Schematic of right SC optogenetic inhibition using eNpHR3.0 and example histology for representative mouse. Scale bar, 0.5 mm. The lack of fluorescence immediately surrounding fiber tip is due to photobleaching by high photostimulation intensity (12 mW, as opposed to 50–120 µW for ChR2 experiments in (**E–H**)). (**B**) Trial structure. Optogenetic illumination is provided for a 1 s period centered around airpuff delivery. (**C**) Effects of SC optogenetic inhibition on saccade endpoints. Top, scatter plots and histograms of endpoints for control (white background, n = 296) and LED on (orange background, n = 235) trials. Middle, endpoint histograms for control (black) and LED on (orange) trials. Bottom, saccade vectors for control (black) and LED on (orange) trials. (**D**) Head-eye amplitude coupling during ear airpuff-evoked gaze shifts for control (black) and LED on (orange) trials. Each dot represents an individual gaze shift. Control: $R^2$ = 0.56, slope = 0.123, p < $10^{-10}$. LED on: $R^2$ = 0.53, slope = 0.127, p < $10^{-10}$. Control and LED on regression slopes were significantly different (p = 0.01, permutation test) due to differences in eye positions from which gaze shifts were generated, because controlling for initial eye position eliminated this difference (***Figure 6— figure supplement 1***). Histograms above and beside scatter plot show distributions of saccade amplitudes and attempted head displacements, respectively. Distribution means were significantly different (p < $10^{-5}$, permutation test). (**E**) Schematic of right SC optogenetic subthreshold stimulation

*Figure 6 continued on next page*

*Figure 6 continued*

using ChR2 and example histology for representative mouse. Scale bar, 0.5 mm. (**F**) Trial structure. Optogenetic illumination is provided for a 1 s period centered around airpuff delivery. (**G**) Effects of weak SC optogenetic stimulation on saccade endpoints. Top, scatter plots and histograms of endpoints for control (white background, n = 547) and LED on (blue background, n = 157) trials. We observed fewer trials in the LED-on condition because SC stimulation increased the probability of spontaneous saccades prior to stimulus onset, and trials with saccades in the 500ms before stimulus delivery were excluded from analysis. Middle, histograms of endpoints for control (black) and LED on (blue) trials. Bottom saccade vectors for control (black) and LED on (blue) trials. (**H**) Head-eye amplitude coupling during ear airpuff-evoked gaze shifts for control (black) and LED on (blue) trials. Each dot represents an individual gaze shift. Attempted head amplitude was measured 150ms after saccade onset. Control: $R^2 = 0.69$, slope = 0.137, $p < 10^{-10}$. LED on: $R^2 = 0.52$, slope = 0.164, $p < 10^{-10}$. Control and LED-on regression slopes were significantly different ($p < 10^{-5}$, permutation test) due to difference in eye positions from which gaze shifts were generated, because controlling for initial eye position eliminated this difference (*Figure 6— figure supplement 1*). Histograms above and beside scatter plot show distributions of saccade amplitudes and attempted head displacements, respectively. Distribution means were significantly different ($p < 10^{-5}$, permutation test).

The online version of this article includes the following figure supplement(s) for figure 6:

**Figure supplement 1.** Controlling for the effects of initial eye position on superior colliculus manipulations.

## Touch-evoked saccades are directionally biased

The first indication that mouse gaze shifts are more flexible than previously appreciated was that an analysis of endpoints revealed that touch-evoked saccades are directionally biased rather than recentering. This conclusion is based on three lines of evidence. First, endpoints of saccades evoked by left and right ear airpuffs are near the left and right edges, respectively, of the range of eye positions observed and overlap minimally, despite trial-to-trial variability. In contrast, endpoints for saccades evoked by left and right auditory stimuli are indistinguishable and centrally located. Second, left and right ear airpuffs evoke saccades traveling in opposite directions from most eye positions; by definition, one of these directions must lead away from center and is thus *centrifugal* rather than *centripetal*. In contrast, saccades evoked by both left and right auditory stimuli travel centripetally from all initial eye positions. Third, from many eye positions, touch-evoked saccades that begin toward the center pass through to reach endpoints at eccentricities between 5 and 10 degrees and cannot accurately be termed centripetal. For these reasons, we conclude that touch-evoked saccades are directionally biased and do not serve to recenter the eyes.

Our findings contrast with and complement previous studies contending that rodents, like other afoveates, use saccades to reset their eyes to more central locations (*Meyer et al., 2018*; *Meyer et al., 2020*; *Michaiel et al., 2020*; *Wallace et al., 2013*). One recent analysis suggested that gaze shifts made during visually guided prey capture involve resetting centripetal saccades that 'catch up' with the head (*Michaiel et al., 2020*). Another found that saccades away from the nose recenter the eye, whereas saccades toward the nose move the eye slightly beyond center (*Meyer et al., 2020*). Although we observed this as well, it does not contribute to our results because we averaged the positions of the left and right eyes, eliminating this asymmetry. Earlier studies in head-fixed mice observed occasional, undirected saccades in response to changes in the visual environment (*Samonds et al., 2018*) and found that mice could be trained to produce visually guided saccades only after weeks of training and at extremely long (~1 s) latencies (*Itokazu et al., 2018*). To our knowledge, ours is the first study demonstrating innate gaze shifts involving directionally biased saccades in mice (or any species lacking a fovea). An important caveat is that the present studies were conducted in head-fixed animals, and mice may behave differently in a freely moving context due to head movement-related feedback mechanisms.

## Touch-evoked saccades do not follow head movements

The prevailing view holds that head movements initiate and determine the amplitude of mouse gaze shifts, with eye movements compensatory by-products. In support of this model, one study found that spontaneous saccades in head-fixed mice are preceded by attempted head rotations. A careful comparison with gaze shifts occurring during visually guided object tracking and social interactions in freely moving mice led the authors to suggest that head-eye coupling is not disrupted during head-fixation, and that gaze shifts in both contexts are head-initiated (*Meyer et al., 2020*). Another study tracked the eyes and head during visually guided cricket hunting and found that gaze shifts are driven by the head, with the eyes following to stabilize and recenter gaze (*Michaiel et al., 2020*). Together, these findings have bolstered the prevailing view that afoveates

such as mice generate gaze shifts driven by head movements, with eye movements compensatory by-products (*Land, 2019*; *Land and Nilsson, 2012*; *Liversedge et al., 2011*). Consistent with published findings in both freely moving and head-fixed mice, we found that spontaneous saccades are preceded by slow attempted head movements in the direction of the ensuing saccade, typically beginning 100–200 ms before saccade onset (*Meyer et al., 2020*; *Michaiel et al., 2020*). In contrast, we found that touch-evoked gaze shifts are not preceded by head movements, suggesting that mouse saccades are not always made in response to head movements. Additional support for this idea came from an analysis of head and eye movements as a function of eye position. If gaze shifts were determined solely by the location of the stimulus relative to the head and saccades were a compensatory by-product of this calculation, eye position should have no effect on head movements. However, we found that the amplitudes, directions, and/or probabilities of stimulus-evoked saccades and attempted head movements vary with initial eye position. The influence of eye position on evoked eye *and* attempted head movements further suggests that saccades are not simply compensatory by-products of head movements. Instead, we contend, touch-evoked head movements and saccades are likely to be specified simultaneously as parts of a coordinated movement whose component movements take into account both stimulus location and initial eye position. Nevertheless, it is possible that the observed relationship between evoked saccades and attempted head movements is due to the lack of head movement-related feedback in head-fixed mice.

The relative timing of head and eye movements during touch-evoked gaze shifts in head-fixed mice resembles that observed during gaze shifts in cats and primates. For example, head-fixed cats and primates generate gaze shifts using directed saccades and then maintain their eyes in the new orbital position, similar to what we have observed in mice (*Freedman, 2008*; *Guitton et al., 1980*). In addition, saccades in head-fixed cats and primates are often accompanied by attempted head rotations, similar to those we observe during touch-evoked gaze shifts in head-fixed mice (*Bizzi et al., 1971*; *Guitton et al., 1984*; *Paré et al., 1994*). In primates and cats able to move their heads, gaze shifts are usually led by directed saccades (with some exceptions), likely because the eyes have lower rotational inertia and can move faster (*Pelisson and Guillaume, 2009*; *Ruhland et al., 2013*). These saccades tend to be followed by a head movement in the same direction that creates vestibular signals that drive slow, centripetal counterrotation of the eyes to maintain fixation (*Bizzi et al., 1972*; *Freedman, 2008*; *Freedman and Sparks, 1997*; *Guitton et al., 1984*). In this way, the animal can rapidly shift its gaze with a directed saccade yet subsequently reset the eyes to a more central position. It is tempting to speculate that a similar coordinated sequence of head and eye movements occurs during touch-evoked gaze shifts in freely moving mice, enabling mice to rapidly shift gaze with their eyes while eventually resetting the eyes in a more central orbital position.

## Head-eye amplitude coupling differs during spontaneous and evoked saccades

An additional feature that distinguishes spontaneous and touch-evoked gaze shifts is the relative contributions of head and eye movements. We found that spontaneous saccades of a given amplitude are coupled to larger head movements than are touch-evoked saccades. This difference arises largely from the absence of a pre-saccadic slow attempted head movement during touch-evoked gaze shifts, as the fast phases are similar. This differential pairing of head and eye movements is reminiscent of reports in primates and cats that the relative contributions of head and eye movements vary for gaze shifts evoked by different sensory modalities (*Goldring et al., 1996*; *Populin, 2006*; *Populin and Rajala, 2011*; *Populin et al., 2004*; *Ruhland et al., 2013*; *Tollin et al., 2005*). However, in those species, vision typically elicits gaze shifts dominated by saccades while hearing typically evokes gaze shifts entailing larger contributions from head movements. In contrast, we observed that sound- and touch-evoked gaze shifts involve larger contributions from saccades than do spontaneous gaze shifts, whereas visual stimuli did not evoke gaze shifts at all. This indicates that although there is general conservation of the involvement of SC in sensory-driven gaze shifts, modality-specific features are not conserved, which may reflect differences in sensory processing across species. We also observed limited variability across mice in the relative contributions of head and eye movements to gaze shifts, which contrasts with the observation that different human subjects are head 'movers' and 'non-movers' during gaze shifts (*Figure 4—figure supplement 5*) Thus, our results reveal that

head-fixed mice are capable of using multiple strategies to shift their gaze but with key differences from other species.

## Directionally biased saccade endpoints reflect different stimulus-dependent relationships between initial eye position and saccade direction and amplitude

Given the prevailing view that head movements drive saccades during mouse gaze shifts, we predicted that directionally biased saccades were the result of different head movements. Indeed, distributions of attempted head displacements and saccade endpoints appeared similar for the stimuli tested. However, an analysis of trials matched for head movement direction and amplitude across stimuli yielded well-separated endpoints, suggesting that more eccentric saccade endpoints and the associated distributions of attempted head displacements are not simply due to differences in the directions and amplitudes of evoked head movements. Instead, we found that saccade endpoint differences across stimuli were associated with distinct stimulus-dependent relationships between initial eye position and saccade amplitude and direction. For example, 5° leftward saccades were evoked by left ear airpuffs from central initial eye positions but by left auditory airpuffs from initial eye positions roughly 5° right of center. In this way, saccades with identical amplitudes, coupled to identical attempted head movements, yielded central endpoints in response to an auditory airpuff and eccentric endpoints in response to a left ear airpuff. In the future, it will be essential to determine the neural mechanisms that instantiate these strategies and to examine these strategies in freely moving mice.

## The role of superior colliculus in sensory-evoked mouse gaze shifts

In other species, SC drives sensory-evoked gaze shifts, and microstimulation and optogenetic stimulation of mouse SC has been shown to elicit gaze shifts (*Masullo et al., 2019*; *Wang et al., 2015*). However, to our knowledge, no study had identified a causal involvement of SC in mouse gaze shifts. We found that optogenetic stimulation of SC elicits directionally biased saccades that coincide with attempted head movements and resembled those elicited by touch. We therefore performed bidirectional optogenetic manipulations that revealed that touch-evoked gaze shifts depend on SC, identifying a conserved, necessary and sufficient role for SC in directed gaze shifts. In addition, we found that SC manipulations did not alter head-eye amplitude coupling. This observation suggests that SC specifies the overall gaze shift amplitude rather than the individual eye or head movement components, consistent with observations in other species (*Freedman et al., 1996*; *Paré et al., 1994*).

## Ethological significance

Prior to the present study, it was believed that species with high-acuity retinal specializations acquired the ability to make directed saccades to scrutinize salient environmental stimuli, because animals lacking such retinal specializations were thought incapable of gaze shifts led by directed saccades (*Land, 2019*; *Land and Nilsson, 2012*; *Liversedge et al., 2011*; *Walls, 1962*). Our discovery that sensory-guided directionally biased saccades are present in head-fixed mice—albeit withless precise targeting of stimulus location than is seen in foveate species—raises the question of what fovea-independent functions these movements ordinarily serve. Although mice have lateral eyes and a large field of view, saccades that shift gaze in the direction of a stimulus, as seems to occur with both ear and whisker tactile stimuli, may facilitate keeping salient stimuli within the field of view. As natural stimuli are often multimodal, directing non-visual stimuli toward the center of view maximizes the likelihood of detecting the visual component of the stimulus. Alternatively, despite mouse retinae lacking discrete, anatomically defined specializations such as foveae or areas centralis, there are subtler nonuniformities in the distribution and density of photoreceptors and retinal ganglion cell subtypes, and magnification factor, receptive field sizes, and response tuning vary across the visual field in higher visual centers; it may be desirable to center a salient tactile stimulus on a particular retinal region (*Ahmadlou and Heimel, 2015*; *Baden et al., 2013*; *van Beest et al., 2021*; *Bleckert et al., 2014*; *Dräger and Hubel, 1976*; *Feinberg and Meister, 2015*; *Li et al., 2020*; *de Malmazet et al., 2018*). Although touch-evoked saccades alone may be too small to center the stimulus location on any particular region of the retina, they may do so in concert with directed head movements. Experiments in freely moving mice will be essential to understanding the behavioral functions of these saccades.

Why tactile stimuli evoked directionally biased saccades in our preparation whereas auditory and visual stimuli did not is unclear. One possibility is the aforementioned speed of saccades relative to head movements may be especially beneficial because tactile stimuli typically derive from proximal objects and as a result may demand rapid responses. Alternatively, the high spatial acuity of the tactile system may enable more precise localization (*Allen and Ison, 2010*; *Diamond et al., 2008*). Auditory stimuli, in contrast, may alert animals to the presence of a salient stimulus in their environments whose location is less precisely ascertained, and as a result drive gaze shifts whose goal is to reset the eyes to a central position that maximizes their chances of sensing and responding appropriately. Finally, our set of stimuli was not exhaustive, and it is possible that as yet unidentified visual or auditory stimuli could elicit gaze shifts with directionally biased saccades.

## Future directions

In this study, we used a head-fixed preparation to eliminate the confound of head movement-related sensory cues and to present stimuli from defined locations. However, in the future, it will be critical to compare touch-evoked gaze shifts in head-fixed and freely moving animals. For example, as noted previously, whereas head-fixed primates and cats generate gaze shifts using directed saccades and then maintain their eyes in the new orbital position, similar to what we have observed, in freely moving primates and cats, gaze shifts are led by directed saccades but typically followed by head movements during which the eyes counterrotate centripetally in order to maintain gaze in the new direction. It will be interesting to know whether similar differences distinguish saccade-led touch-evoked gaze shifts in head-fixed and freely moving mice. By expanding on methods similar to those recently described by Meyer et al. and Michaiel et al., it may be possible to investigate these and other questions (*Meyer et al., 2018*; *Meyer et al., 2020*; *Michaiel et al., 2020*).

Furthermore, a practical implication of our identification of mouse SC-dependent gaze shifts is that this behavioral paradigm could be applied to the study of several outstanding questions. First, there are many unresolved problems regarding the circuitry and ensemble dynamics underlying target selection (*Basso and May, 2017*) and saccade generation (*Gandhi and Katnani, 2011*), and the mouse provides a genetically tractable platform with which to investigate these and other topics. Second, gaze shifts are aberrant in a host of conditions, such as Parkinson's and autism spectrum disorder (*Liversedge et al., 2011*). This paradigm could be a powerful tool for the study of mouse models of a variety of neuropsychiatric conditions. Third, directing saccades toward particular orbital positions during these gaze shifts requires an ability to account for the initial positions of the eyes relative to the target, a phenomenon also known as remapping from sensory to motor reference frames. Neural correlates of this process have been observed in primates (*Groh and Sparks, 1996*; *Jay and Sparks, 1984*) and cats (*Populin et al., 2004*), but the underlying circuitry and computations remain obscure. This behavior may facilitate future studies of this problem. Fourth, the different types of gaze shifts that rely on distinct head-eye coupling we have identified may be useful for understanding mechanisms that control movement coordination. Thus, touch-evoked saccade behavior is likely to be a powerful tool for myriad lines of investigation.

## Conclusions

We have found that mouse gaze shifts are unexpectedly flexible, with mice able to make both spontaneous gaze shifts led by the head and stimulus-evoked gaze shifts involving directionally biased saccades coincident with head movements. Prior studies in species whose retinae lack high-acuity specializations had never observed gaze shifts with these properties, but our study used a broader range of stimuli than previously tested and a head-fixed preparation that allowed spatially precise delivery. Detailed perturbation experiments determined that the circuit mechanisms of sensory-evoked gaze shifts are conserved from mice to primates, suggesting that this behavior may have arisen in a common, afoveate ancestral species long ago. More broadly, our findings suggest that analyzing eye movements of other afoveate species thought not to make directed saccades—such as rabbits, toads, and goldfish—in response to a diverse range of multimodal stimuli may uncover similar flexibility.

## Materials and methods

### Mice

All experiments were performed according to Institutional Animal Care and Use Committee standard procedures. C57BL/6J wild-type (Jackson Laboratory, stock 000664) mice between 2 and 6 months of age were used. Mice were housed in a vivarium with a reversed 12:12 h light:dark cycle and tested during the dark phase. No statistical methods were used to predetermine sample size. Behavioral experiments were not performed blinded as the experimental setup and analyses are automated.

### Surgical procedures

Mice were administered carprofen (5 mg/kg) 30 min prior to surgery. Anesthesia was induced with inhalation of 2.5% isoflurane and buprenorphine (1.5 mg/kg) was administered at the onset of the procedure. Isoflurane (0.5-2.5% in oxygen, 1 L/min) was used to maintain anesthesia and adjusted based on the mouse's breath and reflexes. For all surgical procedures, the skin was removed from the top of the head and a custom titanium headplate was cemented to the leveled skull (Metabond, Parkell) and further secured with dental cement (Ortho-Jet powder, Lang Dental). Craniotomies were made using a 0.5 mm burr and viral vectors were delivered using pulled glass pipettes coupled to a microsyringe pump (Micro4, World Precision Instruments) on a stereotaxic frame (Model 940, Kopf Instruments). Following surgery, mice were allowed to recover in their home cages for at least 1 week.

### Viral injections and implants

Coordinates for SC injections were ML: 1.1 mm, AP: 0.6 mm (relative to lambda), DV: -1.9 and -2.1 mm (100 nL/depth). Coordinates for SC implants were ML: 1.1 mm, AP: 0.6 mm (relative to lambda), DV: -2.0 mm. Fiber optic cannulae were constructed from ceramic ferrules (CFLC440-10, Thorlabs) and optical fiber (400 mm core, 0.39 NA, FT400UMT) using low-autofluorescence epoxy (F112, Eccobond).

### Behavioral procedures

To characterize stimulus-evoked gaze shifts (*Figures 2–5* and related supplements), data were collected from 5 mice over 53 days (maximum of 1 session/mouse/day). Session types were randomly interleaved to yield a total of 6 ear airpuff sessions, 6 ear tactile sessions, 6 whisker airpuff sessions, 10 auditory airpuff sessions, and 5 visual sessions. During experiments, headplated mice were secured in a custom 3D-printed mouse holder. Timing and synchronization of the behavior were controlled by a microcontroller (Arduino MEGA 2560 Rev3, Arduino) receiving serial commands from custom Matlab scripts. All behavioral and data acquisition timing information was recorded by a NI DAQ (USB-6001) for post hoc alignment. All experiments were performed using awake mice. Left and right stimuli were randomly selected and presented at intervals drawn from a 7-12 s uniform distribution. Each session consisted of 350 stimulus presentations and lasted ~55 min. No training or habituation was necessary.

### Stimuli

Airpuff stimuli were generated using custom 3D-printed airpuff nozzles (1.5 mm wide, 10 mm long) connected to compressed air that was gated by a solenoid. 3D-printed nozzles were used to standardize stimulus alignment across experimental setups but similar results were obtained in preliminary experiments using a diverse array of nozzle designs. For whisker airpuffs, the nozzles were spaced 24 mm apart and centered 10 mm beneath the mouse's left and right whiskers. For ear airpuffs, the nozzles were directed toward the ears while maintaining 10 mm of separation between the nozzles and the mouse. For auditory-only airpuffs, the nozzles were directed away from the mouse while maintaining the same azimuthal position as the ear airpuffs. Whisker, ear, and auditory airpuffs produced a 65dB noise measured at the mouse's head. For tactile-only stimulation, the ears were deflected using a thin metal bar coated in epoxy to soften its edges (7122A37, McMaster). A stepper motor (Trinamic, QSH2818-32-07-006 and TMC2208) was programmed to sweep the bar downward against the ear before sweeping back up. The stepper motor was sandwiched between rubber pads (8514K61, McMaster) and elevated on rubber pedestals (20125K73, McMaster) to reduce any sound or tactile stimulation due to vibration. For visual stimulation, white LEDs (COM-00531, Sparkfun) were mounted 6 inches from the mouse at the same azimuthal position as the airpuff nozzles.

## Eye tracking

The movements of both left and right eyes was monitored at 100 Hz using two high-speed cameras (BFS-U3-28S5M-C, Flir) coupled to a 110 mm working distance 0.5X telecentric lens (#67- 303, Edmund Optics). A bandpass filter (FB850-40, Thorlabs) was attached to the lens to block visible illumination. Three IR LEDs (475-1200-ND, DigiKey) were used to illuminate the eye and one was aligned to the camera's vertical axis to generate a corneal reflection. Videos were processed post hoc using Deep-LabCut, a machine learning package for tracking pose with user-defined body parts (*Mathis et al., 2018*). Data in this paper were analyzed using a network trained on 1000 frames of recorded behavior from 8 mice (125 frames per mouse). The network was trained to detect the left and right edges of the pupil and the left and right edges of the corneal reflection. Frames with a DeepLabCut-calculated likelihood of p < 0.90 were discarded from analyses. Angular eye position (E) was determined using a previously described method developed for C57BL/6J mice (Sakatani and Isa, 2004).

## Attempted head rotation tracking

Attempted head rotations were measured using a 3D-printed custom headplate holder coupled to a load cell force sensor (Sparkfun, SEN-14727). Load cell measurements (sampling frequency 80 Hz) were converted to analog signals and recorded using a NI DAQ (sampling frequency 2000 Hz). The data were then low-pass filtered at 80 Hz using a zero-phase second-order Butterworth filter and then upsampled to match the pupil sampling rate.

## Optogenetics

Optogenetic experiments were performed using the ear airpuff nozzles. Fiber optic cables were coupled to implanted fibers and the junction was shielded with black heat shrink. A 470 nm fiber-coupled LED (M470F3, Thorlabs) was used to excite ChR2-expressing neurons, and a 545 nm fiber-coupled LED (UHP-T-SR, Prizmatix) was used to inhibit eNpHR3.0-expressing neurons. Optogenetic excitation was delivered on a random 50% of trials using 1 s of illumination centered around airpuff onset.

## SC inhibition

For optogenetic inactivation of SC neurons, AAV1.hSyn.eNpHR3.0 was injected into the right SC of 5 wild-type mice (0.6 AP, 1.1 ML, -2.1 and -1.9 DV; 100 nl per depth). Experiments were performed 35-40 days post injection. LED power was 12 mW. Mice underwent five sessions each.

## SC stimulation

To examine optogenetically-evoked gaze shifts in *Figure 1*, AAV1.CaMKIIa.hChR2(H134R)-EYFP was injected into the right SC of four wild-type mice. Experiments were performed 60-65 days post injection. For each mouse, SC was stimulated (1mW) for 40-ms every 7-12s for 350 trials.

For subthreshold optogenetic stimulation of SC neurons, AAV1.CaMKIIa.hChR2(H134R)-EYFP was injected into the right SC of four wild-type mice. Experiments were performed 67-71 days post injection. LED power was individually set to an intensity that did not consistently evoke saccades upon LED onset (50-120uW). Mice underwent five sessions each.

## Histology

For histological confirmation of fiber placement and injection site, mice were perfused with PBS followed by 4% PFA. Brains were removed and post-fixed overnight in 4% PFA, and stored in 20% sucrose solution for at least 1 day. Brains were sectioned at 50 µm thickness using a cryostat (NX70, Cryostar), every third section was mounted, and slides were cover-slipped using DAPI mounting medium (Southern Biotech). Tile scans were acquired using a confocal microscope (LSM700, Zeiss) coupled to a 10X air objective.

## Behavioral analysis

Eye position analyses were performed using the averaged left and right pupil positions, and the mean eye position was subtracted from each session prior to combining data across sessions and mice. Similarly, because mice generate different ranges of raw strain gauge measurements when making attempted head rotations, attempted head rotation data was Z-scored prior to combining data across

sessions and mice. Saccades were defined as eye movements that exceeded 100°/s, were at least 3° in amplitude, and were not preceded by a saccade in the previous 100 ms. The initial positions and endpoints of saccades were defined as the first points at which saccade velocity rose above 30°/s and fell below 20°/s, respectively. Analyses focused on horizontal saccades because saccades were strongly confined to the azimuthal axis (*Figure 2—figure supplement 1*). For all subsequent analyses, saccades were defined as being evoked by a sensory or optogenetic stimulus if they occurred within a 100 ms response window following stimulus delivery, selected due to the sharp increase in saccade probability during this period (*Figure 2—figure supplement 2*). Spontaneous saccades were a catch-all category defined as any saccades made outside of an experimental stimulus (i.e. no stimulus in the 500 ms periods preceding or following the saccade).

To examine stimulation-evoked gaze shifts (*Figure 1*), only trials in which the head and eyes were fixated in the 500ms period preceding LED-onset and in which the eyes began in a central orbital position (-2° to 0°) were used for analysis. Head movement amplitude during gaze shifts was defined as the head sensor reading 150ms following saccade onset. Head movement onset was defined as the point relative to LED-onset at which head displacement exceeded 5 standard deviations from baseline (~0.05-0.1Z).

To quantify stimulus-evoked saccade probability (*Figure 2*), we calculated the fraction of trials in which a saccade occurred in the 100 ms period following stimulus onset (i.e. the response window). To quantify stimulus-evoked attempted head movement probability, we calculated the fraction of trials in which the head sensor reading exceeded 0.25Z at the point 150 ms following stimulus onset. To determine the baseline head movement probability, we calculated the fraction of trials in which the head sensor reading exceeded 0.25Z between -500 ms and -350 ms relative to stimulus onset.

To examine saccade endpoints, we first identified trials in which mice maintained fixation in the 500 ms preceding saccade onset. We then considered stimulus-evoked saccades those occurring within 100 ms of stimulus onset. To examine the amplitudes of attempted head movements accompanying stimulus-evoked saccades, we used the head sensor reading 150 ms following saccade onset.

Heatmaps of single-trial head movements (*Figure 4* and accompanying supplements) were sorted according to head movement latency. To calculate attempted head movement latencies, we first identified trials in which the mice maintained head fixation from -1 to -0.5 s prior to saccade onset and used this period as the baseline. Latency was defined as the first frame between -0.5 and 0.5 s relative to saccade onset when the attempted head movement amplitude exceeded 5 standard deviations from that trial's baseline (~0.05-0.1Z).

To examine the timing of attempted head movements relative to saccade onset (*Figure 4*), we first baseline subtracted attempted head movement traces 500 ms before saccade onset for each trial. For each time point between -500 ms and 500 ms surrounding saccade onset, we calculated the fraction of trials with instantaneous attempted head movements that were ipsiversive to their accompanying saccades.

To examine head-eye amplitude coupling during spontaneous and stimulus-evoked gaze shifts, we defined attempted head rotation displacement as the load cell value 150 ms following saccade onset (the time point at which average load cell value plateaus during stimulus-evoked gaze shifts (*Figure 4*)). For certain analyses, we identified saccades matched (without replacement) for initial eye position and/or saccade amplitude using Euclidean distance as a metric and a 3° distance cutoff. For a subset of analyses, we used attempted head movement velocity which was measured 60 ms after saccade onset (the time point when average load cell velocity peaks).

To examine the relationship between initial eye position and stimulus-evoked saccade or head movement probability, we identified trials in which mice did not saccade in the 500 ms preceding stimulus onset. Left and right head movements were defined as those less than -0.25Z or greater than 0.25Z, respectively. Qualitatively similar results were obtained using thresholds ranging from 0.1Z to 2Z. To examine the relationship between initial eye position and spontaneous saccade or head movement probability, we identified 1 s long time periods in which no stimuli were delivered and in which mice did not saccade in the first 500 ms. We then determined the probability of a saccade between 500 and 600 ms, and the probability of a head movement using the head sensor value at 650ms.

Tests for statistical significance are described in the text and figure legends. Data were shuffled 10,000 times to generate a null distribution for permutation tests.

## Acknowledgements

We thank M Brainard, J Horton, A Krishnaswamy, M Scanziani, and members of the Feinberg laboratory for helpful discussions and comments on earlier versions of the manuscript. This work was supported by departmental funds and grants from the E M Ziegler Foundation for the Blind, Sandler Foundation, Klingenstein-Simons Fellowship Award in Neuroscience, Brain and Behavior Research Foundation (NARSAD Young Investigator Awards 25,337 and 27320), Whitehall Foundation, Simons Foundation (SFARI 574347), and US National Institutes of Health (DP2 MH119426 and R01 NS109060) to EHF.

## Additional information

### Funding

| Funder | Grant reference number | Author |
| --- | --- | --- |
| National Institute of Mental Health | DP2MH119426 | Evan H Feinberg |
| National Institute of Neurological Disorders and Stroke | R01NS109060 | Evan H Feinberg |
| Simons Foundation Autism Research Initiative | 574347 | Evan H Feinberg |
| Esther A. and Joseph Klingenstein Fund | | Evan H Feinberg |
| E. Matilda Ziegler Foundation for the Blind | | Evan H Feinberg |
| Whitehall Foundation | | Evan H Feinberg |
| Brain and Behavior Research Foundation | 25337 | Evan H Feinberg |
| Brain and Behavior Research Foundation | 27320 | Evan H Feinberg |
| Sandler Foundation | | Evan H Feinberg |

The funders had no role in study design, data collection and interpretation, or the decision to submit the work for publication.

### Author contributions

Sebastian H Zahler, David E Taylor, Conceptualization, Data curation, Formal analysis, Investigation, Methodology, Resources, Software, Validation, Visualization, Writing – original draft, Writing – review and editing; Joey Y Wong, Investigation, Writing – review and editing; Julia M Adams, Investigation; Evan H Feinberg, Conceptualization, Funding acquisition, Project administration, Resources, Supervision, Visualization, Writing – original draft, Writing – review and editing

### Author ORCIDs

Sebastian H Zahler ⓘ https://orcid.org/0000-0003-0089-3593
David E Taylor ⓘ https://orcid.org/0000-0002-0476-0299
Joey Y Wong ⓘ https://orcid.org/0000-0003-3697-8951
Julia M Adams ⓘ https://orcid.org/0000-0003-1402-1040
Evan H Feinberg ⓘ http://orcid.org/0000-0001-7040-0980

### Ethics

This study was performed in strict accordance with the recommendations in the Guide for the Care and Use of Laboratory Animals of the National Institutes of Health. All animal procedures were approved by the University of California San Francisco Institutional Animal Care and Use Committee (IACUC) (protocol number AN176625), and were conducted in agreement with the Association for Assessment and Accreditation of Laboratory Animal Care (AAALAC).

**Decision letter and Author response**
Decision letter https://doi.org/10.7554/eLife.73081.sa1
Author response https://doi.org/10.7554/eLife.73081.sa2

## Additional files

### Supplementary files
Transparent reporting form

### Data availability
Annotated data have been uploaded to a Dryad repository (https://doi.org/10.7272/Q6V69GTV).

The following dataset was generated:

| Author(s) | Year | Dataset title | Dataset URL | Database and Identifier |
|---|---|---|---|---|
| Zahler SH, Taylor DE, Adams J, Feinberg EH, Wong J | 2021 | Superior colliculus drives stimulus-evoked directionally biased saccades and attempted head movements in head-fixed mice | https://doi.org/10.7272/Q6V69GTV | Dryad Digital Repository, 10.7272/Q6V69GTV |

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
