## [Editor Report]

Animals investigate their environments by directing their gaze towards salient stimuli. However, whether non-foveal mammals like mice can make directed saccades independent of head movements in response to sensory stimuli remains unclear. Feinberg et al. systematically investigate how tactile, auditory and visual stimuli drive saccade and head movement patterns. Mice make sensory-guided gaze shifts that depend on superior colliculus and involve coincident, directionally biased saccades and attempted head movements, with flexibility in saccade kinematics relative to attempted head movements.

---

## [Decision Letter]

**Decision letter after peer review:**

[Editors’ note: the authors submitted for reconsideration following the decision after peer review. What follows is the decision letter after the first round of review.]

Thank you for submitting your work entitled "Mice Make Targeted Saccades" for consideration by *eLife*. Your article has been reviewed by 3 peer reviewers, and the evaluation has been overseen by a Reviewing Editor and a Senior Editor. The reviewers have opted to remain anonymous.

Comments to the Authors:

We are sorry to say that, after consultation with the reviewers, we have decided that your work will not be considered further for publication by *eLife*. While the reviewers found your paper provocative and addressing an important question, they found that the experiments did not support and justify the main conclusions.

*Reviewer #1:*

Zahler, Taylor et al. investigate the role of rapid eye movements (saccades) in mice. Recent studies in head-fixed and freely-moving mice did not find evidence for saccades directed at salient visual stimuli, even during goal-directed behaviors relying on vision. Rather, saccades appear to be part of a "saccade and fixate" gaze pattern: during the "fixate" period the eyes are counter-rotating relative to the head to keep the image of the external world on the retinas stable. The "saccade" period seems to "recenter" the eyes in the orbits via saccadic eye movements (by shifting gaze together with the head). These studies support the idea that mice (and potentially most other afoveate animals) do not make targeted saccades.

To challenge this view, Zahler, Taylor et al. developed a paradigm in passive, head-fixed mice in which the visual stimulus used in previous studies was replaced by a combination of tactile and auditory stimuli. In the first experiment, an airpuff to the left or right whiskers was paired with a loud sound coming from the same side as the airpuff. In a small subset of stimulus presentations, mice made saccadic eye movements in response to the stimulus. The authors termed this behavior "whisker-induced saccade-like reflex" (WISLR).

Using the WISLR task, the authors found that saccade endpoints depended on the type of stimulus (e.g., ear airpuff or whisker deflection). Optogenetic deactivation or activation of the superior colliculus (SC), a brain region that plays a key role in controlling head and eye movements, shifted saccade endpoints in the WISLR task. Finally, the authors use simulations to test the idea that WISLR-related saccades keep stimuli within the visual field.

While I think that the authors are addressing an important and timely question, I have some concerns about their conclusions. In particular, most observations seem to be consistent with the "resetting model" that the authors seek to disprove. However, I would like to emphasize that providing evidence either for or against the resetting model would be extremely valuable to the scientific community. The mouse is becoming an increasingly important species in vision research and a major "model" of human disease. Thus it would be important to know which aspects might translate to human research (and which don't).

Major concerns:

1. The paper is quite inconsistent when it comes to the stimuli evoking saccades. The authors state that the main motivation for developing the WISLR task was the absence of targeted saccades towards salient visual stimuli in previous studies. However, the conclusion based on their final simulation (Figure 4F-H) is that the function of saccades in response to tactile/auditory stimuli is to keep visual stimuli in the animal's visual field of view. How do these two things fit together? How are airpuff-evoked saccades related to a visual stimulus leaving the quite large visual field extending to the front and the sides of the animal as assumed in the modeling work? Currently, the relation between the stimuli in Figure 2 that evoked pronounced saccades (e.g., ear airpuff and whisker bar) and the animal's visual field is not clear. In particular, to reconcile the auditory/tactile observations with the simulations it will be important to know whether the tactile/auditory stimuli were within or outside the visual field.

– There is a large body of literature on gaze shifts towards visual, tactile and auditory stimuli in humans and non-human primates. A common observation in these studies is that gaze shifts towards visual targets show a stronger saccade component than gaze shifts towards auditory targets which rely more strongly on head movements. This seems to be somewhat orthogonal to the idea that tactile/auditory but not visual stimuli should result in targeted saccades, in particular if circuits are conserved across species as stated by the authors. It would be important to discuss the motivation/findings in the context of the existing literature which is largely ignored in the current paper.

– The authors show that average saccade trajectories converge on nearly the same endpoint (close to 0 deg) regardless of initial eye position (Figure 1C-E). Would that not support the idea of resetting saccades that move the eyes back to a more central location? There is a statistically significant, yet very small difference in endpoints (< 5 deg) between the left/right stimulus conditions. This seems to be the only evidence for "targeted saccades". However, studies in head-fixed (Sakatani and Isa, 2007; Itokazu et al., 2018) and freely-moving (Meyer et al., 2020) mice have demonstrated that there is a systematic asymmetry (about 5 deg) in the sizes of nasal vs temporal saccades. It would be important to check that this asymmetry cannot explain the difference in saccade endpoints.

– The authors show in Suppl Figure 4 that mice attempted to reorient their heads during saccades and that the magnitude of attempted head movements is predictive of saccade sizes. This indicates that saccades in the WISLR task were part of combined eye/head gaze shifts as reported in freely moving mice (Michaiel et al., 2020; Meyer et al., 2020). Consequently, attempted head movements need to be taken into account when interpreting the data. Indeed, there seems to be a difference in attempted head movements between the two conditions shown in Suppl Figure 4B,C (spontaneous, WISLR). Based on these data, it is not clear if "targeted saccades" were just a by-product of head-free gaze patterns in head-fixed animals (given the low probability of observing a saccade at all).

– The shift in saccade endpoints during optical activation/deactivation of the superior colliculus (SC) is a novel and interesting finding. However, the SC is also involved in generating head movements in non-human primates (e.g., Freedman et al. 1996) and mice (e.g., Wilson et al. (2018)). In particular, Masullo et al. (2019) show that SC stimulation generates both head and eye movements. Again, it would be important to dissociate the contribution of (attempted) head movements to the measured saccades to support the idea that eye movements, and not the associated head movements, determine saccade endpoints.

– The authors use modeling to test the idea that the role of "targeted saccades" is to keep visual targets in the animal's visual field. Why would the U-shaped curve in Figure 4D not be consistent with a resetting saccade model in which the probability of observing a saccade increases with angular distance from the eye's central position? The small differences in trough locations might be related to the asymmetry of nasal vs temporal eye movements (see comment above).

– The probability of observing a saccade following a non-visual stimulus seems to be quite low (Figure 1B and Suppl Figure 5) and saccade sizes are rather small (~5-15 deg, Suppl Figure 2A) compared to the large visual field of the mouse (>250 deg). Even if mice made targeted saccades: their function in the context of natural behavior is not discussed in the paper.

A major concern is that "targeted saccades" are just a by-product of combined eye/head gaze shifts observed in freely moving mice (Michaiel et al., 2020; Meyer et al., 2020). The authors measured attempted head orienting movements in a subset of experiments, and the data strongly indicate that eye/head coupling is preserved in the WISLR task. To test if differences in saccade end points can be explained by head orienting movements, the authors could measure attempted head movements during the different conditions in Figure 2. This could be combined with a regression model to test if saccade end points can be predicted from attempted head movement amplitude.

In the method section it is mentioned that only saccades occurring within a specific response window following stimulus onset were included in the analysis (p 23). First, it might be useful to explain the choise of the 150 ms response window and the impact of response window length on the results. For example, Itokazu et al. (2018) report a much longer response window for visual stimuli. Second, it might be useful to indicate this response window in Suppl Figure 1A. I would expect that based on the criteria mentioned in the methods section the 2nd saccade in Suppl Figure 1A was excluded from the analysis. Is that correct?

Related to this: Figure 1B shows the probability of observing a saccade in different time bins following an airpuff. Maybe I missed this but what was the overall probablity of observing a saccade within the response time window? To get a sense of the reliability of saccade generation, it might be useful to also show cumulative probability (either in the same figure or in a suppl figure). It is hard to guess from the distribution but I would guess that p(saccade in response window) < 0.2.

*Reviewer #2:*

This study makes a bold and provocative claim – mice make targeted eye saccades. This is a significant claim, since although many studies have shown that species without fovea perform saccades, these are generally coupled to head movements, and serve to shift gaze by recentering the eyes following slower compensatory eye movements. Indeed, recent studies have shown that eye movements in mice follow this pattern, and in fact even the eye movements seen in head-fixed mice are associated with attempted head movements. This study claims that head-fixed mice do make targeted eye saccades, when triggered by tactile stimulation to the ears/whiskers. If it is true that mice make saccades similar to foveate animals, and this had just been missed because previous researchers had not used tactile stimuli, this would be an exciting finding not only for mouse vision, but for the vision field at large.

However, there are two major issues with the claim that mice are making targeted eye saccades. First, the saccades do not appear to be targeted directly to a stimulus location – rather, they return to center with a slight bias of a couple degrees for saccades "to" stimuli at disparate locations, and in fact there is great variability in saccade endpoints (larger than the difference between target locations). Second, it is very possible that the results can all be explained by attempted head movements, consistent with existing findings on mouse eye movements. Thus the findings as presented do not overturn the current thinking, nor suggest that mice are making targeted eye saccades similar to humans.

1. Targeting of eye movements

– The eye movements in response to a left airpuff (Figure 1C) look exactly like one would expect for re-centering saccades – from all initial locations, the eye returns to center. It is only in comparing this to the right airpuff that a 2deg, on average, shift appears. Thus, there is only a +/-1 deg difference for saccades to stimuli on opposite sides of the body. If these saccades are targeting a stimulus, where is it such that left/right are only 2 deg apart? Even for a puff to the ears (Figure 2G) there is at most a 5 deg displacement of target location from central eye position, for a stimulus that is far lateral to the eyes. A much more straightforward description is that these are recentering movements, with a slight bias toward the stimulus side.

– It seems that the authors are assuming that the stimulus location is at the mean endpoint, but that would imply that they had centered the whisker puff stimulus within 2 deg of the center of the visual field for each animal (unlikely), and that the ear puff stimulus is located within 5 deg of the center of the visual field (certainly not true).

– Furthermore, rather than being targeted to a specific location, wherever that happens to correspond to, the saccade endpoints appear highly variable. Indeed, the variability across mean end position based on starting position is similar to the difference due to stimulus location (Figure 2E). The actual variability is probably much greater, as this is masked by averaging a very large number of saccades in figures such as 2C. This averaging makes it appear that all saccades go to the same location, and makes it hard to estimate variability (except to the extent that the error bars after saccade are much larger than for the 5 deg bins pre-saccade). It is important to show the true variability in endpoint position, for example, as a histogram of end positions for left vs right stim. It would also be valuable to see a number of overlaid raw traces for Figures such as 2C,D, to show the true variability before presenting the means.

– The fact that saccade amplitude is equal and opposite to initial eye displacement (negative linear relationship in 1F) is exactly what one would expect for re-centering saccades as well.

2. Confound of head movements/fixation

Previous studies have shown that eye movements are highly disrupted in head-fixed mice, and in fact most eye movements are coupled to attempted head movements. Critically, a previous study (Meyer et al., 2020), and even supplementary data here, show that in head-fixed mice, attempted head movement results in eye saccades in the same direction and proportional in amplitude of the head movement, as one would see in a free moving animal and consistent with the standard gaze-shifting reset model. If the tactile stimulus is evoking attempted head movements (likely) and these are biased in one direction, then that would explain the offset in resulting eye movements. In that case, the reason this study found tactile evoked eye movements is that tactile stimuli evoke head movements, not because they are saccading to the tactile stimulus.

The ideal solution to this would be to perform these experiments in freely moving animals and show that they move their eyes toward the stimulus independent of the head. However, barring that there are a number of key questions that would need to be answered to resolve this issue, potentially based on load cell measurements.

– What is the pattern of head movements resulting from the stimuli? Do stimuli from opposite sides evoke different directions of head movements on average? If so, the bias in head movements could explain the resulting bias in eye movement.

– Do the different types of stimuli (ear, whisker, auditory) evoke different directions/amplitudes of head movement? It could be that ear puff stimulation causes larger eye shifts than whisker not because it is more peripheral, but because it evokes a larger movement. Likewise, auditory stimulation alone may elicit smaller movements or less directionally-biased movements.

– Do the optogenetic manipulations affect head movement as well? This seems likely, since studies in multiple species have shown head movements evoked by SC stimulation or impaired by SC lesion. If so, the shift in eye movements in Figure 3 could be explained by the corresponding change in head movements.

– Finally, which does a better job of predicting the endpoint – head movement or stimulus side? A figure such as 1C-E, but broken up by head movement direction, rather than stimulus side, would test this. Indeed, it would be interesting to see spontaneous eye saccades broken up in this way, to see if there are "targeted" eye movements in the absence of a target.

Additional point

1. Modeling the consequences of eye movements. The authors note that the probability of a saccade increases as a U-shaped curve relative to center position (Figure 4D,E). Indeed, this is exactly what one would expect for recentering saccades – the further the pupil is from center, the more likely it is to reset. However, the authors perform modeling to suggest that instead this represents the goal of keeping the target in the visual field. It's not clear how the modeling supports their claim as presented. The results of the model show that if a stimulus is near the edge of the visual field, then it is more likely to move out of the visual field based on a random movement. Isn't that almost trivially true? It would provide stronger support for their claim if this was quantitative, using physically meaningful values. Specifically, eye position relative to target in 4E varies by roughly +/-10 deg in the data, out of a field of view of 140deg for the mouse. However, this is drastically different than the cartoon shown in 4F, and it doesn't seem that quantitatively a 10deg displacement is going to greatly increase the probability of moving out of a 140deg field of view to the extent shown in Figures4G,H.

Recommendations for the authors

1. The authors state the endpoint should be "strongly dependent" on the stimulus position (p. 3), but highly variable targeting with mean 2 deg shift based on stimulus position does not seem to meet that criteria. If the saccades are indeed targeting a stimulus location, they should be able to systematically vary the stimulus positioning (for example from -30 to 30 deg in the visual field) and show that the targeting eye movements follow.

2. Unless the authors can demonstrate that there is a specific stimulus location that the saccades are targeting, and can explain why they are so highly variable, then the claim that they are making primate-like saccades is not valid. Something like "directionally biased recentering saccades" seems more accurate. But this does not overturn textbook models as they suggest.

3. Throughout the text, there is a notable lack of numerical data for population summary statistics, which makes it difficult to accurately assess the findings. For example, the mean value for left and right endpoints is never provided in the text or figure legends, only the p-value for the comparison (e.g. p 4, line 6). This is not just a style point – it is essential to convey the magnitude of effects observed, as nowhere in the text do the authors state the size of the targeting difference, which is only on the order of a few degrees. In addition, presenting N for number of datapoints (saccades) rather than just number of animals would be valuable.

4. The impact statement, that a "hallmark of human vision" is conserved in mice, seems overstated. In addition to the fact that these saccades don't resemble human saccades, targeted eye movements are not just a hallmark of human vision but many species across the animal kingdom that have retinal specializations.

*Reviewer #3:*

The submitted article titled "Mice Make Targeted Saccades" by Zahler et al. presents some interesting eye movement data that suggests that mice reorient their visual field toward an object of interest. I think the data are thought-provoking and the authors have presented a careful, thorough systematic description of the behavior. My enthusiasm though is tempered for a couple reasons:

First, no vision is really involved in this paradigm so it is hard to be convinced this demonstrates the mice are actually reorienting their visual field. Meyer et al. (2020), as well as the authors, demonstrate that saccadic eye movements coincide with attempted head movements. The mice only made saccades when the mice were "touched" on their head (whiskers or ears). It is not surprising that the mice would attempt to move their heads with a potential threat like that. Therefore, the eye movements might be simply a consequence of the attempted head movement whether or not it is related to a shift in gaze. It is still possible that mice were shifting (or attempting to shift) their visual field, but I would be more convinced if this was demonstrated for targets represented across other senses, especially vision. It might be that more consistent shifts require stimuli farther out in the periphery (remember that mice have a very large visual field due their laterally oriented eyes). In fact, the authors do nicely show this with the U-shaped function in Figure 4. It might be that more conditions are required more peripherally to trigger higher probabilities of saccades and from stimuli represented by sound and vision.

Second, Michaiel et al. (2020) have demonstrated that mice shift their gaze towards an object of interest and this was based on a visual stimulus (cricket) so it should be possible to demonstrate targeted saccades systematically based on vision. The separation of head and eye movement shifts in gaze is not that important of a distinction as nearly all animals combine head and eye movements in some manner to shift gaze, and the superior colliculus jointly represents these saccadic movements.

Suggestions for authors:

I am surprised there was no eye movement triggered towards the auditory stimulus as I have seen the eyes shift towards sounds during experiments. I have also seen eye movements towards tactile stimuli farther back on the body so I do think it is possible to explore this beyond the ears.

Figure 1. I think the data are clear, but it is important to present it fully so that readers understand the similarities *and* the differences with primate behavior. The current presentation is misleading to make it appear more similar to primate behavior than it is in reality.

Figure 1. C,D. It would be helpful to see actual individual saccades plotted here rather than averages (not just the supplemental figure). As plotted, I think this is a little misleading about the precision of the targeted saccades for mice.

Figure 1E. The rightward bias should be explained. The bias probably arises because you are mostly monitoring left eye position. Saccades in mice are on average convergent so the left eye would move more rightward (nasal) than leftward (temporal) (Itokazu et al. 2018; Meyer et al. 2020).

Figure 1B. What is the aggregate probability of a saccade if you integrate over a ~400 ms window after the airpuff? The binning is not clear, but the number appears to be pretty small. For a primate, I would assume this to be close to or at 100%. I assume these are represented in Figure 4.

Figure 1F. Again, I think averages are misleading. Why not just plot all of the data points?

Figure 1G, H. Was regression performed on all data points or just the means? It would be more appropriate to use all of the individual data points.

Figure 2. It would be nice to see airpuff position plotted versus saccade endpoint for each mouse to see if the shift in gaze consistently and systematically follows the position of the airpuff.

Figure 4. Again, would stimuli closer to the edge of visual field result in P approaching 1? Demonstrating this U function for auditory and visual stimuli would be much more convincing to the overall conclusion.

Figure 4. It might be nice to see what this U function would look like for a primate. Also, with more positions, could we predict if the mouse were shifting gaze based on the entire visual field or some restricted specialized region? How does the primate function relate to the fovea size? That might provide some insight about the mouse.

Discussion. When talking about a specialized region of the retina, you should include how the representation of wavelength and binocularity vary across the visual field. The data in Wallace et al. (2011) suggest that mice shift gaze to try to keep the central upper part of their visual field in front of them.

Methods. How often were mice excluded for not making enough saccades?

[Editors’ note: further revisions were suggested prior to acceptance, as described below.]

Thank you for resubmitting your work entitled "A new type of mouse gaze shift is led by directed saccades" for further consideration by *eLife*. Your revised article has been evaluated by Tirin Moore (Senior Editor) and a Reviewing Editor.

The manuscript has been improved but there are some remaining issues that need to be addressed, as outlined below. As you can see, the reviewers see merit in the study, however remain to have major concerns about several of the analyses and interpretations of the analyses and data. This can be partially fixed by carefully interpreting the data in light of the reviewer's comments (who agree with each other on these points) and by reanalyses and experiments. The manuscript should be thoroughly revised to address these concerns that all reviewers agree on.

1) The claim of a 'new' type of eye movement is fairly strong, particularly given that they share many features with standard reset saccades, as well as the fact that this is being demonstrated under the artifact of head fixation. It'd be better to have a less loaded statement, such as "Touch-evoked eye saccade endpoints are biased toward the direction of head motion in head-fixed mice". In any case, the title should certainly include "in head-fixed mice". An alternative suggestion is "Stimulus evoked rapid eye and attempted head movements in the mouse modulated by the superior colliculus".

2) The authors should clarify that *previous studies* have shown that microstimulation in mouse SC produces head and eye movements that are roughly matched to a visual map. What would resolve the major issue that is raised in the introduction of this submission is a direct measurement of combined head and eye movements in mice following SC microstimulation.

3) Short of that, Figure 3 and 4 is the best evidence to support that there is a change in gaze in mice that is consistent with SC eye-head saccades in non-human primates. A major concern with Figure 3 is that it is dependent on how attempted head movements are measured. If there is any sort of threshold in detectable attempted movements in the force meter, weaker/smaller attempted head movements might show an artificial delayed onset. If spontaneous and evoked saccades are genuinely different, then there should be a detectable difference in latency statistics for *matched magnitudes* (not z-scores) of attempted head movements in these two cases. It is possible that spontaneous and evoked saccades are not different, but eye saccades are still genuinely preceding head movements in some cases. Indeed, 30% of spontaneous saccades look just like the evoked saccades with the eye preceding the head. It could be that there is a continuum of gaze changes where smaller/quicker changes have the eye saccade precede the head movement and for larger changes in gaze, the head precedes a resetting of the eye. In that case, it would be important to see it demonstrated in a head-free scenario and/or how it directly relates to SC microstimulation. Figure 2E and 2I is helpful to the case on its own though that the eye movements are not just resetting and maybe could help resolve some of these questions. If the difference holds up in the matched data though, then it would be pretty convincing theses evoked saccades are unique. In that case, maybe the mouse makes faster changes in gaze for more immediate threats, which would definitely be the case for tactile stimulation.

4) For Figure 4, we recommend to see what happens for optogenetic stimulation during spontaneous saccades as well to see if they are driven by circuitry different than SC. it may be difficult to accomplish this with the same experimental timing structure, but it would still be valuable to have the light on for an extended period for some spontaneous saccades and off for other spontaneous saccades. This might not be that beneficial if any of the suggested Figure 3 analysis suggests there is no practical difference between evoked and spontaneous saccades and only a shift in distributions.

5) It is unclear if the data in the end supports the strong claims of a new type of gaze shift that a) is directed toward the stimulus and b) precedes head movement. In the end, these look much like previously described reset saccades, but with an overshoot that is dependent on head movement amplitude. This by itself could be interesting, as it's possible that this represents what happens in an abrupt head movement from rest – a condition that occurs reliably for airpuff stimuli in a head-fixed mouse, and may be present in freely-moving mice albeit less frequently (and hence not seen in previous studies). This would resemble rapid head movements seen in other species, where the eyes "boost" the saccade by moving more rapidly than the head. However, the fact that these are so clearly coupled to head movement in the new data makes it even more important than before to show that this "new" type of movement is present in freely moving mice, rather than trying to interpret attempted head movements in a head-fixed mouse.

6) Although the authors now acknowledge that these are not independent eye movements, but coupled to head movements, they continue to make the association between stimulus location and eye movement. For example, they plot eye movement vs stimulus position, but do not show the equivalent head movements vs stimulus position. They also do not test whether spontaneous saccades, which have zero mean offset, are actually "directed" if they are broken up according to head movement. This was requested in the previous reviews – "What is the pattern of head movements resulting from the stimuli? Do stimuli from opposite sides evoke different directions of head movement on average?" and "Which does a better job of predicting the endpoint – head movement or stimulus side? A figure such as 1C-E, but broken up by head movement direction, rather than stimulus side, would test this. Indeed, it would be interesting to see spontaneous saccades broken up in this way". However, no data is presented in the manuscript to address this. In the review response, the authors describe a regression model but this data is not presented in the text, and it's not clear what the regression is based on – direction of head movement, head displacement during the 0.5 secs before the saccade?

In order to clarify this, it is essential that they show figures such as 2A-F, but broken up by head movement direction rather than stimulus side. This includes breaking up the spontaneous saccades, based on direction of movement during the saccade. Another direct comparison would be side-by-side scatter plots of saccade endpoint vs stimulus side, and saccade endpoint vs head movement direction (and one could do R2 on these plots, rather than a separate regression model across conditions). it can be expected that these will show that head movement predicts the eye movements better than stimulus side does, particularly in cases such as whisker puff. If so, the eye movements are not truly sensory guided or directed toward the stimulus, but are accompanying head movements evoked by getting airpuffed or whiskers touched. Likewise, breaking up spontaneous saccades based on direction of movement will likely show that they are "directed" even in the absence of sensory input.

7) A major claim is that the evoked saccades precede head movements, whereas for spontaneous the head movements precede saccades. This claim is problematic for several reasons. A) For spontaneous, the authors are including movement over the previous 0.5secs, which are slow movements and therefore likely unrelated to the saccade. Saying that movement 0.5 secs before a saccade is "head movements preceding the saccade" is very misleading. B) For both spontaneous and evoked, it's clear that there is a fast movement that occurs right around the time of the saccade. Indeed, the plots of velocity (Figure 3D) show almost identical peaks for evoked and spontaneous, suggesting that these represent the same eye-head coupling, just that evoked saccades do not have movement previous to the stimulus (not surprising, since mice don't know when the stimulus will occur). C) In the plots of velocity, it is clear that both eye and head velocity both rapidly increase right at stimulus onset. Thus, although the peak of head movement is later than eye (presumably due to physical/motor constraints) the onset of the two is nearly simultaneous. Indeed, Figures3E,F would be much more straightforward if presented in terms of head velocity, rather than displacement. Furthermore, calculating latency relative to a threshold on velocity (rather than position), or time of peak head velocity, would likely reveal much closer coupling. Together, these factors make the argument about eye saccades following vs leading very weak. It also points out the challenge in interpreting head movements in a head-fixed animal.

8) The authors clearly demonstrate that there is a higher probability of eye saccade when the eye is initially offset in the opposite direction (Figure 2I-J). This is exactly what one would expect for a reset saccade resulting from head movement, as the eye is reaching the end of its range in that direction. Likewise, they show that amplitude of the eye movement is equal and opposite to initial eye position (Supp Figure 8), again exactly what one would expect for a reset saccade.

9) The endpoint offset relative to center seems to be directly proportional to the head movement amplitude (Supp Figure 6A-D). Combined with the previous point, this suggests that these are similar to previously described saccades, but with an overshoot that is determined by head movement amplitude (rather than stimulus location). Such a mechanism makes sense, since for large head movements this would allow greater dynamic range for the ensuing compensatory phase. Alternately, it could be that this overshoot occurs in head-fixed mice where other feedback mechanisms are lacking.

10) The authors make the claim that head movements depend on eye movement, which would be quite a remarkable finding. However, the actual data is buried in the supplement, and turns out to be a very weak correlation, driven by a small number of outliers with unusually large head movements. Furthermore, only one stimulus condition is presented for this analysis, suggesting that it was not significant for other conditions. Finally, it is easy to envision situations where a weak correlation could arise without a causal role of eye position. For example, if the mouse simply alternates between left and right eye movements on each trial – after a left head movement, the eye would tend to be on the left, so on the subsequent trial when it makes a head movement to the right it would appear to be driven by eye position, even though it's just due to a simple behavioral strategy. If the authors want to support this claim, they should show that it applies for all conditions together, that it is consistent across animals (this is a nested design, and it appears that they used N as # of saccades, rather than # of animals), and that it does not depend on extreme head movements that may represent aberrant conditions.

11) If these are sensory-guided directed saccades, then why do stimuli at the same location result in different saccade locations (2A vs C) and some saccades are directed away from the stimulus (clearly seen in 2G)? At best, these should be described as "directionally biased" rather than "directed".

12) However, a remaining concern is the claim that the data reveal "A new type of mouse gaze shift is led by directed saccades" as stated in the title. The data are fully consistent with the "saccade and fixate" pattern that is widely shared across species (e.g., Land (2019)) and which has recently been identified in freely-moving mice (Michaiel et al. 2020; Meyer et al. 2020). The present study shows that in mice this pattern appears more flexible than previously thought which is a new and important finding. Nevertheless, the rather small (yet important) variations compared to the freely-moving data might not justify the classification as a new type of gaze shift. In particular, Figure 3 suggests that the relative initial contributions of the head and eyes to gaze shifts lie along a continuum and even spontaneous saccades (the baseline condition) can "lead" attempted head rotations; for airpuff-evoked saccades, the head closely follows those saccades (median latency of 30 ms after saccade onset) which corresponds to about half the typical saccade duration (e.g., Sakatani and Isa (2005)). In other words: the saccade and fixate head-eye coupling is preserved for the airpuff-evoked saccades but the relative contributions of the head and eyes are shifted along the continuum. The worry is that the claim of a "new type of mouse gaze shift" might lead to a misperception of this study that shows stronger flexibility of an existing, but not a new, gaze pattern in mice.

13) Figure 2 could use some improvement. They should probably plot saccade start vs saccade end (this data should help their case, but it is presented awkwardly). They should also include average saccade sizes (with direction as sign) versus starting points. Attempted head movement data needs to be looked at in all these scenarios too. It is unclear how well a single trial attempted head movement can be captured. Individual trials look pretty noisy, but the average dynamics look pretty informative. Results may be similar to the eye movements.

14) Another concern is that head-fixed non-human primates will learn to reduce attempted head movements over a short period, once they learn it will not accomplish anything. It is possible the results in Figure 3 are from a similar mechanism. The directed saccades might have provided more reinforcement for learning the failure to move and that is why you rarely see the attempted head movement before the saccade (and why attempted head movements overall appear to be weaker). In addition, spontaneous saccades may decrease over time as the animal is habituated to the experiments and therefore may include fewer saccades during the post-learning period. We recommend that the data from Figure 3 therefore be plotted over time to see if anything observed is from an effect of learning.

---

## [Author Response]

[Editors’ note: the authors resubmitted a revised version of the paper for consideration. What follows is the authors’ response to the first round of review.]

Reviewer #1:[…] Major concerns:1. The paper is quite inconsistent when it comes to the stimuli evoking saccades. The authors state that the main motivation for developing the WISLR task was the absence of targeted saccades towards salient visual stimuli in previous studies. However, the conclusion based on their final simulation (Figure 4F-H) is that the function of saccades in response to tactile/auditory stimuli is to keep visual stimuli in the animal's visual field of view. How do these two things fit together? How are airpuff-evoked saccades related to a visual stimulus leaving the quite large visual field extending to the front and the sides of the animal as assumed in the modeling work? Currently, the relation between the stimuli in Figure 2 that evoked pronounced saccades (e.g., ear airpuff and whisker bar) and the animal's visual field is not clear. In particular, to reconcile the auditory/tactile observations with the simulations it will be important to know whether the tactile/auditory stimuli were within or outside the visual field.

We thank the reviewers for noting that there was a gap between the stated motivation for our study and the conclusions we drew from our final simulation. We should have explicated an implicit step in our logic, which is that natural stimuli are often multimodal such that touch- or sound-evoked saccades could benefit the animal by maximizing the likelihood of detecting the visual component of a natural stimulus. This is likely why other species make gaze shifts towards auditory and tactile stimuli, and why freely moving mice have been observed to make orienting head movements towards auditory stimuli (as well as visual stimuli such as crickets), despite their large visual field. As an aside, the stimuli we deliver to the whiskers and ears do fall within the mouse’s visual field. Nonetheless, our revised manuscript adopts a different framework and we have removed the simulation.

– There is a large body of literature on gaze shifts towards visual, tactile and auditory stimuli in humans and non-human primates. A common observation in these studies is that gaze shifts towards visual targets show a stronger saccade component than gaze shifts towards auditory targets which rely more strongly on head movements. This seems to be somewhat orthogonal to the idea that tactile/auditory but not visual stimuli should result in targeted saccades, in particular if circuits are conserved across species as stated by the authors. It would be important to discuss the motivation/findings in the context of the existing literature which is largely ignored in the current paper.

We thank Reviewer 1 for pointing out this oversight. In our revised introduction, we place the motivation for our experiments in the context of existing literature on gaze shifts towards visual, tactile, and auditory stimuli (page 2). This literature is even more relevant in light of our additional load-cell experiments and analyses showing that head-eye coupling varies for spontaneous and touch-evoked gaze shifts. We agree that it is interesting that, despite the conserved requirement for SC in driving sensory-driven gaze shifts in mice, certain features such as modality-specific differences in reliance on head and eye movements are not conserved. Our discussion goes into more detail on the interspecies commonalities and differences in gaze shifts evoked by stimuli of different modalities and potential mechanisms (pages 22-25).

– The authors show that average saccade trajectories converge on nearly the same endpoint (close to 0 deg) regardless of initial eye position (Figure 1C-E). Would that not support the idea of resetting saccades that move the eyes back to a more central location? There is a statistically significant, yet very small difference in endpoints (< 5 deg) between the left/right stimulus conditions. This seems to be the only evidence for "targeted saccades". However, studies in head-fixed (Sakatani and Isa, 2007; Itokazu et al., 2018) and freely-moving (Meyer et al., 2020) mice have demonstrated that there is a systematic asymmetry (about 5 deg) in the sizes of nasal vs temporal saccades. It would be important to check that this asymmetry cannot explain the difference in saccade endpoints.

We agree with Reviewer 1 that for some tactile stimuli, such as whisker airpuffs, saccade endpoints are close to center and narrowly separated. However, as noted above, mean endpoints for more peripheral stimuli (*i.e.*, ear airpuffs) are quite well separated and not near the center (Figure 2A). Moreover, left and right ear airpuffs evoke saccades in opposite directions from most eye positions; by definition, one of these directions leads away from center and is thus *centrifugal* rather than *centripetal* (Figure 2E). In contrast, both left and right auditory stimuli evoke saccades in the same, centripetal direction from all initial eye positions (Figure 2H). Finally, many puff-evoked saccades that begin towards the center pass through to reach endpoints at eccentricities of 5 to 10 degrees—often more eccentric than the initial eye position—and cannot fairly be termed centripetal (Figure 2E). These results are incompatible with the idea that touch-evoked gaze shifts involve resetting centripetal saccades that move the eyes back to a more central position. Instead, it appears that saccades evoked by more central tactile stimuli may be directed towards more central orbital positions.

The second concern—whether the difference in mean endpoints could be an averaging artifact due to asymmetries in the amplitudes of nasal and temporal saccades—is addressed by showing more raw, unaveraged data, as the reviewers requested. We note in our revised manuscript that we observe this subtle asymmetry. However, it is clear that these asymmetries do not account for the effects we observe. First, in our revision, all data presented in the main figures are from tracking both eyes and represent the average position of both eyes; because saccades are conjugate, each gaze shift involves a nasal saccade of one eye and a temporal for the eye, such that averaging the two eyes eliminates any contribution of the nasal/temporal asymmetry. Second, Figure 2A-D shows the endpoints of all ear airpuff-evoked, auditory airpuff-evoked, whisker-airpuff evoked, and spontaneous saccades for the entire cohort of mice tested. Strikingly, regardless of initial eye position, virtually all saccades elicited by right ear stimulation are directed to the right (nasal), whereas virtually all saccades elicited by left ear stimulation are directed to the left (temporal) (Figure 2E). In other words, from the same initial eye positions, left and right stimuli elicit saccades in opposite directions. Thus, the differences in mean endpoints for these two stimuli are not averaging artifacts of asymmetries in nasal and temporal saccade amplitudes. Moreover, auditory airpuff-evoked saccades obey the same head-eye coupling as touch-evoked saccades, and display the same asymmetries in nasal and temporal saccade amplitudes, yet their endpoints fit the recentering model (Figure 2D, H, Supplementary Figure 6). For these reasons, asymmetries in amplitudes of nasal and temporal saccades do not create an artifact that could explain the directed touch-evoked saccades. We thank the reviewers for raising this potential issue.

In addition, the reviewers’ comments prompted us to examine the ~2-fold discrepancy between the range of saccade amplitudes observed in Meyer *et al.*, 2020 and those reported in our original manuscript. Whereas Meyer *et al.*, 2020 used the methods described in Sakatani and Isa, 2004 to convert pupil position from pixel values into angular eye position, we used the methods described in Stahl *et al.*, 2000. This led to a substantial reduction in the range of saccade amplitudes we observed compared to those in Meyer *et al.*, 2020. In our revised paper, we re-analyze our data using the methods referenced in Meyer *et al.*, 2020 to facilitate direct comparisons. Using this approach, we find that the mean saccade endpoints for left and right ear airpuffs are separated by ~11 degrees (Figure 2A).

– The authors show in Suppl Figure 4 that mice attempted to reorient their heads during saccades and that the magnitude of attempted head movements is predictive of saccade sizes. This indicates that saccades in the WISLR task were part of combined eye/head gaze shifts as reported in freely moving mice (Michaiel et al., 2020; Meyer et al., 2020). Consequently, attempted head movements need to be taken into account when interpreting the data. Indeed, there seems to be a difference in attempted head movements between the two conditions shown in Suppl Figure 4B,C (spontaneous, WISLR). Based on these data, it is not clear if "targeted saccades" were just a by-product of head-free gaze patterns in head-fixed animals (given the low probability of observing a saccade at all).

We agree with the reviewers that we should have devoted more consideration to attempted head movements when interpreting the saccade data. In our revision, head movements are a focus: whereas the original manuscript included only a supplemental figure of head movements that accompany spontaneous and whisker airpuff-evoked saccades, most main figures of the revision (Figures 1, 3, and 4) present both head and eye movements in response to all the stimuli tested, and additional analyses and visualizations that directly address concerns raised by the reviewers.

These efforts confirmed Reviewer 1’s hypothesis that attempted head movements differ between spontaneous and touch-evoked saccades. In fact, there are multiple differences. First, more in-depth analyses found that touch-evoked saccades are coupled to much smaller head movements than are spontaneous saccades of the same amplitude, as Reviewer 1 noticed in our original Supplemental Figure 4. Second, we found that touch-evoked saccades precede attempted head rotations, an observation that is incompatible with the notion that directed eye movements are head-initiated and a “by-product of head-free gaze patterns.” In contrast, spontaneous saccades are coupled to biphasic head rotations, with a slow pre-saccadic phase followed by a fast post-saccadic phase, just as Meyer *et al.* documented in both freely moving and head-fixed mice (Meyer *et al.*, Figure 5C, 7D). This distinction largely accounts for the difference in head-eye amplitude coupling between spontaneous and evoked gaze shifts. Thus, touch-evoked gaze shifts are “part of combined eye/head gaze shifts,” as the Reviewer notes, but not as reported in freely moving mice by Michaiel *et al.* and Meyer *et al.*

The fact that touch-evoked saccades precede rather than react to head movements and are coupled to smaller head movements indicates they are part of a new type of gaze shift: One that permits mice to more rapidly shift their gaze using their eyes. This contrasts with the previously known type of head-eye coupling, present in spontaneous and visually guided gaze shifts, which is driven by slower head movements followed by compensatory saccades that do not appear directed in either head-free or head-fixed contexts. Moreover, the endpoints of touch-evoked saccades are not a “by-product” of attempted head movements. Ff the goal of touch-evoked gaze shifts were merely to reorient the head relative to the location of a tactile stimulus, then head movement direction and amplitude should be constant for stimuli applied to the head (e.g. the ears or whiskers). However, we present new data and analyses that show that attempted head movement direction and amplitude depend on both stimulus location *and* initial eye position (Revision Supplementary Figure 8). This indicates that touch-evoked saccades are part of a new type of gaze shift involving different eye-head coupling in which saccades are not a product of head movements but part of a coordinated eye and head movement that takes into account the current position of the eyes and the location of the stimulus. In other words, head movements are (at least partly) determined by the eyes.

Nevertheless, it is worth contemplating these gaze shifts in the freely moving context in which they would ordinarily be made. To this end, a discussion of gaze shifts in head-free versus head-fixed cats and primates may be illuminating. Sensory-guided saccades in head-fixed primates and cats are often accompanied by attempted head rotations, similar to those we observe during touch-evoked gaze shifts in head-fixed mice. In head-unrestrained cats and primates, gaze shifts are initiated by directed saccades closely followed by directed head movements that are paired with smooth counter-rotatory eye movements. In this way, saccades rapidly shift gaze, and ensuing head and eye movements allow animals to return the eyes to more central locations while maintaining the new gaze direction. It is possible that mice use a similar strategy to shift gaze in response to tactile stimuli in the freely moving condition. The discussion in our revision addresses these ideas on pages 22 and 25.

– The shift in saccade endpoints during optical activation/deactivation of the superior colliculus (SC) is a novel and interesting finding. However, the SC is also involved in generating head movements in non-human primates (e.g., Freedman et al. 1996) and mice (e.g., Wilson et al. (2018)). In particular, Masullo et al. (2019) show that SC stimulation generates both head and eye movements. Again, it would be important to dissociate the contribution of (attempted) head movements to the measured saccades to support the idea that eye movements, and not the associated head movements, determine saccade endpoints.

We thank each of the reviewers for raising related questions regarding whether saccades and their endpoints are “determined” or “explained” by or “by-products of” head movements. This is an important point and one we did not address adequately in our original manuscript. As mentioned in the previous response, if the goal of touch-evoked gaze shifts were only to reorient the head relative to the location of a tactile stimulus, then head movement direction and amplitude should be constant for stimuli applied to the head (e.g. the ears or whiskers), because their distance from the rest of the head are fairly fixed. However, our data show that attempted head movement direction and amplitude depend on both stimulus location *and* initial eye position, similar to directed saccades (Revision Supplementary Figure 8). This indicates that rather than being specified solely by the necessary head movements, with saccades a by-product, touch-evoked gaze shifts involve coordinated eye and head movements that take into account the location of the stimulus *and* the current position of the eyes.

We agree it is interesting to know how SC manipulations affect both head and eye movements during touch-evoke gazed shifts. In our revision, we repeated the optogenetic experiments from our original manuscript while measuring saccades and attempted head movements. We used two unilateral manipulations: inhibition using the light-gated chloride pump eNpHR3.0 and weak optogenetic activation using ChR2. We found that reducing SC activity unilaterally causes ipsiversive shifts in both saccade and head movement amplitudes. In contrast, sub-threshold SC activation causes contraversive shifts in both saccade and head movement amplitudes (Figure 4, Supplementary Figure 10). These results indicate that head and eye movements are coordinately specified by SC.

– The authors use modeling to test the idea that the role of "targeted saccades" is to keep visual targets in the animal's visual field. Why would the U-shaped curve in Figure 4D not be consistent with a resetting saccade model in which the probability of observing a saccade increases with angular distance from the eye's central position? The small differences in trough locations might be related to the asymmetry of nasal vs temporal eye movements (see comment above).

We agree that the U-shaped function relating initial position with saccade probability would also be consistent with a recentering/resetting saccade model; the key difference is the location of the troughs, which for recentering saccades (such as auditory-evoked) are in the center, whereas the troughs for touch-evoked coincide with the more peripheral saccade endpoints (Figure 2I-L). Nevertheless, in our revised manuscript, we remove this modeling. As addressed in response 3, the differences we observe are not related to the asymmetry in nasal and temporal saccade amplitudes.

– The probability of observing a saccade following a non-visual stimulus seems to be quite low (Figure 1B and Suppl Figure 5) and saccade sizes are rather small (~5-15 deg, Suppl Figure 2A) compared to the large visual field of the mouse (>250 deg). Even if mice made targeted saccades: their function in the context of natural behavior is not discussed in the paper.

We thank the reviewers for raising this important point. In our revised manuscript, we have expanded our discussion of what roles these movements may play in natural behaviors in the section “Ethological significance” on pages 24 and 25.

A major concern is that "targeted saccades" are just a by-product of combined eye/head gaze shifts observed in freely moving mice (Michaiel et al., 2020; Meyer et al., 2020). The authors measured attempted head orienting movements in a subset of experiments, and the data strongly indicate that eye/head coupling is preserved in the WISLR task. To test if differences in saccade end points can be explained by head orienting movements, the authors could measure attempted head movements during the different conditions in Figure 2. This could be combined with a regression model to test if saccade end points can be predicted from attempted head movement amplitude.

We thank the reviewer for raising this question. The first part of their question, whether saccade endpoints are a “by-product” of eye-head gaze shifts, is answered in responses 4 and 5. Briefly, the influence of initial eye position on touch-evoked head movement amplitude (Supplementary Figure 8) indicates that touch-evoked head and eye movement amplitudes are coordinated to take into account both stimulus location on the head and current eye position. Thus, touch-evoked saccade endpoints are not “explained by” or “by-products” of head movements.

To address whether eye/head coupling is preserved in the WISLR task, as suggested, we examined the spatiotemporal profiles of attempted head rotations relative to saccades in different stimulus conditions, as well as during spontaneous gaze shifts (Figure 3, Supplementary Figure 6). We found that all stimulus-evoked gaze shifts obeyed a similar head-eye coupling pattern that differed markedly from patterns observed during spontaneous gaze shifts in head-fixed mice, and those previously characterized in freely moving mice. Strikingly, stimulus-evoked saccades preceded attempted head rotations, whereas the opposite held true for head-fixed spontaneous saccades and those previously characterized in freely moving mice*.* In addition, stimulus-evoked saccades of a given amplitude were consistently coupled to smaller attempted head rotations than were spontaneous saccades. Together, these data indicate that mice are capable of multiple types of gaze shift and that directed saccades are not a “by-product of combined eye/head gaze shifts observed in freely moving” and head-fixed mice.

Second, as requested, we trained and cross-validated a regression model to test if saccade amplitudes can be predicted from attempted head movement amplitude. We trained the model using data collected during spontaneous gaze shifts and compared its predictive ability to that of stimulus location. We found that this model is a worse predictor than stimulus location (R^2^ = 0.37 vs. R^2^ = 0.51, *P* = 0.031). That said, there is a correlation between head movement amplitude and saccade amplitude for touch-evoked gaze shifts. However, as noted in the first paragraph of this response, our new data indicate that rather than being a by-product of attempted head rotations, touch-evoked saccades are part of a coordinated head-eye movement in which attempted head movement amplitude varies according to initial eye position. In other words, there is a correlation between touch-evoked head movement direction and saccade direction, but this is because these movements are specified coordinately, not because either head movements or saccades are a “by-product” of the other.

Third, because touch-evoked and spontaneous gaze shifts appear to obey different head-eye coupling patterns, we developed separate regression models for each to examine how well head movement amplitude predicts saccade amplitude at different timepoints relative to saccade onset. For touch-evoked gaze shifts, head movements are only predictive of saccade amplitude after saccade onset, and never as strongly predictive as stimulus side (Author response image 1) . In contrast, for spontaneous gaze shifts, head movements are strongly predictive of saccade amplitude well before saccade onset (Author response image 1) . Once again, these data indicate that mice are capable of multiple types of gaze shifts and that differences in saccade trajectories are not simply due to differences in head movement amplitudes.

**Author response image 1. sa2fig1:** Performance of linear regression models for saccade amplitude and head position. Performance of linear models trained separately for spontaneous (blue) and puff-evoked (red) gaze shifts at different times relative to saccade onset. Each model was trained on distributions of trials matched for initial eye position and amplitude (6 mice, 20 sessions, 1920 matched saccades). Traces denote population mean R2 values +/- s.e.m. Shaded gray area indicates mean saccade duration.

Taken together, these new analyses show that saccade endpoints are not determined by head movement and that directed saccades are not a by-product of previously observed head-eye coupling patterns. Instead, they suggest that touch-evoked directed saccades are part of a novel type of mouse gaze shift involving coordinated movements of the head and eyes. Our revision provides data, analyses, and visualizations that illustrate these points.

In the method section it is mentioned that only saccades occurring within a specific response window following stimulus onset were included in the analysis (p 23). First, it might be useful to explain the choice of the 150 ms response window and the impact of response window length on the results. For example, Itokazu et al. (2018) report a much longer response window for visual stimuli. Second, it might be useful to indicate this response window in Suppl Figure 1A. I would expect that based on the criteria mentioned in the methods section the 2nd saccade in Suppl Figure 1A was excluded from the analysis. Is that correct?

We agree that we need to clarify how we selected this response window. We used this time window because it is the period in which saccade probability is clearly elevated above baseline, although probability sags to a very low but slightly above baseline level for at least 500 ms after stimulus delivery. This is now illustrated in Supplementary Figure 2 in which we show that extending the response window beyond 100 ms does not capture additional evoked saccades and merely dilutes the results with spontaneous saccades, which prompted us shorten our response window to 100 ms. In addition, our response window coincides to the latencies of sensory-guided express saccades in other species, The extremely long latencies (~1 s) observed by Itokazu et al., do not resemble those of sensory-guided saccades in other species, or those we observed in mice, suggesting that those saccades were driven by different mechanisms. Reviewer 1 is also correct that the second saccade in the original Supplementary Figure 1A would not be counted as stimulus-evoked. As Reviewer 1 suggested, we have clarified these points.

Related to this: Figure 1B shows the probability of observing a saccade in different time bins following an airpuff. Maybe I missed this but what was the overall probablity of observing a saccade within the response time window? To get a sense of the reliability of saccade generation, it might be useful to also show cumulative probability (either in the same figure or in a suppl figure). It is hard to guess from the distribution but I would guess that p(saccade in response window) < 0.2.

We agree that these values are useful for the reader and will include them in our revision on page 5.

Reviewer #2:This study makes a bold and provocative claim – mice make targeted eye saccades. This is a significant claim, since although many studies have shown that species without fovea perform saccades, these are generally coupled to head movements, and serve to shift gaze by recentering the eyes following slower compensatory eye movements. Indeed, recent studies have shown that eye movements in mice follow this pattern, and in fact even the eye movements seen in head-fixed mice are associated with attempted head movements. This study claims that head-fixed mice do make targeted eye saccades, when triggered by tactile stimulation to the ears/whiskers. If it is true that mice make saccades similar to foveate animals, and this had just been missed because previous researchers had not used tactile stimuli, this would be an exciting finding not only for mouse vision, but for the vision field at large.However, there are two major issues with the claim that mice are making targeted eye saccades. First, the saccades do not appear to be targeted directly to a stimulus location – rather, they return to center with a slight bias of a couple degrees for saccades "to" stimuli at disparate locations, and in fact there is great variability in saccade endpoints (larger than the difference between target locations). Second, it is very possible that the results can all be explained by attempted head movements, consistent with existing findings on mouse eye movements. Thus, the findings as presented do not overturn the current thinking, nor suggest that mice are making targeted eye saccades similar to humans.1. Targeting of eye movements– The eye movements in response to a left airpuff (Figure 1C) look exactly like one would expect for re-centering saccades – from all initial locations, the eye returns to center. It is only in comparing this to the right airpuff that a 2deg, on average, shift appears. Thus, there is only a +/-1 deg difference for saccades to stimuli on opposite sides of the body. If these saccades are targeting a stimulus, where is it such that left/right are only 2 deg apart? Even for a puff to the ears (Figure 2G) there is at most a 5 deg displacement of target location from central eye position, for a stimulus that is far lateral to the eyes. A much more straightforward description is that these are recentering movements, with a slight bias toward the stimulus side.

We thank the reviewer for pointing out our imprecise and poorly defined use of the term “targeted”. We agree that these movements do not necessarily center the pupil on the stimulus location, particularly given the laterality of the eyes and the limited range of eye positions in mice, and we regret not having made this clear. Rather, our use of “targeted” was meant to indicate that mice direct their pupils to particular orbital positions specified by the location of the tactile stimulus. Nevertheless, a better descriptor for these saccades is “directed,” which we have adopted throughout our revision.

With regards to the movements themselves, this is an instance in which showing more raw, unaveraged data, as the reviewers requested, is clarifying. Figure 2 and Supplemental Figure 4 of the revised manuscript show the endpoints and trajectories of all ear airpuff-evoked, whisker airpuff-evoked, ear tactile-evoked, auditory airpuff-evoked, and spontaneous saccades made by two different cohorts of mice. Although endpoints of whisker airpuff-evoked saccades are somewhat near the center, as the Reviewer noted, those for saccades evoked by the more peripheral ear airpuffs are near the edges of the pupils’ range, with their means separated by ~11 degrees and the distributions overlapping minimally (Figure 2A, Supplementary Figure 4A). Furthermore, virtually all saccades elicited by right ear stimulation are directed to the right (nasal), whereas virtually all saccades elicited by left ear stimulation are directed to the left (temporal) (Figure 2E, Supplementary Figure 4E); by definition, from any eye position, one of these directions leads away from center and is thus *centrifugal* rather than *centripetal* (i.e., not recentering). Moreover, many saccades begin towards the center but pass through to reach endpoints at eccentricities of 5 to 10 degrees and as such cannot be accurately described as recentering (Figure 2E, Supplementary Figure 4E). Thus, it is clear that these ear airpuff saccades are directed, not slightly biased recentering. The less eccentric endpoints of whisker-evoked saccades may simply reflect the fact that the whiskers are located more centrally. Sound-evoked saccade endpoints, in contrast, are well described by the recentering model (Figure 2D, H, Supplementary Figure 4D, H).

In addition, the reviewers’ comments prompted us to examine the ~2-fold discrepancy between the range of saccade amplitudes and eye positions observed in Meyer *et al.*, 2020 and those reported in our study. Whereas Meyer *et al.*, 2020 used the methods described in Sakatani and Isa, 2004 to convert pupil position from pixel values into angular position, we used the methods described in Stahl *et al.*, 2000. This led to a substantial reduction in the range of saccade amplitudes we observed compared to those in Meyer *et al.*, 2020. In our revised paper, we will re-analyze our data using the methods referenced in Meyer *et al.*, 2020 to facilitate direct comparisons.

– It seems that the authors are assuming that the stimulus location is at the mean endpoint, but that would imply that they had centered the whisker puff stimulus within 2 deg of the center of the visual field for each animal (unlikely), and that the ear puff stimulus is located within 5 deg of the center of the visual field (certainly not true).

We again thank the reviewer for pointing out our imprecise and poorly defined use of the term “targeted.” As noted above, our use of this term was meant to imply targeting of a particular orbital location, but we failed to define the term. For a variety of reasons, including their lateral eyes and limited range of eye movements, we agree that mouse saccades could never center both pupils on the stimulus location. We regret not conveying this clearly. As described in response 11, we have adopted the term “directed” in the revised manuscript.

– Furthermore, rather than being targeted to a specific location, wherever that happens to correspond to, the saccade endpoints appear highly variable. Indeed, the variability across mean end position based on starting position is similar to the difference due to stimulus location (Figure 2E). The actual variability is probably much greater, as this is masked by averaging a very large number of saccades in figures such as 2C. This averaging makes it appear that all saccades go to the same location, and makes it hard to estimate variability (except to the extent that the error bars after saccade are much larger than for the 5 deg bins pre-saccade). It is important to show the true variability in endpoint position, for example, as a histogram of end positions for left vs right stim. It would also be valuable to see a number of overlaid raw traces for Figures such as 2C,D, to show the true variability before presenting the means.

We thank the reviewer for noting that averaging can obscure variability. These comments spurred us to develop visualizations that better communicate the overall trends while showing raw, unaveraged data. As suggested, we use scatter plots and histograms to portray the complete distributions of endpoints for saccades evoked by left and right ear airpuffs (Figure 2A-H). These illustrate both the extent of endpoint variability and the fact that this variability is small enough that there is minimal endpoint overlap for ear airpuff-evoked saccades. The reviewer noted that average endpoint location varied as a function of initial eye position, which is also evident in the raw amplitude distributions; however, these raw traces also show that the difference due to stimulus side completely overshadows the variability due to starting eye position (Figure 2E-H). Although comparisons to primates are subjective and we agree with the reviewers that those made in our previous draft were excessive, it is worth mentioning that touch-evoked saccades in primates are also highly variable (~20 degree range of endpoints for a single stimulus, Groh and Sparks 1996i, Figure 3) and much of this variability also depends on initial eye position, possibly due to the inability to make saccade-coupled head movements in head-fixed animals (Groh and Sparks 1996i, Figure 5).

– The fact that saccade amplitude is equal and opposite to initial eye displacement (negative linear relationship in 1F) is exactly what one would expect for re-centering saccades as well.

We agree and should have been clearer in our argument. A negative linear relationship is a necessary but not a sufficient criterion for both targeted and directed saccades. The key difference is whether these linear fits pass through the origin, i.e., saccades of amplitude zero are made when the eyes are in the center. This is true for recentering saccades but not for directed saccades, whose linear fits pass through zero at values statistically insignificantly different from the saccade endpoints for those stimuli (Author response image 2) .

**Author response image 2. sa2fig2:** Relationship between saccade amplitude and initial eye position for ear airpuff- and auditory-evoked saccades. Each point denotes a single saccade. Brighter areas indicate higher point density. Lines are fits of linear regression model. Dashed horizontal line indicates saccade amplitude of 0**°**. Dashed vertical line indicates initial eye position at which linear fit intersects with dashed horizontal line, i.e., the eye position at which predicted saccade amplitude is 0**°**. For left and right ear airpuffs, these intercepts are at -6.35**°** +/- 0.15 and 6.20**°** +/- 0.10 (mean +/- s.e.m) and R2 values are 0.352 and 0.506, respectively. For left and right auditory airpuffs, intercepts are at 0.14**°** (+/- 0.45) and 0.19**°** (+/- 0.43) and R2 values are 0.648 and 0.615, respectively.

2. Confound of head movements/fixationPrevious studies have shown that eye movements are highly disrupted in head-fixed mice, and in fact most eye movements are coupled to attempted head movements. Critically, a previous study (Meyer et al., 2020), and even supplementary data here, show that in head-fixed mice, attempted head movement results in eye saccades in the same direction and proportional in amplitude of the head movement, as one would see in a free moving animal and consistent with the standard gaze-shifting reset model. If the tactile stimulus is evoking attempted head movements (likely) and these are biased in one direction, then that would explain the offset in resulting eye movements. In that case, the reason this study found tactile evoked eye movements is that tactile stimuli evoke head movements, not because they are saccading to the tactile stimulus.

We thank Reviewer 2 for raising this alternative explanation of our data, which we address at length in our revised manuscript. As Reviewer 2 suggested, these questions can be addressed using strain gauge measurements that enable presentation of stimuli at precise craniotopic locations. Indeed, Meyer *et al.*, 2020 performed careful comparisons of head-eye coupling between freely moving and head-fixed mice and found that head-eye coupling—especially during saccades—is highly preserved across these two contexts. In particular, their data show that spontaneous saccades follow head movements or attempted head movements, consistent with the idea that these saccades are a “result” of the head movements. Our new experiments recapitulate this published relationship between saccades and attempted head movements during spontaneous gaze shifts, which we agree is consistent with observations in freely moving mice and the prevailing view of the “gaze-shifting reset model” (Figure 3B, D, G).

The reviewer posits that touch-evoked saccades are also consistent with the standard gaze-shifting recentering model (*i.e.*, tactile stimuli evoke gaze shifts led by head movements that “result” in compensatory recentering saccades). However, our new data show that airpuff-evoked saccades precede attempted head movements—an observation that is incompatible with the idea that they are a “result” of head movements (Figure 3B, D, G). In addition, there are other differences in head-eye coupling between spontaneous and touch-evoked gaze shifts: (1) spontaneous saccades are coupled to biphasic head rotations, with a slow pre-saccadic phase followed by a fast post-saccadic phase, very similar to what is observed during freely moving spontaneous saccades (Figure 3B, D) (Meyer *et al.*, 2020 Figure 5C); and (2) touch-evoked saccades are coupled to smaller head movements overall because they lack a slow pre-saccadic phase (Figure 3B, H). Thus, in both timing and amplitude, touch-evoked gaze shifts are not consistent with the standard model of gaze shifts. We are grateful to Reviewer 2 for suggesting these helpful additional analyses that demonstrate additional differences between touch-evoked and spontaneous gaze shifts.

The ideal solution to this would be to perform these experiments in freely moving animals and show that they move their eyes toward the stimulus independent of the head. However, barring that there are a number of key questions that would need to be answered to resolve this issue, potentially based on load cell measurements.– What is the pattern of head movements resulting from the stimuli? Do stimuli from opposite sides evoke different directions of head movements on average? If so, the bias in head movements could explain the resulting bias in eye movement.– Do the different types of stimuli (ear, whisker, auditory) evoke different directions/amplitudes of head movement? It could be that ear puff stimulation causes larger eye shifts than whisker not because it is more peripheral, but because it evokes a larger movement. Likewise, auditory stimulation alone may elicit smaller movements or less directionally-biased movements.

These are interesting points and we thank Reviewer 2 for raising them. They suggest that the endpoints of directed saccades evoked by tactile stimuli can be “explained” by the direction or amplitudes of head movements elicited by these stimuli. As noted in response 5, each of the reviewers asked a version on this question—whether the saccades are “explained” or “determined” by or are “by-products” of head movements—illustrating that this an important point we did not address adequately in our original manuscript. As the reviewer suggests, it is true that, for example, left ear stimuli elicit primarily leftward saccades that are associated with mostly leftward head movements. However, this correlation does not indicate that head movements cause eye movements. First, as the previous response noted, touch-evoked saccades precede attempted head movements. Second, as noted in response 5, if the goal of touch-evoked gaze shifts were merely to reorient the head relative to the location of a tactile stimulus, with saccades a by-product, then head movement direction and amplitude should be constant across trials for stimuli applied to the head because the distance between the ears or whiskers and the tip of the nose is largely invariant. However, our data show that attempted head movement directions and amplitudes vary with both stimulus location *and* initial eye position (Revision Supplementary Figure 8). This dependence of head movements on eye position is incompatible with the idea that saccade endpoints are “explained” or “determined” by head movements, as it shows that head movements are also determined by the eyes. Instead, this result indicates that head and eye movement amplitudes are coordinated to take into account both stimulus location on the head and current eye position, a result that is incompatible with the “gaze-shifting reset model”.

– Do the optogenetic manipulations affect head movement as well? This seems likely, since studies in multiple species have shown head movements evoked by SC stimulation or impaired by SC lesion. If so, the shift in eye movements in Figure 3 could be explained by the corresponding change in head movements.

We thank the reviewers for raising this point. As noted in response 5 and the immediately preceding response, the dependence of head movement probability, amplitude, and direction on initial eye position and stimulus location indicates that touch-evoked saccades are not “explained” by head movements. Instead, touch-evoked saccades and head movements are specified as coordinated movements that take into account initial eye position and stimulus location.

Although head movements do not explain touch-evoked saccades, we agree that it is important to know how SC manipulations affect touch-evoke gaze shift generation overall and we provide these data in our revision. These manipulations caused parallel effects on saccades and head movements, i.e., weak SC stimulation causes both saccades and head movements to shift contraversively and vice versa for SC inhibition (Figure 4, Supplementary Figure 10). These data indicate that the change in saccade endpoints does not compensate for the effects of these SC manipulations on head movements. Instead, SC activity appears to specify touch-evoked gaze shift amplitude, as is observed in other species.

– Finally, which does a better job of predicting the endpoint – head movement or stimulus side? A figure such as 1C-E, but broken up by head movement direction, rather than stimulus side, would test this. Indeed, it would be interesting to see spontaneous eye saccades broken up in this way, to see if there are "targeted" eye movements in the absence of a target.

We thank the reviewer for raising this issue. As mentioned previously, if one assumes that head-eye coupling is the same for spontaneous and touch-evoked saccades, a regression model trained on a mixture of both is a worse predictor of saccade amplitude than is stimulus side (R^2^ = 0.37 vs. R^2^ = 0.51, p = 0.031). However, we subsequently showed that head-eye coupling differs for spontaneous and touch-evoked saccades (Figure 3, Supplementary Figure 6). We therefore trained separate linear regression models on spontaneous and touch-evoked saccade and attempted head movement data. The model trained specifically on puff-evoked gaze shifts performs better than the spontaneous model and almost as well as stimulus side (R^2^ = 0.46 vs. R^2^ = 0.51, p > 0.05). Thus, stimulus side is a better predictor of saccade amplitude even using a model trained solely on this new type of gaze shift.

Because of the distinct head-eye coupling for spontaneous and touch-evoked saccades, regression models trained on each class of gaze shift perform differently during the perisaccadic epoch: Spontaneous saccade amplitude can be predicted before saccade onset, but touch-evoked saccade amplitude cannot be predicted until after the saccade has ended (Author response image 1) . More importantly, touch-evoked gaze shifts are a different type from those previously observed in mice. Finally, we agree that it may be possible to “break up” the wide distribution of observed spontaneous gaze shifts to create populations with some separation in their endpoint distributions--although the separation would be smaller than that observed for ear airpuff-evoked saccades. However, as detailed in responses 15 and 16, this correlation between head movements and saccades does not indicate that head movements cause eye movements. Instead, our data indicate that eye and head movements are jointly specified as part of a coordinated movement that takes into account both stimulus location and current eye position.

Additional point1. Modeling the consequences of eye movements. The authors note that the probability of a saccade increases as a U-shaped curve relative to center position (Figure 4D,E). Indeed, this is exactly what one would expect for recentering saccades – the further the pupil is from center, the more likely it is to reset. However, the authors perform modeling to suggest that instead this represents the goal of keeping the target in the visual field. It's not clear how the modeling supports their claim as presented. The results of the model show that if a stimulus is near the edge of the visual field, then it is more likely to move out of the visual field based on a random movement. Isn't that almost trivially true? It would provide stronger support for their claim if this was quantitative, using physically meaningful values. Specifically, eye position relative to target in 4E varies by roughly +/-10 deg in the data, out of a field of view of 140deg for the mouse. However, this is drastically different than the cartoon shown in 4F, and it doesn't seem that quantitatively a 10deg displacement is going to greatly increase the probability of moving out of a 140deg field of view to the extent shown in Figures 4G,H.

We agree with the reviewer and have removed the model from our revision.

Recommendations for the authors1. The authors state the endpoint should be "strongly dependent" on the stimulus position (p. 3), but highly variable targeting with mean 2 deg shift based on stimulus position does not seem to meet that criteria. If the saccades are indeed targeting a stimulus location, they should be able to systematically vary the stimulus positioning (for example from -30 to 30 deg in the visual field) and show that the targeting eye movements follow.

Reviewer 2 correctly notes that there are not objective criteria for “strongly dependent” and we should be clearer in defining this. In response to the concern that endpoint differences are small and variable, we would like to point out that endpoints for left and right ear airpuffs are near the edges of the range of eye positions observed and overlap minimally, despite trial-to-trial variability (Figure 2A). In addition, as elaborated in response 11, we recognized a ~2-fold discrepancy between the range of saccade amplitudes observed by us and Meyer *et al.*, 2020 caused by methodological differences. In our revised paper, we will re-analyze our data using the methods referenced in Meyer *et al.*, 2020 to enable direct comparisons. Using this alternate method, the separation of mean endpoints for the ear airpuff stimuli is ~11 degrees. Although comparisons to primates are subjective and those made in our previous draft were excessive, it is worth noting that touch-evoked primate saccades are also highly variable (~20 degree range of endpoints for single stimulus to the hand, Groh and Sparks, 1996i, Figure 1).

As the reviewer suggested, when we varied the locations of airpuffs from peripheral (the ears) to central (the whiskers), the saccade endpoints followed (Figure 2, Supplemental Figure 3, page 8). That said, as noted previously, we agree that these saccades are not necessarily targeting the location of the stimulus itself, particularly with both eyes, given the laterality of the mouse’s eyes and the limited range of eye positions, and that this use of the word “targeted” may have been misleading. In our revision, we more clearly define our terminology and call these movements “directed” rather than “targeted.”

2. Unless the authors can demonstrate that there is a specific stimulus location that the saccades are targeting, and can explain why they are so highly variable, then the claim that they are making primate-like saccades is not valid. Something like "directionally biased recentering saccades" seems more accurate. But this does not overturn textbook models as they suggest.

We thank the reviewers for pointing out the subjectivity inherent in such interspecies comparisons and the overstatement in our claims. As such, and as detailed in the introduction, our revision focuses on our results solely in the context of mouse gaze shifts, even though the caveats listed (saccades targeting a particular orbital location that does not necessarily correspond to the stimulus location and highly variable endpoints) are also observed in primate touch-evoked saccades (*cf.* Groh and Sparks, 1996i). The textbook model for gaze shifts, found in volumes such as the “Oxford Handbook of Eye Movements” and espoused by recent papers such as Meyer *et al.* 2020 and Michaiel *et al.* 2020, posits that mice and all other afoveates make gaze shifts that involve directed head movements followed by recentering saccades. As Michaiel *et al.* concluded in *eLife* last year, “mice do not perform either directed eye saccades or smooth pursuit.” Thus, setting aside any comparison to primates, our findings that mouse make gaze shifts that involve directed rather than recentering saccades and that these saccades do not follow head movements is incompatible with the prevailing view that is also found in textbooks.

3. Throughout the text, there is a notable lack of numerical data for population summary statistics, which makes it difficult to accurately assess the findings. For example, the mean value for left and right endpoints is never provided in the text or figure legends, only the p-value for the comparison (e.g. p 4, line 6). This is not just a style point – it is essential to convey the magnitude of effects observed, as nowhere in the text do the authors state the size of the targeting difference, which is only on the order of a few degrees. In addition, presenting N for number of datapoints (saccades) rather than just number of animals would be valuable.

We agree with the reviewers that we need to include more raw data and statistics. We have changed the visualizations to show exclusively raw data and added precise values to the text and figure legends.

4. The impact statement, that a "hallmark of human vision" is conserved in mice, seems overstated. In addition to the fact that these saccades don't resemble human saccades, targeted eye movements are not just a hallmark of human vision but many species across the animal kingdom that have retinal specializations.

We agree. Because our revision uses a different framework, mouse gaze shifts, and has a new impact statement: “Mice are capable of gaze shifts led by the eyes rather than the head.”

Reviewer #3:The submitted article titled "Mice Make Targeted Saccades" by Zahler et al. presents some interesting eye movement data that suggests that mice reorient their visual field toward an object of interest. I think the data are thought-provoking and the authors have presented a careful, thorough systematic description of the behavior. My enthusiasm though is tempered for a couple reasons:

We thank the reviewer for their kind words and suggestions.

First, no vision is really involved in this paradigm so it is hard to be convinced this demonstrates the mice are actually reorienting their visual field. Meyer et al. (2020), as well as the authors, demonstrate that saccadic eye movements coincide with attempted head movements. The mice only made saccades when the mice were "touched" on their head (whiskers or ears). It is not surprising that the mice would attempt to move their heads with a potential threat like that. Therefore, the eye movements might be simply a consequence of the attempted head movement whether or not it is related to a shift in gaze. It is still possible that mice were shifting (or attempting to shift) their visual field, but I would be more convinced if this was demonstrated for targets represented across other senses, especially vision. It might be that more consistent shifts require stimuli farther out in the periphery (remember that mice have a very large visual field due their laterally oriented eyes). In fact, the authors do nicely show this with the U-shaped function in Figure 4. It might be that more conditions are required more peripherally to trigger higher probabilities of saccades and from stimuli represented by sound and vision.

We agree with Reviewer 3 that the behavioral function of these saccades was not discussed adequately in the original manuscript. Our goal was not to elicit saccades with visual stimuli, but simply to determine whether mice generated sensory-evoked saccades. Implicit in our reasoning was that many objects provide multisensory input, which is presumably why primates, cats, and other species are well known make directed saccades towards auditory and tactile stimuli. Importantly, although sound- and touch-evoked gaze shifts are not elicited by visual stimuli, the visual field, i.e., the portion of the world projected onto the eyes, necessarily changes every time the animal saccades.

Reviewer 3 suggests that the saccades “might simply be a consequence of the attempted head movement.” As noted in response 5 and several subsequent responses, each of the reviewers asked a version on this question—whether the saccades are “explained” or “determined” by or are “by-products” or “consequences” of head movements—illustrating that this an important point we did not address adequately in our original manuscript. As noted in response 5, if the goal of touch-evoked gaze shifts were merely to reorient the head relative to the location of a tactile stimulus, with saccades a by-product, then head movement direction and amplitude should be constant across trials for stimuli applied to the head, because the airpuffs are at fixed locations relative to the head. However, our data show that attempted head movement directions and amplitudes vary with both stimulus location *and* initial eye position (Revision Supplementary Figure 8). This dependence of head movements on eye position is incompatible with the idea that saccade endpoints are “explained” or “determined” by head movements, as it shows that head movements are also determined by the eyes. Instead, they suggest that head and eye movement amplitudes are coordinated to take into account both stimulus location on the head and current eye position, a result that is incompatible with the “gaze-shifting reset model”. Likewise, that touch-evoked saccades precede attempted head rotations is incompatible with the notion that directed eye movements are a “consequence” of directed head movements (Figure 3B, D, E-G). In contrast, spontaneous saccades are a coupled to biphasic head rotations, with a slow pre-saccadic phase followed by a fast post-saccadic phase (Figure 3B, D, H). An important outcome of this difference is that touch-evoked saccades are coupled to smaller head movements, further evidence that the saccades we observed are not an expected “consequence” of head movements or of the previously published relationship between head and eye movements (Figure 3B, D, H).

We agree that it would be interesting if visual stimuli also evoked directed saccades. That said, and although absence of evidence is not evidence of absence, many groups (including ours) have unsuccessfully attempted to elicit directed saccades using a variety of visual stimuli. For completeness, we include an example of such negative data in our revision, finding that a very bright LED does not evoke gaze shifts in head-fixed mice (Figure 1G, L). The reviewer suggests it may be the physical proximity of a tactile stimulus that explains this difference, but less proximal auditory stimuli also evoked saccades (albeit at a lower probability, and these were not directed) (Figure 1F, K; Figure 2D, H, L; Supplementary Figure 4D, H, L). Similarly, this difference is not simply a result of stimulus eccentricity, as the visual stimulus was placed at the same peripheral location as the ear airpuffs. Moreover, for tactile and auditory stimuli, the probability of making a gaze shift is not simply a function of how peripheral stimuli are. In fact, we see much larger modulation of saccade probability by eye position than by stimulus location. For example, ear airpuffs were roughly twice as likely as whisker airpuffs to elicit saccades overall, but mice were roughly four times as likely to saccade in response to a central tactile stimulus (right whisker airpuff) when the eyes began to the left than they were to saccade in response to a peripheral tactile stimulus (right ear airpuff) when the eyes began to the right (Figure 2I, J). In contrast, mice did not respond to a peripheral visual stimulus regardless of their initial eye position (Figure 1G). Thus, it is not simply that the visual stimuli were insufficiently proximal or peripheral. It is possible that an as yet unidentified visual stimulus could robustly elicit saccades in head-fixed animals, although it is worth noting that every tactile stimulus we have delivered to any location on the mouse’s body, from the tip of the nose to the trunk, elicited saccades. Therefore, it seems more likely that the effects we and others have observed may simply reflect differences across modalities, something we discuss in greater depth in our revision.

Second, Michaiel et al. (2020) have demonstrated that mice shift their gaze towards an object of interest and this was based on a visual stimulus (cricket) so it should be possible to demonstrate targeted saccades systematically based on vision.

We thank the reviewer for mentioning this highly relevant study. However, it is critical to clarify what Michaiel *et al.*, 2020 claimed. They conclude their abstract by asserting that during cricket hunting “orienting movements are driven by the head, with the eyes following in coordination to sequentially stabilize and recenter the gaze.” In their results, they note that “eye movements are not targeting the cricket more precisely, but simply ‘catching up’ with the head, by re-centering following a period of stabilization.” Moreover, they state that “mice do not perform either directed eye saccades or smooth pursuit.” In other words, throughout their manuscript, they explicitly state that mice do not make targeted (or directed) saccades based on vision (for the stimuli they tested, at least). Instead, they contend that visual stimuli evoke head movements directed towards the stimulus location, followed by compensatory saccades that recenter the eyes. This is in marked contrast to our observation that touch-evoked eye movements are directed, precede attempted head rotations, and shape gaze shift direction, amplitude, and probability. As noted in our previous response, to our knowledge no group has observed an innate ability of mice to make directed saccades based on vision; Itokazu et al. were able to train mice to make saccades in response to visual stimuli, but this required many months and the extremely long latencies (~1 s) relative to those of sensory-guided saccades in other species or, as our data show, in mice suggest that these saccades were not sensory-guided. Thus, ours is the first study of which we are aware to show that mice make directed, sensory-guided saccades. We agree it is interesting that we observe this separation on the basis of sensory modality and address this in our revised discussion.

The separation of head and eye movement shifts in gaze is not that important of a distinction as nearly all animals combine head and eye movements in some manner to shift gaze, and the superior colliculus jointly represents these saccadic movements.

We agree that our previous draft failed to place our study in the appropriate context and as such the significance of this distinction was obscured. As mentioned in our previous response, Michaiel *et al.* explicitly stated that mice do not make directed saccades. Instead, they bolstered the model found in textbooks such as the Oxford Handbook of Eye Movements that mouse gaze shifts are led by the head, with these head movements triggering compensatory saccades that recenter the eyes. The emphasis of this idea in the penultimate sentence of an *eLife* abstract from last year and by each of the reviewers of our original manuscript underscore the importance of this longstanding distinction and how surprising our findings are.

Another point the reviewer raises is that SC is known to drive head and eye movements. It is important to clarify that although SC is known to drive head and eye movements in many species, the prevailing view holds that recentering saccades driven by head movements (*i.e.,* the quick phase of nystagmus), such as those in the previously observed in mouse gaze shifts, arise independently of SC and rely on brainstem circuits receiving vestibular input from the semicircular canals (reviewed in Curthoys, 2002). Indeed, Michaiel *et al.* conclude that “orienting movements are driven by the head, with the eyes following in coordination to sequentially stabilize and recenter the gaze,” and Meyer *et al.* invoke vestibular mechanisms as a likely explanation for the saccades they observed during spontaneous gaze shifts. Once again, such recentering movements are thought to be generated independent of SC circuits, which could explain the different head-eye coupling we observe for spontaneous and touch-evoked saccades. To our knowledge, no study has ever shown that mice make SC-dependent gaze shifts, although multiple studies have shown SC stimulation can evoke saccades and head movements in mice.

We agree that our observations that mice generate SC-dependent, sensory-evoked, saccade-led gaze shifts feel intuitive, but this is because they reconcile the longstanding logical disconnect between the aforementioned studies of mouse gaze shifts and SC microstimulation results. Indeed, it was such an intuition motivated us to undertake this study and to test a broader range of stimuli than had been previously studied. However, the fact that the last half-century of studies of afoveates, including high-profile papers in the last year such as Michaiel *et al.* and Meyer *et al.* unanimously asserted that mouse gaze shifts are (1) led by the head and (2) involve recentering saccades that are believed to be SC-independent shows the importance of this distinction and highlights the novelty of our findings. Our revision discusses this literature in more depth in order to more clearly convey the importance and implications of these findings.

I am surprised there was no eye movement triggered towards the auditory stimulus as I have seen the eyes shift towards sounds during experiments. I have also seen eye movements towards tactile stimuli farther back on the body so I do think it is possible to explore this beyond the ears.

We were also surprised. In fact, while doing these experiments we also had the impression that the auditory-evoked saccades we observed were directed. However, once we analyzed the trajectories and endpoints it became clear that this notion was incorrect. That said, and although our tests were not exhaustive, absence of evidence is not evidence of absence, and other auditory stimuli we did not identify may be capable of eliciting directed saccades.

Figure 1. I think the data are clear, but it is important to present it fully so that readers understand the similarities *and* the differences with primate behavior. The current presentation is misleading to make it appear more similar to primate behavior than it is in reality.Figure 1. C,D. It would be helpful to see actual individual saccades plotted here rather than averages (not just the supplemental figure). As plotted, I think this is a little misleading about the precision of the targeted saccades for mice.

As noted previously, we thank the reviewers for pointing out the limitations of interspecies comparisons and have reframed our manuscript to focus on mouse gaze shifts. In addition, we present much more raw data (e.g., endpoints and trajectories for all saccades rather than means) as the reviewers suggested.

Figure 1E. The rightward bias should be explained. The bias probably arises because you are mostly monitoring left eye position. Saccades in mice are on average convergent so the left eye would move more rightward (nasal) than leftward (temporal) (Itokazu et al. 2018; Meyer et al. 2020).

This is an interesting point. It is possible that this difference is attributable to the asymmetry in nasal and temporal saccades suggested. On the other hand, it seems likely that the apparent bias is a result of the somewhat arbitrarily defined “center” eye position. For example, if we define the center as the mean endpoint of spontaneous saccades, rather than the median eye position, the apparent rightward bias is greatly reduced. Nevertheless, to eliminate any potential nasal/temporal bias caused by recording from one eye, we recorded both eyes and averaged their positions in the revised manuscript.

Figure 1B. What is the aggregate probability of a saccade if you integrate over a ~400 ms window after the airpuff? The binning is not clear, but the number appears to be pretty small. For a primate, I would assume this to be close to or at 100%. I assume these are represented in Figure 4.Figure 1F. Again, I think averages are misleading. Why not just plot all of the data points?Figure 1G, H. Was regression performed on all data points or just the means? It would be more appropriate to use all of the individual data points.Figure 2. It would be nice to see airpuff position plotted versus saccade endpoint for each mouse to see if the shift in gaze consistently and systematically follows the position of the airpuff.

We thank Reviewer 3 for these suggestions. We provide aggregate probabilities in our revision on page 5. We plot all data points throughout the revision. We show airpuff position vs. endpoint in Revision Supplementary Figure 3.

[Editors’ note: what follows is the authors’ response to the second round of review.]

Essential revisions:1) The claim of a 'new' type of eye movement is fairly strong, particularly given that they share many features with standard reset saccades, as well as the fact that this is being demonstrated under the artifact of head fixation. It'd be better to have a less loaded statement, such as "Touch-evoked eye saccade endpoints are biased toward the direction of head motion in head-fixed mice". In any case, the title should certainly include "in head-fixed mice". An alternative suggestion is "Stimulus evoked rapid eye and attempted head movements in the mouse modulated by the superior colliculus".

We thank the reviewers for their suggestions. We agree that our claims of a new type of gaze shift is fairly strong and have attempted to modulate the tone. Therefore, we have attempted to combine the above suggestions and retitled our manuscript “Superior colliculus drives stimulus-evoked directionally biased saccades and attempted head movements in head-fixed mice.”

2) The authors should clarify that *previous studies* have shown that microstimulation in mouse SC produces head and eye movements that are roughly matched to a visual map. What would resolve the major issue that is raised in the introduction of this submission is a direct measurement of combined head and eye movements in mice following SC microstimulation.

We thank the reviewers for raising this issue. In order to better motivate our studies, we now provide in Figure 1 our own data on combined head and eye movements evoked by SC optogenetic stimulation. These data show that stimulation-evoked saccades are coincident with attempted head movements and directionally biased. This differs from the temporal relationship reported for mouse spontaneous and visually guided gaze shifts which are thought to recenter the eyes. This finding nicely motivates the subsequent experiments and analyses.

3) Short of that, Figure 3 and 4 is the best evidence to support that there is a change in gaze in mice that is consistent with SC eye-head saccades in non-human primates. A major concern with Figure 3 is that it is dependent on how attempted head movements are measured. If there is any sort of threshold in detectable attempted movements in the force meter, weaker/smaller attempted head movements might show an artificial delayed onset. If spontaneous and evoked saccades are genuinely different, then there should be a detectable difference in latency statistics for *matched magnitudes* (not z-scores) of attempted head movements in these two cases. It is possible that spontaneous and evoked saccades are not different, but eye saccades are still genuinely preceding head movements in some cases. Indeed, 30% of spontaneous saccades look just like the evoked saccades with the eye preceding the head. It could be that there is a continuum of gaze changes where smaller/quicker changes have the eye saccade precede the head movement and for larger changes in gaze, the head precedes a resetting of the eye. In that case, it would be important to see it demonstrated in a head-free scenario and/or how it directly relates to SC microstimulation. Figure 2E and 2I is helpful to the case on its own though that the eye movements are not just resetting and maybe could help resolve some of these questions. If the difference holds up in the matched data though, then it would be pretty convincing theses evoked saccades are unique. In that case, maybe the mouse makes faster changes in gaze for more immediate threats, which would definitely be the case for tactile stimulation.

We thank the reviewers for raising this issue and for suggesting analyses whose results we agree are “pretty convincing.” As suggested, we provide an Author response image in which we compare trials matched for the raw attempted head movement magnitude. Because different mice generate different ranges of raw magnitudes of head movements (which is our reason for Zscoring attempted head movements for population analyses), we provide separate plots for 5 mice showing within-session comparisons of interleaved spontaneous and evoked gaze shifts (Author response image 3) . The results from each mouse are consistent with those in Figure 4, with spontaneous saccades on average preceded by slow attempted head movements whereas touch-evoked saccades are not. Thus, this matched analysis shows that spontaneous and touch-evoked gaze shifts differ and that the differences we observe are not related to the amplitudes of the evoked gaze shifts. We agree with the reviewers’ suggestion that the immediacy of tactile threats may warrant faster gaze shifts and raise this possibility in the discussion.

**Author response image 3. sa2fig3:** Ear airpuff-evoked attempted head movement latencies using matched, raw strain gauge measurements for individual mice. (A) Left, distributions of raw attempted head movement amplitude-matched trials accompanying spontaneous (blue) and ear airpuff-evoked (red) gaze shifts for a single animal. Trials are matched for raw attempted head movement magnitude at 150 ms after saccade onset as used for amplitude analyses in Figure 4. Middle, mean attempted head movement traces for raw magnitude-matched head movements accompanying spontaneous (blue) and ear airpuff-evoked (red) gaze shifts. Saccade onset is at 0 ms. Right, head movement latencies for magnitude-matched head movements accompanying spontaneous (blue) and ear airpuff-evoked (red) gaze shifts. Dashed vertical line indicates saccade onset. (B-E) As in A for four other animals. Medians indicated by blue and red vertical lines. Spontaneous and evoked medians are significantly different for all animals (p< 10-5, permutation test)

We also agree with the reviewers that there are some spontaneous saccades without preceding head movements. However, it is important to note we, like most in the field, use “spontaneous” as a non-specific, catch-all descriptor for any saccade made when the experimenters were not delivering a sensory stimulus. As such, these “spontaneous” saccades likely include saccades evoked by auditory or tactile stimuli not under experimental control, such as sound or vibration from a door closing down the hall or the elevator ascending in the adjacent shaft, or from stochastic activity in the saccadic circuitry, including SC. Likewise, what we quantify as evoked saccades, i.e., those in the post-stimulus period when saccade probability is elevated, undoubtedly includes spontaneous gaze shifts that began before the stimulus was delivered and that were unrelated to the stimulus itself. Thus, we would not expect either population of gaze shifts to be pure.

4) For Figure 4, we recommend to see what happens for optogenetic stimulation during spontaneous saccades as well to see if they are driven by circuitry different than SC. it may be difficult to accomplish this with the same experimental timing structure, but it would still be valuable to have the light on for an extended period for some spontaneous saccades and off for other spontaneous saccades. This might not be that beneficial if any of the suggested Figure 3 analysis suggests there is no practical difference between evoked and spontaneous saccades and only a shift in distributions.

We thank the reviewers for this suggestion. We include Author response images on the effects of weak and strong SC optogenetic stimulation on spontaneous saccades (Author response image 4; Author response image 5). We observed an increase in saccade probability during weak optogenetic stimulation. Saccades generated during weak optogenetic stimulation appear spontaneous-like in that they tend to be preceded by attempted head movements. In contrast, saccades generated during strong optogenetic stimulation resemble touch-evoked saccades in that they are not preceded by head movements. These data suggest that spontaneous saccades may also be modulated by SC.

**Author response image 4. sa2fig4:** Effect of subthreshold ChR2 stimulation on spontaneous saccades. (A) Saccade probability before and during (blue shading) a 1 second subthreshold ChR2 stimulus. (B) Mean trajectories of spontaneous saccades occurring during LED off (black) and LED on (blue) periods. (C) Mean attempted head movement amplitudes accompanying spontaneous saccades occurring during LED off (black) and LED on (blue) periods. (D) Mean velocities of spontaneous saccades occurring during LED off (black) and LED on (blue) periods. (E) Mean head movement velocities accompanying spontaneous saccades occurring during LED off (black) and LED on (blue) periods. (F, G) Timing of attempted head movements relative to spontaneous saccades occurring during LED off and LED on periods. Each row corresponds to a single gaze shift. Darker shades indicate larger attempted head displacement. Purple hues denote attempted displacement in the same direction as the saccade (ipsiversive; note that this is contraversive relative to stimulated SC), and orange hues denote displacement in the opposite direction of the saccade (contraversive). Dashed vertical line indicates time of saccade onset. Trials are sorted by latency of attempted head movements. (H, I) As in (F, G) but for attempted head movement velocity. (J) Instantaneous fraction of LED off spontaneous saccades with ipsiversive attempted head displacements. (K) As in (J) but for attempted head velocity.

**Author response image 5. sa2fig5:** Effect of suprathreshold ChR2 stimulation on head-eye coupling. (A) Saccade probability during a 40 ms suprathreshold ChR2 stimulus. (B) Mean trajectories of saccades during spontaneous (black) and opto-evoked (blue) gaze shifts. (C) Mean attempted head movement amplitudes accompanying saccades during spontaneous (black) and opto-evoked (blue) gaze shifts. (D) Mean velocities of saccades during spontaneous (black) and opto-evoked (blue) gaze shifts. (E) Mean head movement velocities accompanying saccades during spontaneous (black) and opto-evoked (blue) gaze shifts. (F, G) Timing of attempted head movements relative to saccades occurring spontaneous (black) and opto-evoked (blue) gaze shifts. Each row corresponds to a single gaze shift. Darker shades indicate larger attempted head displacement. Purple hues denote attempted displacement in the same direction as the saccade (ipsiversive, which is contraversive to stimulated SC), and orange hues denote displacement in the opposite direction of the saccade (contraversive). Dashed vertical line indicates time of saccade onset. Trials are sorted by latency of attempted head movements. (H, I) As in (F, G) but for attempted head movement velocity. (J) Instantaneous fraction of trials with ipsiversive attempted head displacements relative to saccade onset for spontaneous (black) and opto-evoked (blue) gaze shifts. (K) As in (J) but for attempted head velocity.

5) It is unclear if the data in the end supports the strong claims of a new type of gaze shift that a) is directed toward the stimulus and b) precedes head movement. In the end, these look much like previously described reset saccades, but with an overshoot that is dependent on head movement amplitude. This by itself could be interesting, as it's possible that this represents what happens in an abrupt head movement from rest – a condition that occurs reliably for airpuff stimuli in a head-fixed mouse, and may be present in freely-moving mice albeit less frequently (and hence not seen in previous studies). This would resemble rapid head movements seen in other species, where the eyes "boost" the saccade by moving more rapidly than the head. However, the fact that these are so clearly coupled to head movement in the new data makes it even more important than before to show that this "new" type of movement is present in freely moving mice, rather than trying to interpret attempted head movements in a head-fixed mouse.

We agree with the reviewers that our claims of a “new” type of gaze shift may have been too strong. We also agree that it would be ideal to examine this behavior in the context of freely moving mice, but do not currently have the ability to do these experiments and could not realistically develop the necessary preparations quickly enough to submit our revision “in a timely manner” as requested. Therefore, we have changed the title of our manuscript, as suggested, and modulated language throughout the revised manuscript to emphasize that our analyses are confined to head-fixed mice.

6) Although the authors now acknowledge that these are not independent eye movements, but coupled to head movements, they continue to make the association between stimulus location and eye movement. For example, they plot eye movement vs stimulus position, but do not show the equivalent head movements vs stimulus position. They also do not test whether spontaneous saccades, which have zero mean offset, are actually "directed" if they are broken up according to head movement. This was requested in the previous reviews – "What is the pattern of head movements resulting from the stimuli? Do stimuli from opposite sides evoke different directions of head movement on average?" and "Which does a better job of predicting the endpoint – head movement or stimulus side? A figure such as 1C-E, but broken up by head movement direction, rather than stimulus side, would test this. Indeed, it would be interesting to see spontaneous saccades broken up in this way". However, no data is presented in the manuscript to address this. In the review response, the authors describe a regression model but this data is not presented in the text, and it's not clear what the regression is based on – direction of head movement, head displacement during the 0.5 secs before the saccade?In order to clarify this, it is essential that they show figures such as 2A-F, but broken up by head movement direction rather than stimulus side. This includes breaking up the spontaneous saccades, based on direction of movement during the saccade. Another direct comparison would be side-by-side scatter plots of saccade endpoint vs stimulus side, and saccade endpoint vs head movement direction (and one could do R2 on these plots, rather than a separate regression model across conditions). it can be expected that these will show that head movement predicts the eye movements better than stimulus side does, particularly in cases such as whisker puff. If so, the eye movements are not truly sensory guided or directed toward the stimulus, but are accompanying head movements evoked by getting airpuffed or whiskers touched. Likewise, breaking up spontaneous saccades based on direction of movement will likely show that they are "directed" even in the absence of sensory input.

We thank the reviewers for the analyses suggested and inspired by this comment. Specifically, this comment asks us to illustrate head movements elicited by stimuli, now added to Figure 3, for figures showing eye movements broken up by head movement direction (Author response image 6) , and for R^2^ values of regression models predicting saccade endpoint from stimulus side versus head movement direction. As expected, because head and eye movements are highly correlated, head movements mirror eye movements and thus predict eye movements better than stimulus side for many individual stimuli. The reviewers suggest that such a result would indicate that “eye movements are not truly sensory guided or directed toward the stimulus, but are accompanying head movements evoked by getting airpuffed or whiskers touched”. We respectfully disagree that this correlation implies causation for the following reasons, which we elaborate on below: (1) whisker and ear airpuff trials matched for head movement direction and amplitude yield well-separated saccade endpoint distributions, indicating that head movements alone do not predict saccade endpoints; (2) head movements are worse at predicting saccade endpoints than are stimulus locations for multiple stimuli; (3) the distribution of saccade endpoints for a stimulus reflects the relationship between starting eye position and evoked saccade amplitude, which differs across stimuli; (4) head movement direction, amplitude, and probability depend on initial eye position, suggesting that head movements do not simply “accompany” saccades.

**Author response image 6. sa2fig6:** Endpoints and trajectories of sensory-evoked saccades organized by head movement direction. (A-D) Endpoints for ear airpuff-, whisker airpuff-, ear tactile-, and auditory airpuff-evoked saccade organized by head movement direction. Top, schematics of stimuli. Middle, scatter plots showing endpoints of all saccades for all animals (n = see below, 5 animals) made spontaneously (blue) and during left (green) and right (magenta) attempted head movements. Darker shading indicates areas of higher density. Bottom, histograms of endpoint distributions for spontaneous and evoked saccades. (E) As in (A-D) but with endpoints for spontaneous saccades organized by head movement direction. (F-J) Trajectories of individual stimulus-evoked saccades. Each arrow denotes the trajectory of a single saccade. Saccades are sorted according to initial eye positions, which fall on the dashed diagonal line. Saccade endpoints are indicated by arrowheads. Because the probability of evoked gaze shifts differed across stimuli, data for ear and whisker airpuffs are randomly subsampled (15% and 30% of total trials, respectively) to show roughly equal numbers of trials for each condition. Saccade numbers in A-J: ear airpuff sessions, spontaneous = 7146, evoked left head movement = 951 (143 in E), evoked right head movement = 1204 (181 in E); whisker airpuff sessions: spontaneous = 7790, evoked left head movement = 486 (146 in F), evoked right head movement = 560 (168 in F); ear tactile sessions, spontaneous = 6706, evoked left head movement = 167 evoked right head movement = 152; auditory sessions, spontaneous = 10240 evoked left head movement = 134, evoked right head movement = 164; spontaneous = 7146, spontaneous left head movement = 3565 (171 in J), spontaneous right head movements = 3581 (168 in J).

To test the hypothesis that head movements explain saccade endpoints, we performed an additional analysis in which we matched trials by attempted head movement direction and amplitude across stimuli. If the distribution of head movements determines the distribution of saccade endpoints, then saccade endpoints for different stimuli should be similar if we compare trials matched for head movement direction and amplitude. Comparisons of matched left whisker/ear airpuff trials and comparisons of matched right whisker/ear airpuff trials yielded well separated endpoints (left ear: -3.93 ± 4.42° vs. left whiskers: -1.56 ± 3.93°, p <10-9, n = 253 matched trials; right ear: 4.47 ± 3.00° vs. right whiskers: 2.05 ± 3.33°, p <10-24, n = 403 matched trials; mean ± s.d., Welch’s t-test) (Figure 4—figure supplement 4). This illustrates that the saccade endpoint differences between stimuli are not simply the result of head movement differences between stimuli and argues there are other differences in the gaze shifts evoked by different stimuli.

Because of the strong correlation between saccades and attempted head movements and the symmetry of responses to a given stimulus on opposite sides, it is not surprising that head movement direction is as good or better than stimulus side at predicting saccade endpoints for left and right delivery of a single stimulus. However, this correlation does not indicate causation. For example, if the only association between stimulus location and saccade endpoint is that saccades accompany head movements, then it should be true that head movements alone are also able to predict saccade endpoints across stimulus locations. To test this, we pooled equal numbers of left and right whisker and ear airpuff-evoked gaze shifts (1046 of each) and asked whether saccade endpoints were better predicted by head movement direction and amplitude or stimulus location. These analyses show that stimulus location is a better predictor of saccade endpoint (R^2^ = 0.537) than is attempted head movement direction (R^2^ = 0.416) or even head movement direction and amplitude (R^2^ = .480).

The preceding results suggest that saccade endpoints are not simply the result of biases in evoked head movements. In our revision, we present data suggesting that saccade endpoint differences between stimulus types and locations are due to differences in the relationship between starting eye position and saccade amplitude (Figure 5). For example, a 5° saccade can be directed towards central or peripheral endpoints depending on the initial position of the eyes from which it is made. It follows that changing the average initial eye position from which a 5° saccade is made will shift the distribution of saccade endpoints accordingly. This is exactly what we observe. This is most clearly illustrated by comparing the relationship between initial eye position and saccade amplitude for left and right ear airpuffs. The lines of best fit for these distributions are well separated, consistent with the resultant endpoint separation. In contrast, the lines of best fit for left and right whisker airpuffs are less separated.

Lastly, we present additional data showing that attempted head movement direction and amplitude depend on initial eye position (Figure 5, Figure 5—figure supplement 4). In contrast, both saccade and attempted head movement direction and amplitude depend weakly if at all on initial attempted head displacement (Figure 5—figure supplement 4). This dependence of attempted head movements on initial eye position argues that eye movements do not simply “accompany” head movements. We contend that a more parsimonious explanation is that evoked head and eye movements are jointly specified to account for stimulus location and starting eye position. This explanation is further supported by our data showing that evoked head and eye movements are coincident and SC-dependent. Nevertheless, it is possible that these relationships are an artifact of head fixation and we acknowledge this in our discussion.

7) A major claim is that the evoked saccades precede head movements, whereas for spontaneous the head movements precede saccades. This claim is problematic for several reasons. A) For spontaneous, the authors are including movement over the previous 0.5secs, which are slow movements and therefore likely unrelated to the saccade. Saying that movement 0.5 secs before a saccade is "head movements preceding the saccade" is very misleading. B) For both spontaneous and evoked, it's clear that there is a fast movement that occurs right around the time of the saccade. Indeed, the plots of velocity (Figure 3D) show almost identical peaks for evoked and spontaneous, suggesting that these represent the same eye-head coupling, just that evoked saccades do not have movement previous to the stimulus (not surprising, since mice don't know when the stimulus will occur). C) In the plots of velocity, it is clear that both eye and head velocity both rapidly increase right at stimulus onset. Thus, although the peak of head movement is later than eye (presumably due to physical/motor constraints) the onset of the two is nearly simultaneous. Indeed, Figures3E,F would be much more straightforward if presented in terms of head velocity, rather than displacement. Furthermore, calculating latency relative to a threshold on velocity (rather than position), or time of peak head velocity, would likely reveal much closer coupling. Together, these factors make the argument about eye saccades following vs leading very weak. It also points out the challenge in interpreting head movements in a head-fixed animal.

We thank the reviewer for pointing out the potential flaws in our methods for measuring head movement latencies during spontaneous and evoked saccades. In particular, as the reviewer notes in (A), we agree that latency statistics based on event detection on continuous signals, such as from the load cells, are potentially problematic, and as such have removed the latency measurements in Figure 3G (now Figure 4G). We also agree there is no predictive power of slow head movements at 500 ms prior to saccade onset—as is customary in the field, we chose to show a time window that starts at a baseline timepoint well before movement onset. However, we respectfully disagree with the statement that slow head movements are unrelated to the saccade. In place of latency statistics, Figure 3G (now 4G) now shows a plot demonstrating that slow head movements are predictive of saccade direction starting roughly 200 milliseconds before onset of spontaneous but not evoked saccades. Likewise, we have amended panels 3E and F (now 3E and F) to illustrate whether head displacements are in the same or opposite direction as the saccade. These data more clearly illustrate that presaccadic head displacements are strongly predictive of spontaneous but not evoked saccade direction. In response to the suggestion that we use head velocity rather than displacement in our analyses, we now present both head displacement and velocity for spontaneous and evoked saccades. Although noisier, the velocity data similarly show that, starting roughly 200 ms before saccade onset, slow head movements tend to precede and predict the direction of the ensuing spontaneous saccades.

In addition, we agree with the reviewers’ suggestion in (B) that the fast attempted head movement occurring right around the time of the saccade is similar for spontaneous and evoked gaze shifts. We have attempted to more clearly explain in our revision that the major difference between spontaneous and evoked gaze shifts is the presence or absence of slow-phase movements preceding the saccade.

We respectfully disagree that it is not surprising that there is no preceding head movement since mice don’t know when an airpuff stimulus would occur. Mice also do not know when a visual stimulus will occur. However, visually evoked gaze shifts (e.g., as shown in Figure 6G of Meyer et al., 2020) start with slow head movements followed 100-200 ms later by saccades coupled to fast head movements, similar to spontaneous gaze shifts. Thus, it is quite surprising that touch-evoked gaze shifts do not begin with slow attempted head movements followed 100200 ms later by saccades. Instead, we found, touch-evoked saccades begin on average 30 ms after stimulus delivery, roughly coincident with a fast attempted head movement.

Finally, we agree with (C) that the onsets of saccades and the fast phase of attempted head movements are very close. As the SC microstimulation data we have added to the revision suggest, this close coupling is likely the result of a shared motor command, analogous to what Bizzi and colleagues first described in primates. We therefore place less emphasis in our revision on estimating the precise temporal ordering and instead emphasize that the main difference between spontaneous and evoked gaze shifts is that the latter lack a slow presaccadic head movement. To this end, we have retitled this figure. In addition, we have amended our discussion accordingly.

8) The authors clearly demonstrate that there is a higher probability of eye saccade when the eye is initially offset in the opposite direction (Figure 2I-J). This is exactly what one would expect for a reset saccade resulting from head movement, as the eye is reaching the end of its range in that direction. Likewise, they show that amplitude of the eye movement is equal and opposite to initial eye position (Supp Figure 8), again exactly what one would expect for a reset saccade.

This is an interesting point and one we now mention in our revised discussion. Absent freely moving data, we agree it is important to entertain alternative explanations, such as whether these saccades are resetting. However, we consider this explanation less plausible for several reasons. First, resetting saccades occur to compensate for the eye nearing the end of its range of travel during the slow (100-200 ms) presaccadic head movement (e.g., see Meyer et al., 2020, Figure 5C). As evoked saccades are made at very short latency and there is typically not a presaccadic slow attempted head movement, it is not clear that there are slow counter-rotatory eye movements for which mice would need to reset the eyes. Second, the probabilities of ear airpuff-evoked eye movements are higher when the eyes begin in the center than when the eyes begin on the side towards which the eyes are moving; it is unclear why a reset saccade would be necessary from an already central eye position. In addition, from central eye positions, left and right ear airpuffs evoke saccades in opposite directions, arguing this cannot be the end of the range of travel in either direction. Moreover, for other stimuli, saccade probability as a function of eye positions differs across stimuli such that the probability is lowest for initial eye positions that correspond to the mean endpoint of saccades elicited by that stimulus (Figure 5— figure supplement 2). Third, Figure 5 shows that different stimuli evoke saccades with different endpoints by using different linear relationships between saccade direction/amplitude and initial eye position. This means that saccade amplitude is equal and opposite to the initial eye position only for stimuli that evoke saccades with fairly central endpoints. Instead, for stimuli that evoke saccades with more eccentric endpoints, the amplitude is not equal and opposite to the distance from the center, which the reviewer notes one would expect for a reset saccade. We thank the reviewers for raising this analysis and spurring us to more clearly illustrate these relationships. Nevertheless, because our data are from head-fixed mice, it is possible that there are differences in freely moving mice and we attempt to more fully acknowledge this possibility in the revision.

9) The endpoint offset relative to center seems to be directly proportional to the head movement amplitude (Supp Figure 6A-D). Combined with the previous point, this suggests that these are similar to previously described saccades, but with an overshoot that is determined by head movement amplitude (rather than stimulus location). Such a mechanism makes sense, since for large head movements this would allow greater dynamic range for the ensuing compensatory phase. Alternately, it could be that this overshoot occurs in head-fixed mice where other feedback mechanisms are lacking.

This is an interesting idea and one that we explored but failed to mention in our previous revision. The endpoint offset relative to the center is not proportional to the head movement amplitude. For example, ear airpuff stimuli elicit saccades directed to average endpoints roughly ninefold more eccentric than those elicited by whisker airpuffs (average eccentricity 5.4° vs. 0.6°) but ear airpuffs elicit attempted head movements that are only roughly twice as large (Figure 3; Figure 4—figure supplement 1A, B, red traces in middle right panels). In addition, ear tactile stimuli and auditory airpuffs elicit roughly equal distributions of attempted head movement amplitudes (Figure 4—figure supplement 1C, D, red traces in middle right panels) even though ear tactile-evoked saccades have eccentric (on average, 2.7°) and well separated endpoints whereas auditory airpuff-evoked saccades have central (on average, 0.1°) and minimally separated central endpoints (Figure 3C, D). Moreover, we have performed an analysis to control for any differences in attempted head movement amplitudes by comparing trials with matched head movement amplitudes across stimulus conditions. In this analysis, whisker and ear airpuffs continue to evoke well-separated saccade endpoint distributions, with ear airpuff-evoked saccade endpoints far more eccentric (Figure 4—figure supplement 4). As noted previously, the differences in these saccade endpoints at least partly reflect differences in the initial eye positions from which these gaze shifts were made. Nevertheless, we agree that there are caveats related to head-fixing, including the lack of feedback mechanisms, and address this possibility in the revised discussion.

10) The authors make the claim that head movements depend on eye movement, which would be quite a remarkable finding. However, the actual data is buried in the supplement, and turns out to be a very weak correlation, driven by a small number of outliers with unusually large head movements. Furthermore, only one stimulus condition is presented for this analysis, suggesting that it was not significant for other conditions. Finally, it is easy to envision situations where a weak correlation could arise without a causal role of eye position. For example, if the mouse simply alternates between left and right eye movements on each trial – after a left head movement, the eye would tend to be on the left, so on the subsequent trial when it makes a head movement to the right it would appear to be driven by eye position, even though it's just due to a simple behavioral strategy. If the authors want to support this claim, they should show that it applies for all conditions together, that it is consistent across animals (this is a nested design, and it appears that they used N as # of saccades, rather than # of animals), and that it does not depend on extreme head movements that may represent aberrant conditions.

We thank the reviewer for this suggestion. As requested, we present data for each condition both for the population and individual animals after removing “outlier” trials with large head movements (larger than 3 standard deviations) (Figure 5, Figure 5—figure supplement 1, Author response image 7). We find that across the population, whisker airpuffs, auditory airpuffs, and ear tactile stimuli show a clear relationship between initial eye position and attempted head movement direction and amplitude. These effects are also consistent across animals, particularly for stimuli with central endpoints, which elicit mixtures of ipsiversive and contraversive saccades (whisker airpuffs and auditory airpuffs) (Author response image 7) . These effects on amplitude and direction are more variable and less evident for ear airpuff-evoked saccades and only significant for one side, perhaps because ear airpuffs evoke nearly exclusively contraversive saccades. Nevertheless, it is clear that initial eye position influences ear airpuff-evoked head movements as well because the probability of evoked head movements varies with eye position and in a manner that mirrors saccade probability (Figure 5). Thus, for every stimulus we tested that elicited gaze shifts, initial eye position influences direction, amplitude, and/or probability of attempted head movements. Interestingly, initial eye position has a similar effect on spontaneous saccades, suggesting that these gaze shifts may also take into account initial eye position. We thank the reviewer for suggesting ways to more convincingly illustrate and for their kind words regarding what we agree is “quite a remarkable” finding.

**Author response image 7. sa2fig7:** Evoked attempted head movement amplitude varies according to initial eye position across individual mice. (A-D) Attempted head movement amplitude as a function of initial eye position for left and right stimulus-evoked gaze shifts for an example animal. Each dot corresponds to a single gaze shift. (E-F) Summary of the slopes of the lines of best fit for initial eye position vs. attempted head movement data for 5 mice after removing outlier head movements. Asterisks indicate p < 0.05 using Student’s t-test.

Regarding the idea that mice are alternating leftward and rightward saccades, it is worth noting that we deliberately do not present the stimuli in an alternating pattern to prevent mice from using strategies such as this. Instead, stimuli are presented in a pseudorandom sequence (e.g., in sample trace in Figure 1), such that a left stimulus is equally likely to be followed by a left or a right stimulus, and vice versa. Thus, a strategy of alternation could not produce the results we observe. In addition, stimuli are presented infrequently, mice do not saccade in response to every stimulus presentation, and between stimulus presentations mice often make spontaneous saccades. The most parsimonious explanation we can offer is that eye position is the driver of these effects, but we agree that without causal studies of the role of eye position this relationship could be correlative and driven by other unidentified factors.

11) If these are sensory-guided directed saccades, then why do stimuli at the same location result in different saccade locations (2A vs C) and some saccades are directed away from the stimulus (clearly seen in 2G)? At best, these should be described as "directionally biased" rather than "directed".

We thank the reviewer for raising this issue. There are several candidate reasons that the ear airpuff and tapper stimuli in Figure 2A and C (now Figure 3A and C) have slightly different endpoints. First, it is difficult to precisely ascertain the relative locations the airpuff and tapper bar stimulate. It is possible that they stimulate different parts of the ear. They also approach the ear from different directions and motion direction may factor into saccade selection. Second, the airpuff stimulus is multisensory, consisting of a mix of a tactile stimulus and an auditory stimulus, and we do not know how these modalities synergize. Third, it is likely that the intensities of these stimuli differ, which may underlie the difference in saccade probabilities, and it is possible that stimulus location and intensity jointly specify saccade parameters. Fourth, and relevant to the comment regarding Figure 2G (now Figure 3G), the arrow plot shows every saccade that occurred in a short time window following stimulus delivery. As noted in a previous response, since the stimuli do not evoke saccades on every trial, and in some instances a spontaneous gaze shift has begun before stimulus delivery, these plots include some spontaneous saccades, which are on average recentering. Given that spontaneous saccade probability is fairly fixed, we would expect a larger fraction of the saccades made in this peristimulus period to be spontaneous (i.e., net recentering) for stimuli that evoke saccades less frequently, e.g, the air tactile stimulus relative to the ear airpuff stimulus. This appears to be the case. In addition, the inclusion of more spontaneous saccades, which are on average recentering, may partly underlie the central shift in the distribution of endpoints of ear tactile-evoked saccades.

12) However, a remaining concern is the claim that the data reveal "A new type of mouse gaze shift is led by directed saccades" as stated in the title. The data are fully consistent with the "saccade and fixate" pattern that is widely shared across species (e.g., Land (2019)) and which has recently been identified in freely-moving mice (Michaiel et al. 2020; Meyer et al. 2020). The present study shows that in mice this pattern appears more flexible than previously thought which is a new and important finding. Nevertheless, the rather small (yet important) variations compared to the freely-moving data might not justify the classification as a new type of gaze shift. In particular, Figure 3 suggests that the relative initial contributions of the head and eyes to gaze shifts lie along a continuum and even spontaneous saccades (the baseline condition) can "lead" attempted head rotations; for airpuff-evoked saccades, the head closely follows those saccades (median latency of 30 ms after saccade onset) which corresponds to about half the typical saccade duration (e.g., Sakatani and Isa (2005)). In other words: the saccade and fixate head-eye coupling is preserved for the airpuff-evoked saccades but the relative contributions of the head and eyes are shifted along the continuum. The worry is that the claim of a "new type of mouse gaze shift" might lead to a misperception of this study that shows stronger flexibility of an existing, but not a new, gaze pattern in mice.

We agree that our claims were somewhat strong and have modulated our descriptions throughout, including the title.

13) Figure 2 could use some improvement. They should probably plot saccade start vs saccade end (this data should help their case, but it is presented awkwardly). They should also include average saccade sizes (with direction as sign) versus starting points. Attempted head movement data needs to be looked at in all these scenarios too. It is unclear how well a single trial attempted head movement can be captured. Individual trials look pretty noisy, but the average dynamics look pretty informative. Results may be similar to the eye movements.

We thank the reviewers for this suggestion. Although the arrow plots previously shown in Figure 2 (now Figure 3) do indicate saccade start vs. saccade end, we have added scatter plots of initial eye position vs. saccade amplitude to Figure 5, plots of average saccade sizes and endpoints in Figure 5—figure supplement 1, and scatter plots of initial eye position vs. saccade endpoint in Figure 3—figure supplement 1. We include in Figure 5 and Figure 5—figure supplement 1 scatter plots and summary data of attempted head movements as a function of eye position. As the reviewers predicted and as discussed in more depth in previous responses, the results are very similar to those for eye movements.

14) Another concern is that head-fixed non-human primates will learn to reduce attempted head movements over a short period, once they learn it will not accomplish anything. It is possible the results in Figure 3 are from a similar mechanism. The directed saccades might have provided more reinforcement for learning the failure to move and that is why you rarely see the attempted head movement before the saccade (and why attempted head movements overall appear to be weaker). In addition, spontaneous saccades may decrease over time as the animal is habituated to the experiments and therefore may include fewer saccades during the post-learning period. We recommend that the data from Figure 3 therefore be plotted over time to see if anything observed is from an effect of learning.

We thank the reviewers for this suggestion. We include analyses in our revision that examine both within sessions (dividing each into 15-minute segments) and across sessions (Figure 4— figure supplement 3). Interestingly, touch-evoked head movements are significantly smaller on the fifth session than on the previous 4 sessions (p < 0.05) but spontaneous head movements are not changed. However, in every time segment of every session (15/15), evoked saccades are coupled to smaller head movements than are spontaneous saccades. Therefore, although we thank the reviewers for raising this interesting idea and find that mice appear to gradually learn the futility of evoked attempted head movements, the differences in attempted head movement amplitudes for evoked and spontaneous gaze shifts are not attributable to learning.